# Selective inhibition reveals the regulatory function of DYRK2 in protein synthesis and calcium entry

Tiantian Wei[1,2,3†], Jue Wang[4†], Ruqi Liang[1,2,4†], Wendong Chen[5†], Yilan Chen[6], Mingzhe Ma[4], An He[7], Yifei Du[4], Wenjing Zhou[8], Zhiying Zhang[1], Xin Zeng[1,4], Chu Wang[1,4], Jin Lu[9,10], Xing Guo[11], Xiao-Wei Chen[1,8], Youjun Wang[6], Ruijun Tian[5,6*], Junyu Xiao[1,2,12*], Xiaoguang Lei[1,2,4,13*]

[1]The State Key Laboratory of Protein and Plant Gene Research, School of Life Sciences, Peking University, Beijing, China; [2]Peking-Tsinghua Center for Life Sciences, Peking University, Beijing, China; [3]Academy for Advanced Interdisciplinary Studies, Peking University, Beijing, China; [4]Beijing National Laboratory for Molecular Sciences, Key Laboratory of Bioorganic Chemistry and Molecular Engineering of Ministry of Education, College of Chemistry and Molecular Engineering, Peking University, Beijing, China; [5]SUSTech Academy for Advanced Interdisciplinary Studies, Southern University of Science and Technology, Shenzhen, China; [6]Beijing Key Laboratory of Gene Resource and Molecular Development, Key Laboratory of Cell Proliferation and Regulation Biology, Ministry of Education, College of Life Sciences, Beijing Normal University, Beijing, China; [7]Department of Chemistry, Southern University of Science and Technology, Shenzhen, China; [8]Institute of Molecular Medicine, Peking University, Beijing, China; [9]Peking University Institute of Hematology, People's Hospital, Beijing, China; [10]Collaborative Innovation Center of Hematology, Suzhou, China; [11]Life Sciences Institute, Zhejiang University, Hangzhou, China; [12]Beijing Advanced Innovation Center for Genomics (ICG), Peking University, Beijing, China; [13]Institute for Cancer Research, Shenzhen Bay Laboratory, Shenzhen, China

*For correspondence:
tianrj@sustech.edu.cn (RT);
junyuxiao@pku.edu.cn (JX);
xglei@pku.edu.cn (XL)

†These authors contributed equally

**Abstract:** The dual-specificity tyrosine phosphorylation-regulated kinase DYRK2 has emerged as a critical regulator of cellular processes. We took a chemical biology approach to gain further insights into its function. We developed C17, a potent small-molecule DYRK2 inhibitor, through multiple rounds of structure-based optimization guided by several co-crystallized structures. C17 displayed an effect on DYRK2 at a single-digit nanomolar $IC_{50}$ and showed outstanding selectivity for the human kinome containing 467 other human kinases. Using C17 as a chemical probe, we further performed quantitative phosphoproteomic assays and identified several novel DYRK2 targets, including eukaryotic translation initiation factor 4E-binding protein 1 (4E-BP1) and stromal interaction molecule 1 (STIM1). DYRK2 phosphorylated 4E-BP1 at multiple sites, and the combined treatment of C17 with AKT and MEK inhibitors showed synergistic 4E-BP1 phosphorylation suppression. The phosphorylation of STIM1 by DYRK2 substantially increased the interaction of STIM1 with the ORAI1 channel, and C17 impeded the store-operated calcium entry process. These studies collectively further expand our understanding of DYRK2 and provide a valuable tool to pinpoint its biological function.

## Editor's evaluation

This manuscript will be of interest to researchers studying signal transduction and protein kinase function. The authors developed a selective ATP-competitive inhibitor (C17) for the kinase DYRK2 and use this reagent in combination with global phosphoproteomics to identify potential cellular substrates. Two substrate proteins, 4E-BP1 and STIM1 were biochemically validated as DYRK2 substrates. C17 will likely be a useful pharmacological reagent for exploring DYRK2 function and the phosphoproteomic dataset generated in this study serves as a good starting point for understanding the roles that DYRK2 plays in signaling.

## Introduction

Dual-specificity tyrosine phosphorylation-regulated kinases (DYRKs) belong to the CMGC group of kinases together with other critical human kinases, such as cyclin-dependent kinases (CDKs) and mitogen-activated protein kinases (MAPKs) (*Aranda et al., 2011*; *Becker and Joost, 1999*; *Manning et al., 2002*). DYRKs uniquely phosphorylate tyrosine residues within their activation loops in cis during biosynthesis, although mature proteins display exclusive serine/threonine kinase activities (*Lochhead et al., 2005*). There are five DYRKs in humans: DYRK1A, DYRK1B, DYRK2, DYRK3, and DYRK4. DYRK1A has been extensively studied due to its potential function in the pathogenesis of Down syndrome and neurodegenerative disorders (*Becker and Sippl, 2011*; *Wegiel et al., 2011*). DYRK3 has been shown to function as a central 'dissolvase' to regulate the formation of membrane-less organelles (*Rai et al., 2018*; *Wippich et al., 2013*). On the other hand, DYRK2 is a crucial regulator of 26S proteasome activity (*Guo et al., 2016*).

The 26S proteasome degrades the majority of proteins in human cells and plays a central role in many cellular processes, including the regulation of gene expression and cell division (*Collins and Goldberg, 2017*; *Coux et al., 1996*). Recent discoveries have revealed that the 26S proteasome is subjected to intricate regulation by reversible phosphorylation (*Guo et al., 2017*; *Guo et al., 2016*; *Liu et al., 2020*). DYRK2 phosphorylates the Rpt3 subunit in the regulatory particle of the proteasome at Thr25, leading to the upregulation of proteasome activity (*Guo et al., 2016*). DYRK2 is overexpressed in several tumors, including triple-negative breast cancer and multiple myeloma, which are known to rely heavily on proteasome activity for progression, and perturbation of DYRK2 activity impedes cancer cell proliferation and inhibits tumor growth (*Banerjee et al., 2018*; *Banerjee et al., 2019*).

Our knowledge of the physiological functions of DYRK2 remains in its infancy, and DYRK2 likely has cellular targets in addition to Rpt3. Substrates of many kinases, especially Ser/Thr kinases, remain insufficiently identified. A major obstacle to discovering physiologically relevant substrates of a kinase is the lack of highly specific chemical probes that allow precise modulation of kinase function. Some DYRK2 inhibitors have been reported; however, these compounds also inhibit other kinases, mostly other DYRK family members, to various degrees (*Chaikuad et al., 2016*; *Jouanne et al., 2017*). We have recently identified LDN192960 as a selective DYRK2 inhibitor and showed that LDN192960 could alleviate multiple myeloma and triple-negative breast cancer progression by inhibiting DYRK2-mediated proteasome phosphorylation (*Banerjee et al., 2019*). To obtain even more potent and selective DYRK2 inhibitors, we applied a structure-guided approach to further engineer chemical compounds based on the LDN192960 scaffold. One of the best compounds we generated, compound C17 (C17), displays an effect on DYRK2 at a single-digit nanomolar $IC_{50}$ with moderate to excellent selectivity against kinases closely related to DYRK2. Using this potent DYRK2 inhibitor as a tool, we treated U266 cells with C17. We performed quantitative phosphoproteomic analyses, which led to identifying several novel DYRK2 targets, including eukaryotic translation initiation factor 4E-binding protein 1 (4E-BP1) and stromal interaction molecule 1 (STIM1). These results demonstrate that DYRK2 plays critical regulatory roles in multiple cellular processes, including protein translation and store-operated calcium entry, and indicate that C17 can serve as a valuable probe for the study of DYRK2 function.

# Results

## Structure-based optimization of DYRK2 inhibitors

LDN192960 was identified as a DYRK2 inhibitor (*Banerjee et al., 2019*; *Cuny et al., 2010*; *Cuny et al., 2012*). It occupies the ATP-binding pocket of DYRK2 and mediates extensive hydrophobic and hydrogen bond interactions (*Banerjee et al., 2019*). Nevertheless, LDN192960 also inhibits other DYRK2-related kinases, especially Haspin and DYRK3 (*Banerjee et al., 2019*). To generate DYRK2 inhibitors with better selectivity, we synthesized a series of new compounds based on the same acridine core structure (*Table 1*). The amine side chain was first changed to a protected amine (compounds 1–3), a cyano group (compound 4), or a cyclic amine (compounds 5–6) (*Figure 1—figure supplement 1A*, *Table 1*). Among these candidates, compound 6 exhibited the most potent inhibitory effect towards DYRK2, with an in vitro $IC_{50}$ of 17 nM. In comparison, LDN192960 showed an $IC_{50}$ of 53 nM when the same protocol was used (*Table 1*)—treating HEK293T cells transiently expressing DYRK2 with increasing concentrations of compound 6 efficiently inhibited Rpt3-Thr25 phosphorylation, with the maximal effect observed at an inhibitor concentration of less than 3 µM (*Figure 1—figure supplement 1B*). Notably, compound 6 also displays good selectivity towards DYRK2 than other kinases, including DRYK1A, DRYK1B, DYRK3, Haspin, and MARK3 ($IC_{50}$ values of 889, 697, 121305, 45, and 100 nM, respectively; *Table 1*). Therefore, compound 6 was chosen as the lead compound for further chemical modification.

We subsequently crystallized DYRK2 in complex with compound 6 and determined the structure at a resolution of 2.2 Å (*Figure 1A*, *Figure 1—figure supplement 1C*). Not surprisingly, compound 6 binds the ATP-binding site of DYRK2 like LDN192960. A water molecule is located deep inside the binding pocket acting as a bridge in the interactions between LDN192960 and the protein. The newly added amino side chain displays apparent densities and adopts an extended conformation. An in-depth analysis of the crystal structure revealed several additional sites for chemical expansion that may further strengthen its interaction with DYRK2 (*Figure 1—figure supplement 2A-B*). First, a hydrophilic group can be introduced into the acridine core to functionally replace the aforementioned water molecule and maintain constant contact with DYRK2. Second, a bulky functional group can replace the methoxy groups to mediate other interactions with DYRK2. Finally, the amine side chain can be altered to stabilize its conformation (*Figure 1—figure supplement 2A-B*). To this end, we synthesized 9 new compounds (compounds 7–15) and evaluated their inhibitory effects on DYRK2 and related kinases (*Figure 1—figure supplement 2C-D*). We also determined the co-crystallized structures of several of these compounds with DYRK2 to visualize their detailed interactions (*Figure 1*, *Figure 1—figure supplement 3*). Compound 7, introducing a hydroxymethyl group to the acridine core, inhibits DRYK2 efficiently as compound 6 while displaying better selectivity against other DRYK family members (*Table 1*). The co-crystallized structure shows that the hydroxymethyl group directly contacts the main chain amide group of Ile367 and indirectly coordinates Glu266 and Phe369 via a water molecule (*Figure 1D*). Compared to compound 7, compounds 8–10, which contain a carboxyl, aminomethyl, and fluoromethyl group, respectively, instead of a hydroxymethyl group, display reduced inhibition towards DYRK2. Compounds 11–15, designed to replace the methoxy group with a bulkier side chain, showed significantly decreased activity and selectivity and were not further pursued (*Table 1*).

Further chemical modification was carried out based on compound 7. By changing the 6-membered ring to a straight-chain or smaller ring, we synthesized compounds 16–19 (*Table 1*, *Figure 1—figure supplement 2E*). Among these compounds, C17 with an (*S*)–3-methylpyrrolidine side-chain exhibited the best potency and selectivity among all the analogs (*Table 1*). Interestingly, we noticed that compound 18 containing an (*R*)–3-methylpyrrolidine side chain was not as good as C17, indicating that the chirality of the 3-methylpyrrolidine motif plays an essential role in both potency and selectivity. Further modification of compound 17 (leading to compound 20) to promote further hydrogen bond interactions with DYRK2 failed to improve the inhibitory effect. We also wondered whether acridine was the best aromatic core structure and synthesized two new compounds (compounds 21 and 22) by changing one side of the benzene group to a sulfur-containing thiazole structure (*Figure 1—figure supplement 2F*), which we thought might facilitate hydrophobic interactions with DYRK2 within the ATP-binding pocket; however, they did not have as effective an inhibitory effect as compound 17 (*Table 1*).

**Table 1.** The Inhibitory activity and selectivity of acridine analogs of DYRK2.

| Cmpd. | R1 | R2 | R3 | R4 | IC50 at molecular level (nM) | | | | | | Selectivity | | | | |
|---|---|---|---|---|---|---|---|---|---|---|---|---|---|---|---|
| | | | | | DYRK2 | DYRK1A | DYRK1B | DYRK3 | Haspin | MARK3 | DYRK2& DYRK1A | DYRK2& DYRK1B | DYRK2& DYRK3 | DYRK2& Haspin | DYRK2& MARK3 |
| LDN192960 | | -CH3 | -CH3 | -H | 53 ± 2 | 1859 ± 30 | 2900 ± 39 | 22 ± 4 | 18 ± 2 | 611 ± 19 | 35 | 55 | ~ | ~ | 12 |
| 1 | -NHBoc | -CH3 | -CH3 | -H | 38 ± 2 | 651 ± 29 | 1401 ± 91 | 115 ± 4 | 34 ± 3 | 36 ± 2 | 17 | 17 | 3 | ~ | ~ |
| 2 | -NHAc | -CH3 | -CH3 | -H | 31 ± 1 | 731 ± 36 | 1477 ± 128 | 94 ± 9 | 27 ± 3 | 27 ± 5 | 24 | 48 | 3 | ~ | ~ |
| 3 | -NHAc | -CH3 | -CH3 | -H | 41 ± 2 | 1018 ± 78 | 2495 ± 88 | 157 ± 18 | 24 ± 1 | 33 ± 7 | 25 | 61 | 4 | ~ | ~ |
| 4 | -CN | -CH3 | -CH3 | -H | 53 ± 2 | 964 ± 14 | 1386 ± 21 | 234 ± 10 | 30 ± 1 | 96 ± 3 | 18 | 26 | 4 | ~ | 2 |
| 5 | -NH | -CH3 | -CH3 | -H | 89 ± 2 | 1026 ± 96 | 3488 ± 86 | 311 ± 22 | 53 ± 4 | 91 ± 5 | 12 | 39 | 3 | ~ | 1 |
| 6 | -NH | -CH3 | -CH3 | -H | 20 ± 3 | 889 ± 131 | 697 ± 67 | 110 ± 11 | 45 ± 3 | 100 ± 4 | 44 | 35 | 6 | 2 | 5 |
| 7 | -NH | -CH3 | -CH3 | -CH2OH | 13 ± 1 | 2844 ± 49 | 2049 ± 116 | 26 ± 2 | 65 ± 5 | 107 ± 4 | 219 | 158 | 2 | 5 | 8 |
| 8 | -NH | -CH3 | -CH3 | -COOH | 342 ± 77 | 7713 ± 1,249 | 6311 ± 1,380 | 8009 ± 130 | 308 ± 26 | 1613 ± 24 | 23 | 18 | 23 | ~ | 5 |
| 9 | -NH | -CH3 | -CH3 | -CH2NH2 | 797 ± 26 | 8774 ± 508 | 7799 ± 81 | 665 ± 28 | 716 ± 48 | 3390 ± 301 | 11 | 10 | 2 | ~ | 4 |
| 10 | -NH | -CH3 | -CH3 | -CF2H | 522 ± 210 | 53206 ± 16,384 | 47964 ± 3,582 | 402 ± 13 | 163 ± 21 | 460 ± 25 | 102 | 92 | ~ | ~ | ~ |
| 11 | -NH2 | -Bn | -CH3 | -H | 646 ± 164 | 139908 ± 677 | 4975 ± 328 | 2026 ± 600 | 1608 ± 52 | 555 ± 36 | 217 | 8 | 3 | 3 | ~ |
| 12 | -NH | -Bn | -CH3 | -H | 427 ± 109 | 12504 ± 3,260 | 8203 ± 674 | 539 ± 353 | 1085 ± 139 | 1062 ± 54 | 29 | 19 | 1 | 3 | 2 |

*Table 1 continued*

| Cmpd. | R1 | R2 | R3 | R4 | IC50 at molecular level (nM) | | | | | | Selectivity | | | | |
|---|---|---|---|---|---|---|---|---|---|---|---|---|---|---|---|
| | | | | | DYRK2 | DYRK1A | DYRK1B | DYRK3 | Haspin | MARK3 | DYRK2& DYRK1A | DYRK2& DYRK1B | DYRK2& DYRK3 | DYRK2& Haspin | DYRK2& MARK3 |
| 13 | NH | -Bn | -CH₃ | -H | 124 ± 27 | 21608 ± 3,431 | 2812 ± 543 | 1142 ± 129 | 1588 ± 40 | 359 ± 17 | 174 | 23 | 9 | 13 | 3 |
| 14 | NH | -iPr | -CH₃ | -H | 85 ± 17 | 984 ± 127 | 3787 ± 234 | 93 ± 28 | 300 ± 21 | 215 ± 12 | 12 | 45 | 1 | 4 | 3 |
| 15 | NH | -Bn | -Bn | -H | 623 ± 18 | 19244 ± 1,551 | 21110 ± 1,388 | 496 ± 36 | 18643 ± 1,365 | 1183 ± 127 | 31 | 34 | ~ | 30 | 2 |
| 16 | NH₂ | -CH₃ | -CH₃ | -CH₂OH | 25 ± 9 | 2243 ± 74 | 2257 ± 279 | 33 ± 6 | 90 ± 9 | 134 ± 8 | 90 | 90 | 1 | 4 | 5 |
| 17 | NH | -CH₃ | -CH₃ | -CH₂OH | 9 ± 2 | 2145 ± 100 | 2272 ± 134 | 68 ± 5 | 26 ± 5 | 87 ± 7 | 240 | 252 | 8 | 3 | 10 |
| 18 | NH | -CH₃ | -CH₃ | -CH₂OH | 18 ± 2 | 1250 ± 95 | 1222 ± 168 | 73 ± 13 | 16 ± 3 | 116 ± 13 | 69 | 68 | 4 | ~ | 6 |
| 19 | NH | -CH₃ | -CH₃ | -CH₂OH | 23 ± 3 | 1531 ± 52 | 3443 ± 294 | 108 ± 17 | 50 ± 1 | 210 ± 4 | 67 | 150 | 5 | 2 | 9 |
| 20 | NH | -CH₃ | -CH₃ | -CH₂NC(NH₂)₂ | 1498 ± 104 | 21535 ± 1910 | 25850 ± 1,571 | 8477 ± 655 | 26509 ± 733 | 25535 ± 1,385 | 14 | 16 | 6 | 18 | 17 |
| 21 | | | | | 159 ± 7 | 3014 ± 137 | 3514 ± 511 | 69 ± 6 | 1564 ± 252 | 1315 ± 87 | 19 | 22 | ~ | 10 | 8 |
| 22 | | | | | 3761 ± 202 | 24733 ± 1,669 | 25948 ± 540 | 2426 ± 257 | 9750 ± 127 | 16770 ± 1,788 | 7 | 7 | ~ | 3 | 4 |

The online version of this article includes the following source data for table 1:

**Source data 1.** Raw data of inhibitors against kinases for *Table 1*.

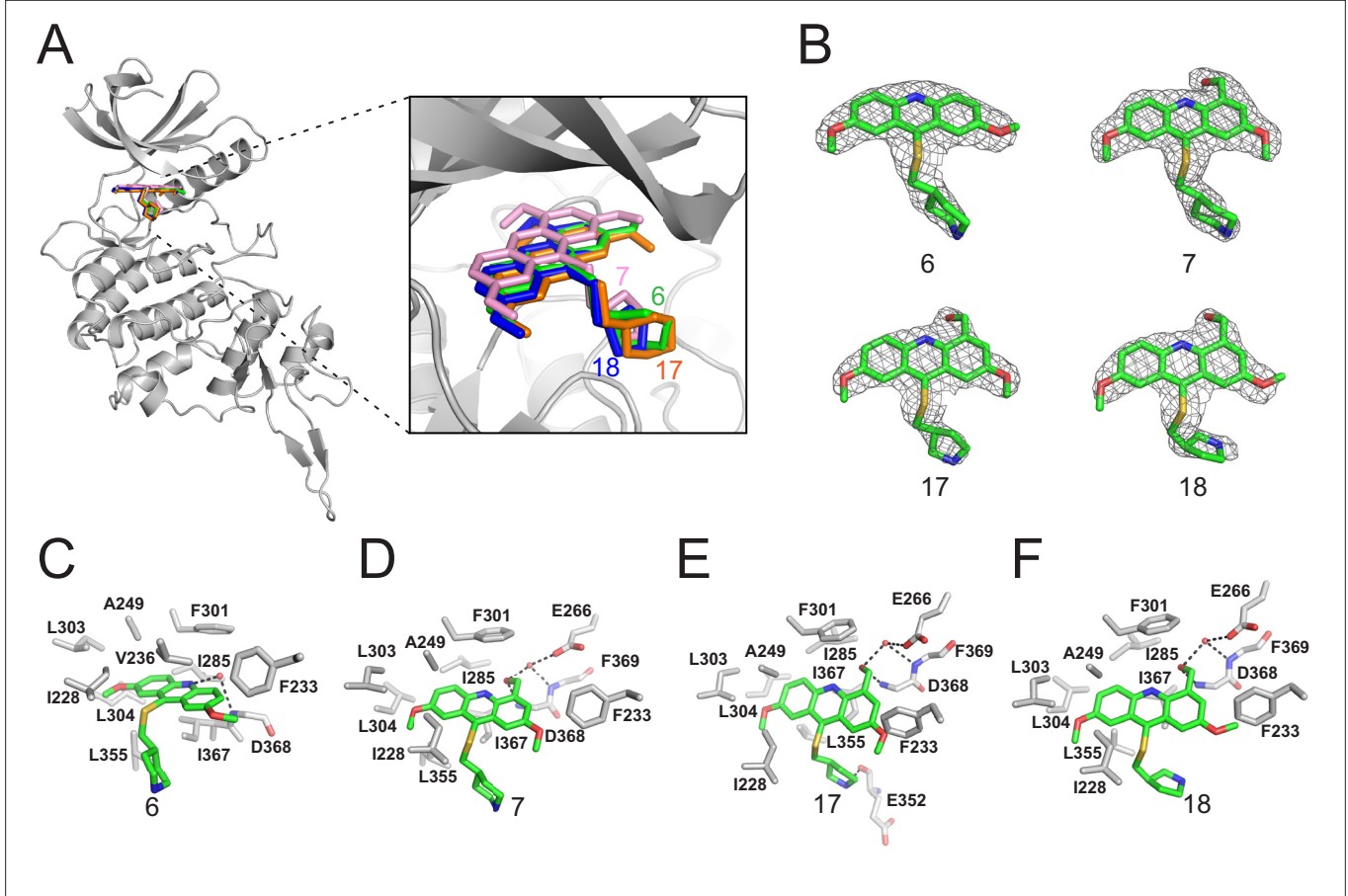

**Figure 1.** Crystal structures of DYRK2 bound to novel inhibitors. (**A**) Overall structure of DYRK2 (grey) bound to 6 (green), 7 (pink), C17 (orange), and 18 (blue). (**B**) Composite omit maps are contoured at 1.5σand shown as gray meshes to reveal the presence of compounds 6, 7, 17, and 18 in the respective crystal structures. (**C–F**) Close-up view of the DYRK2 binding pocket with compounds 6, 7, 17, and 18. Hydrogen bonds are shown as dashed lines. Water molecules are indicated with red spheres.

The online version of this article includes the following source data and figure supplement(s) for figure 1:

**Source data 1.** Data collection and refinement statistics of crystal structures of DYRK2 with different inhibitors.

**Figure supplement 1.** Chemical compounds derived from LDN192960.

**Figure supplement 1—source data 1.** Raw data of western blot for *Figure 1—figure supplement 1B*.

**Figure supplement 2.** Structure-guided engineering of DYRK2 inhibitors based on compound 6.

**Figure supplement 3.** The 2Fo-Fc composite omit maps (1.5 σsurrounding compounds 5, 10, 13, 14, 19, and 20 are shown in the co-crystal structures with DRYK2, respectively).

## C17 is a potent and selective DYRK2 inhibitor

We set to comprehensively characterize the inhibitory function of compound 17 (*Figure 2A*), referred to as C17 hereafter. In vitro, C17 displays an effect on DYRK2 at a single-digit nanomolar $IC_{50}$ value (9 nM) (*Figure 2B*, *Figure 2—figure supplement 1A*). To further evaluate the selectivity of C17, we performed kinome profiling analyses. Among the 468 human kinases tested, C17 targeted only DYRK2, Haspin, and MARK3 at a concentration of 500 nM (*Figure 2C*). Nonetheless, the in vitro $IC_{50}$ values of C17 for Haspin and MARK3 (26 nM and 87 nM, respectively) were 3–10-fold higher than that for DYRK2 (*Figure 2B*, *Figure 2—figure supplement 1B-F*). Similarly, C17 also inhibited DYRK3 to a lesser extent ($IC_{50}$ of 68 nM). In contrast, LDN192960 inhibited DYRK3 and Haspin more than it inhibited DYRK2 (*Banerjee et al., 2019*; *Cuny et al., 2010*). Significantly, C17 also efficiently suppressed DYRK2 activity in the cell and abolished Rpt3-Thr25 phosphorylation at an inhibitor concentration of less than 1 µM (*Figure 2D*). Taken together, these data demonstrate that C17 is a highly potent and selective DYRK2 inhibitor both *in vitro* and *in vivo*.

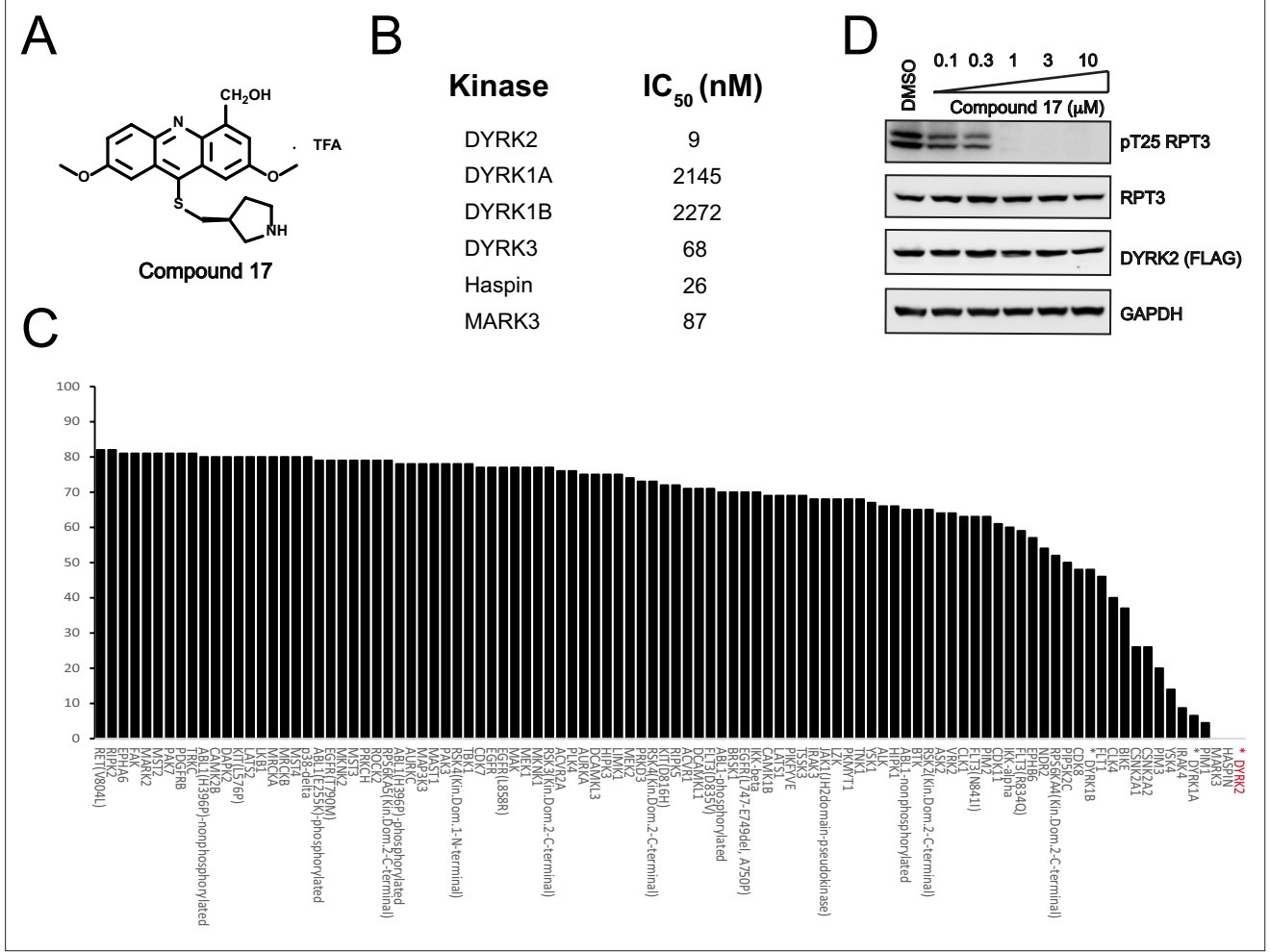

**Figure 2.** C17 is a potent and selective inhibitor of DYRK2. (**A**) Chemical structure of C17. (**B**) IC$_{50}$ values of C17 against DYRK1A, DYRK1B, DYRK3, Haspin and MARK3. (**C**) Kinome profiling of C17 at 500 nM was carried out using 468 human kinases (https://www.discoverx.com/). (**D**) C17 inhibits Rpt3-Thr25 phosphorylation. HEK293T cells stably expressing FLAG-DYRK2 were treated with the indicated concentrations of C17 for 1 hr. The cells were lysed, and immunoblotting was carried out with the indicated antibodies.

The online version of this article includes the following source data and figure supplement(s) for figure 2:

**Source data 1.** Raw data of C17 Kinome profiling list for *Figure 2C*.

**Source data 2.** Raw data of western blot for *Figure 2D*.

**Figure supplement 1.** IC$_{50}$ of C17 on DYRK2 and its main off targets.

**Figure supplement 2.** Binding strength of LDN192960 and C17 with calf thymus DNA.

Acridine derivatives have traditionally been used as antibacterial, antiparasitic, and anticancer agents since these compounds usually show strong DNA intercalating effects (*Chaikuad et al., 2016*; *Jouanne et al., 2017*). Considering the potential toxicity of C17 due to its possible DNA-binding capacity, we also assessed the DNA-binding effect of C17 (*Figure 2—figure supplement 2*). Isothermal titration calorimetry revealed that C17 (Kd = 22.9 μM) binds to DNA with significantly lower affinity than LDN192960 binds to DNA (Kd = 198 nM), possibly because of the introduction of hydroxymethyl group on the acridine core, which is not present in LDN192960.

## DYRK2 substrate profiling by quantitative phosphoproteomic analyses

Quantitative phosphoproteomic approaches have significantly expanded the scope of phosphorylation analysis, enabling the quantification of changes in thousands of phosphorylation sites simultaneously (*Álvarez-Salamero et al., 2017*). To obtain a comprehensive list of potential DYRK2 targets, we

treated the myeloma U266 cells with C17 and carried out quantitative phosphoproteomic analyses (*Chen et al., 2018*; *Hogrebe et al., 2018*). We prepared lysates of U266 cells treated with C17 or the DMSO control and trypsinized them. Phosphorylated peptides were then enriched using $Ti^{4+}$-immobilized metal ion affinity chromatography (IMAC) tips and analyzed by LC-MS/MS (*Figure 3A*). A total of 15,755 phosphosites were identified, among which 12,818 (81%) were serine, and 2,798 (18%) were threonine. A total of 10,647 (68%) phosphosites were Class I (localization probability >0.75), 2557 (16%) were Class II (0.5 < localization probability ≤0.75), and 2401 (16%) were Class III (0.25 < localization probability ≤0.5) (*Figure 3B*). This is a very comprehensive phosphoproteomic dataset prepared for DYRK2 substrate profiling by treating the U266 cells with 10 μM of C17. A good Pearson correlation coefficient of 0.9 was obtained for the phosphosite intensities among the treatment and control samples (*Figure 3—figure supplement 1*), and the coefficient of variance of the intensities of the majority of the phosphosites was lower than 20% (*Figure 3—figure supplement 2*), demonstrating the high quantification precision of our label-free phosphoproteomic analysis. Remarkably, C17 treatment led to significant downregulation of 373 phosphosites (*Figure 3C*), including pThr37 of the eukaryotic translation initiation factor 4E-binding protein 1 (4E-BP1), as well as pSer519 and pSer521 in the stromal interaction molecule 1 (STIM1) (*Figure 3D*, *Figure 3—figure supplement 3*). Interestingly, another 445 phosphosites were upregulated (*Figure 3C*), suggesting that DYRK2 likely inhibited some downstream kinases or activated phosphatases, and suppressing its activity reversed these effects. Together, these data demonstrate that DYRK2 is involved in a network of phosphorylation events and can directly or indirectly regulate the phosphorylation status of many proteins. The top pathways with which DYRK2 may participate were revealed by a global analysis of the significantly up-and down-regulated phosphoproteins (*Figure 3E*).

## 4E-BP1 is a direct cellular target of DYRK2

We set out to determine whether some of the 373 downregulated phosphosites are genuine DYRK2 targets. We first examined 4E-BP1 for several reasons. First, C17 treatment decreased the pThr37 level in U266 cells (*Figure 3C*). Second, a previous study showed that Ser65 and Ser101 in 4E-BP1 can be phosphorylated by DYRK2 in vitro, indicating that 4E-BP1 is a potential DYRK2 substrate (*Wang et al., 2003*). And lastly, several phosphosite-specific antibodies for 4E-BP1 are commercially available. 4E-BP1 is a master regulator of protein synthesis. It has been well established that its phosphorylation by other kinases such as mTORC1 leads to its dissociation from eukaryotic initiation factor 4E (eIF4E), allowing mRNA translation (*Laplante and Sabatini, 2012*; *Ma et al., 2009*).

Using an antibody that detects 4E-BP1 only when it is phosphorylated at Thr37 and Thr46, we found that C17 treatment significantly reduced the level of pThr37/pThr46 of endogenous 4E-BP1 in HEK293T cells (*Figure 4A*), consistent with our mass spec analyses in U266 cells. Further investigations using two other 4E-BP1 phosphosite-specific antibodies showed that C17 also decreased the phosphorylation of Ser65 in endogenous 4E-BP1 (*Figure 4A*) by a previous study *Wang et al., 2003*; as well as Thr70. Knockdown of endogenous DYRK2 using a short hairpin RNA (shRNA) also significantly reduced the phosphorylation of these sites (*Figure 4B*). Successful knockdown is demonstrated by quantitative RT-PCR analysis (*Figure 4—figure supplement 1*). Similarly, C17 suppressed DYRK2-mediated phosphorylation of 4E-BP1 when overexpressed in the HEK293 cells (*Figure 4C*). To ascertain whether DYRK2 can directly phosphate 4E-BP1, we performed an in vitro kinase assay using purified DYRK2 and 4E-BP1 proteins. DYRK2 efficiently phosphorylated 4E-BP1 at multiple sites, including Thr37/Thr46, Ser65, and Thr70, whereas the kinase-deficient DYRK2 mutant (D275N) displayed no activity (*Figure 4D*). C17 suppressed the phosphorylation of these sites in a dose-dependent manner (*Figure 4E*). These results demonstrate that DYRK2 effectively phosphorylated 4E-BP1 on multiple sites in vivo and in vitro.

4E-BP1 is targeted by multiple kinases (*Qin et al., 2016*). Indeed, C17 or DYRK2 shRNA decreased but did not abolish the phosphorylation of 4E-BP1 (*Figure 4A and B*). A previous study showed that combined inhibition of AKT and MEK kinases suppressed 4E-BP1 phosphorylation and tumor growth (*She et al., 2010*). We observed similar results when we treated the HEK293A cells with AKTi (an AKT1/AKT2 inhibitor) and PD0325901 (a MEK inhibitor). Significantly, knockdown of DYRK2 in the presence of these compounds further markedly diminished 4E-BP1 phosphorylation (*Figure 4B*). To assess whether C17 can also elicit a synergistic effect with these kinase inhibitors, we treated HEK293A, HCT116, and U266 cells with these molecules, either alone or in combination, and examined the

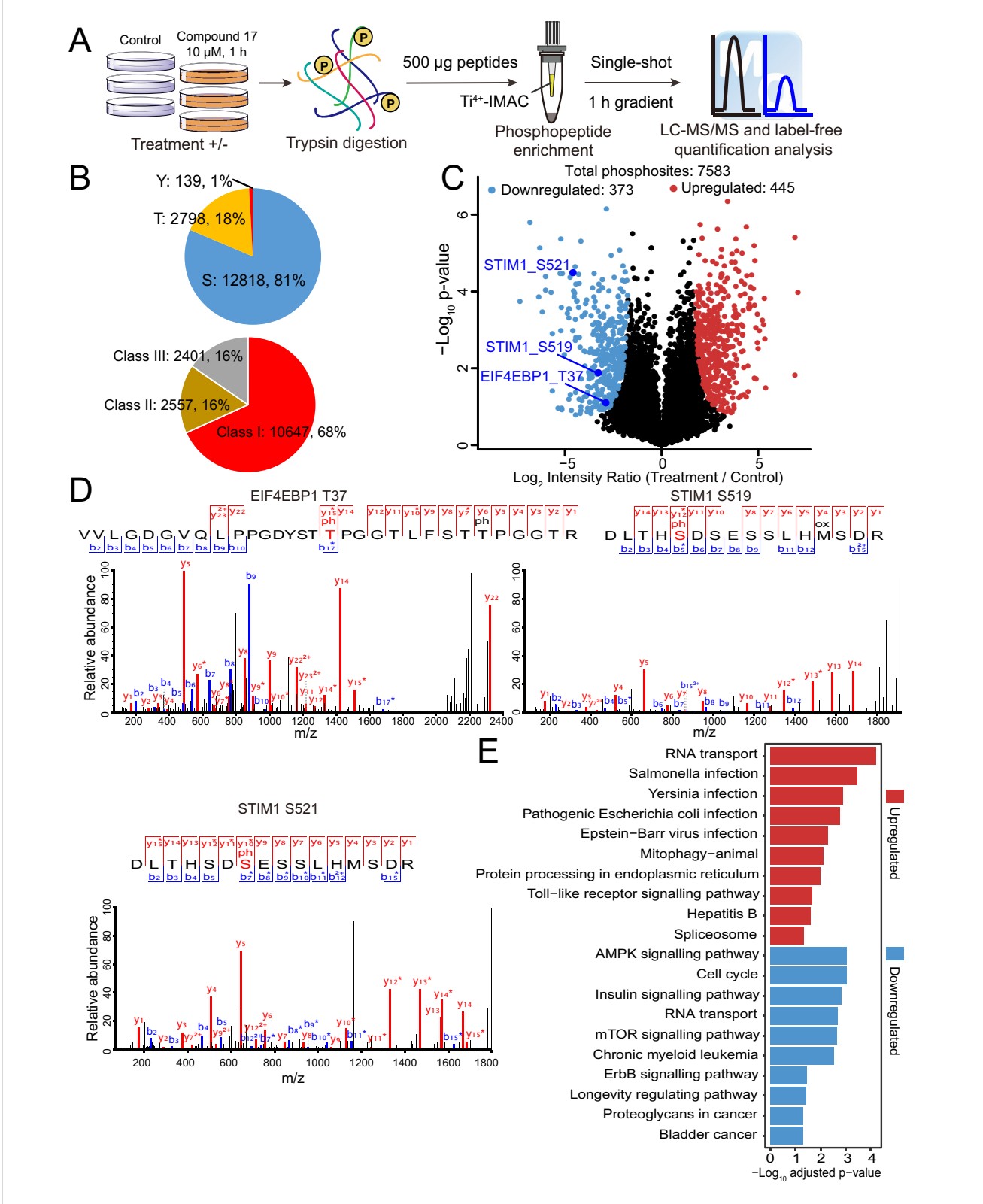

**Figure 3.** Quantitative phosphoproteomic analysis of U266 cells treated with C17. (**A**) Workflow of the phosphoproteomic approach. Triplicate samples treated with/without 10 µM C17 for 1 hr were separately lysed and digested, and the phosphorylated peptides were enriched by the Ti$^{4+}$-IMAC tip and analyzed by LC-MS/MS. (**B**) Distribution of the assigned amino acid residues and their localization probabilities (Class I > 0.75, Class II > 0.5 and ≤ 0.75, Class III > 0.25 and ≤ 0.5) for all identified phosphorylation sites. (**C**) Volcano plot (FDR < 0.05 and S0 = 2) shows the significantly up-and downregulated

*Figure 3 continued on next page*

*Figure 3 continued*

phosphosites after C17 treatment. (**D**) MS/MS spectra of the phosphosites of two potential DYRK2 substrates, pT37 of 4E-BP1 and pS519 and pS521 of STIM1. (**E**) Global canonical pathway analysis of the significantly up-and downregulated phosphoproteins. –Log$_{10}$ adjusted p-values associated with a pathway are presented.

The online version of this article includes the following source data and figure supplement(s) for figure 3:

**Source data 1.** Raw data of the significantly up- and down-regulated phosphosites after U266 cells treated with C17 for *Figure 3C*.

**Figure supplement 1.** Correlation of the intensities of phosphosites between any two samples in phosphoproteomic analysis of U266 cells treated with/without C17.

**Figure supplement 1—source data 1.** Raw data of the intensities of phosphosites in phosphoproteomic analysis of U266 cells treated with/without C17.

**Figure supplement 2.** Coefficient of variance of the intensities of phosphosites in phosphoproteomic analysis of U266 cells treated with/without C17.

**Figure supplement 2—source data 1.** Raw data of the intensities of phosphosites in phosphoproteomic analysis of U266 cells treated with/without C17.

**Figure supplement 3.** The intensities for pT37 phosphosite of EIF4E-BP1 and pS519, pS521 phosphosites of STIM1.

**Figure supplement 3—source data 1.** Raw data of the intensities for pT37 phosphosite of 4E-BP1 and pS519, pS521 phosphosites of STIM1 for *Figure 3—figure supplement 3*.

phosphorylation status of endogenous 4E-BP1 (*Figure 4F–H*). The presence of C17 potentiated the inhibitory effect of AKTi and PD0325901 in all these cells. Together, these results confirm that 4E-BP1 is a direct cellular target of DYRK2 and suggest the potential use of DYRK2 inhibitors in combination with other kinase inhibitors for cancer therapy.

## DYRK2 promotes STIM1-ORAI1 interaction to modulate SOCE

In addition to 4E-BP1, another potential target of DYRK2 is STIM1, as the phosphorylation levels of both Ser519 and Ser521 in endogenous STIM1 were significantly reduced upon DYRK2 inhibition in our mass spectrometry analyses (*Figure 3C*). STIM1 is a single-pass transmembrane protein residing in the endoplasmic reticulum (ER) and plays a vital role in the store-operated calcium entry (SOCE) process (*Collins et al., 2013*). The luminal domain of STIM1 senses calcium depletion in the ER and induces protein oligomerization and puncta formation (*Liou et al., 2005*; *Prakriya and Lewis, 2015*; *Zheng et al., 2018*). Oligomerized STIM1 then travels to the ER-plasma membrane contact site and activates the ORAI1 calcium channel. The cytosolic region of STIM1 contains multiple phosphorylation sites, and it has been shown that the function of STIM1 is regulated by several kinases, including ERK1/2 (*Pozo-Guisado et al., 2013*; *Pozo-Guisado and Martin-Romero, 2013*).

Purified wild-type DYRK2, but not the kinase-dead mutant D275N, induced mobility changes of the cytosolic region of STIM1 (STIM1$^{235-END}$) in SDS-PAGE gel (*Figure 5A*). As increasing amounts of DYRK2 lead to greater shifts of STIM1$^{235-END}$, there are likely multiple DYRK2 phosphorylation sites in STIM1. Consistently, DYRK2 induced a mobility shift of STIM1 when they were co-expressed in the HEK293A cells (*Figure 5B*). To further pinpoint DYRK2-specific phosphorylation sites, we co-expressed DYRK2, Orai1, and STIM1 in HEK293A cells, treated the cells with C17, isolated STIM1 using Anti-FLAG agarose, and then subjected it to label-free quantitative mass spectrometry analyses. The phosphorylation levels of at least eight phosphosites on four peptides of STIM1 were significantly reduced upon treatment with C17 compared with the untreated sample (*Figure 5—figure supplement 1A-B*), including Ser519 and Ser521 that were identified in the U266 phosphoproteome analysis (*Figure 3C*). In a separate mass spec experiment, phosphorylation of Ser608 and Ser616 were also reduced by C17. Together, these results demonstrate that DYRK2 can phosphorylate multiple sites in the cytosolic region of STIM1.

STIM1 puncta formation indicates its oligomerization and activation (*Liou et al., 2005*; *Prakriya and Lewis, 2015*; *Zheng et al., 2018*). To assess the functional outcome of STIM1 phosphorylation by DYRK2, we co-expressed STIM1 and DYRK2 in an Orai-deficient (Orai-KO) cell line, which has all three Orai genes genetically ablated (*Zheng et al., 2018*). DYRK2 induced the appearance of STIM1 puncta under resting conditions, indicating that DYRK2 promotes STIM1 oligomerization (*Figure 5C*). In contrast, the STIM1 puncta were not observed in the presence of C17. DYRK2 also failed to promote the punctate formation of STIM1-10M, a STIM1 variant with all ten potential DYRK2 phosphorylation sites mutated to Ala (*Figure 5C*, *Figure 5—figure supplement 1C*).

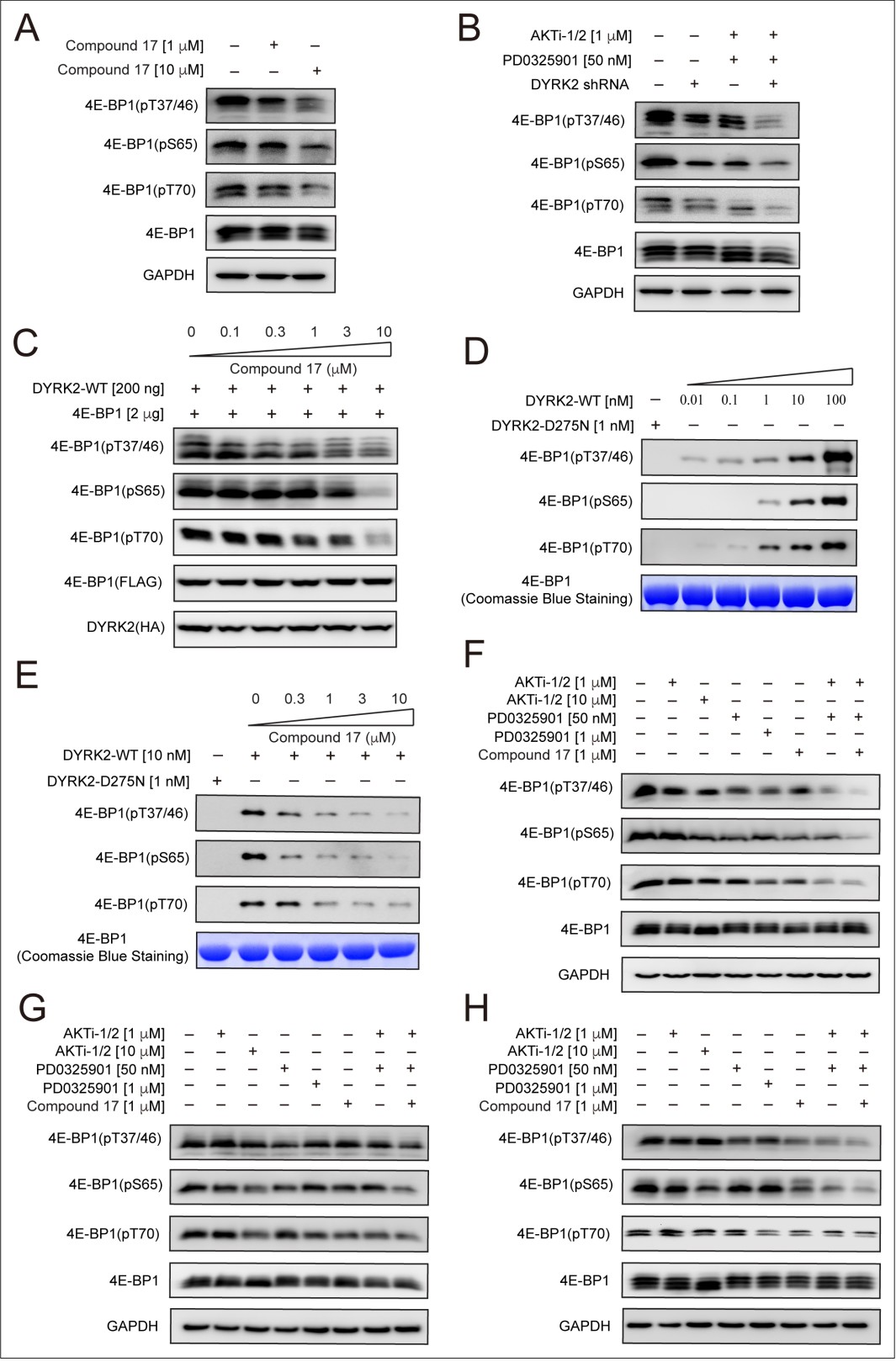

**Figure 4.** 4E-BP1 is a substrate of DYRK2. (**A**) C17 treatment for 1 hr reduced the phosphorylation of endogenous 4E-BP1 in HEK293T cells. The phosphorylation status of 4E-BP1 was analyzed by immunoblotting cell lysates using indicated antibodies. (**B**) DYRK2 knockdown decreases the phosphorylation of endogenous 4E-BP1 in HEK293T cells. (**C**) HEK293A cells stably expressing HA-DYRK2 and FLAG-4E-BP1 were treated with indicated concentrations

*Figure 4 continued on next page*

*Figure 4 continued*

of C17 for 1 hr. The cells were lysed, and immunoblotting was carried out with indicated antibodies. (**D**) DYRK2 directly phosphorylated 4E-BP1 at multiple sites. (**E**) C17 inhibited DYRK2-mediated 4E-BP1 phosphorylation in a concentration-dependent manner. (**F–H**) C17 displayed a synergistic effect with AKT and MEK inhibitors to suppress 4E-BP1 phosphorylation in HEK293A (**F**), HCT116 (**G**), and U266 cells. (**H**) The cells were treated with indicated concentrations of PD032590, AKTi-1/2, and C17 alone or in combination for 1 hr. Cell lysates were immunoblotted with indicated antibodies.

The online version of this article includes the following source data and figure supplement(s) for figure 4:

**Source data 1.** Raw data of Western blot for *Figure 4A, B*.

**Source data 2.** Raw data of western blot for *Figure 4C, D*.

**Source data 3.** Raw data of western blot for *Figure 4E, F*.

**Source data 4.** Raw data of western blot for *Figure 4G, H*.

**Figure supplement 1.** Knockdown efficiency of DYRK2-expression in wild-type HEK293T stably expressed DYRK2 shRNA was measured by qPCR.

**Figure supplement 1—source data 1.** Raw qPCR data of knockdown efficiency of DYRK2-expression in wild-type HEK293T for *Figure 4—figure supplement 1*.

**Figure supplement 2.** All primer sequences for qRT-PCR, shRNA targeting sequences are listed.

To further understand the importance of STIM1 phosphorylation, we examined the interaction between STIM1 and Orai1 using co-immunoprecipitation. Expression of WT DYRK2 significantly increased the interaction between STIM1 and Orai1, whereas expression of DYRK2-KD exerted no such effect (*Figure 5D*). Treating cells with C17 effectively abolished the DYRK2-dependent STIM1-Orai1 interaction. Notably, both STIM1-1-491, a C-terminal truncated STIM1 (*Figure 5C*, *Figure 5—figure supplement 1C*), and STIM1-10M displayed significantly reduced interaction with Orai1 even in the presence of WT DYRK2 (*Figure 5E*), suggesting that DYRK2-mediated phosphorylation is essential to promote the binding between STIM1 and Orai1. C17 also decreased the interaction between STIM1 and Orai1 without exogenously expressing DYRK2 (*Figure 5F*).

We examined fluorescence resonance energy transfer (FRET) between STIM1-YFP and CFP-Orai1 to validate the regulatory function of DYRK2 on STIM1-Orai1 interaction. The FRET signals between STIM1-YFP and CFP-Orai1 were significantly increased in HEK293 cells in the presence of WT DYRK2 (*Figure 5G*). To exclude the influence of endogenous STIM1, we performed further analyses in a STIM1-STIM2 DKO cell line (*Zheng et al., 2018*). The FRET signals between STIM1-1-491 and Orai1 were unaltered by DYRK2 (*Figure 5H*), indicating that the effect of DYRK2 is dependent on the C-terminal region of STIM1. Furthermore, the FRET signals between STIM1-10M and Orai1 were unaffected by DYRK2 (*Figure 5I*). These results are consistent with the co-immunoprecipitation results and demonstrate that DYRK2 can promote the STIM1-Orai1 interaction via STIM1 phosphorylation.

Lastly, to examine the physiological relevance of the STIM1-Orai1 interaction regulated by DYRK2, we performed SOCE analyses in HEK293A cells expressing GCaMP6f, a genetically encoded calcium sensor (*Nakai et al., 2001*). Treating cells grown in a calcium-free medium containing thapsigargin resulted in a transient increase in GCaMP6f fluorescence due to calcium release from the ER to the cytosol (*Figure 5J*, black line). Subsequent addition of calcium to the cell culture medium resulted in a marked increase in GCaMP6f signaling, indicating calcium entry into the cells, further augmented by STIM1 overexpression (*Figure 5J*, blue line). Pre-treating cells with C17 for 1 hr substantially reduced SOCE in cells with either endogenous (*Figure 5J*, green line) or overexpressed STIM1 (*Figure 5J*, red line). Quantifications of these results are present in *Figure 5K*. Taken together, our results strongly suggest that DRYK2 can directly enhance SOCE by phosphorylating STIM1 and promoting its interaction with ORAI1, which can all be effectively inhibited by C17.

## Discussion

We used a structure-based approach to design, synthesize and evaluate a series of new analogs based on the acridine core structure and eventually identified C17 as a potent and selective DYRK2 inhibitor. We showed that C17 affects DYRK2 at a single-digit nanomolar IC50 and inhibits DYRK2 more potently than closely related kinases such as DYRK3, Haspin, and MARK3. The crystal structure

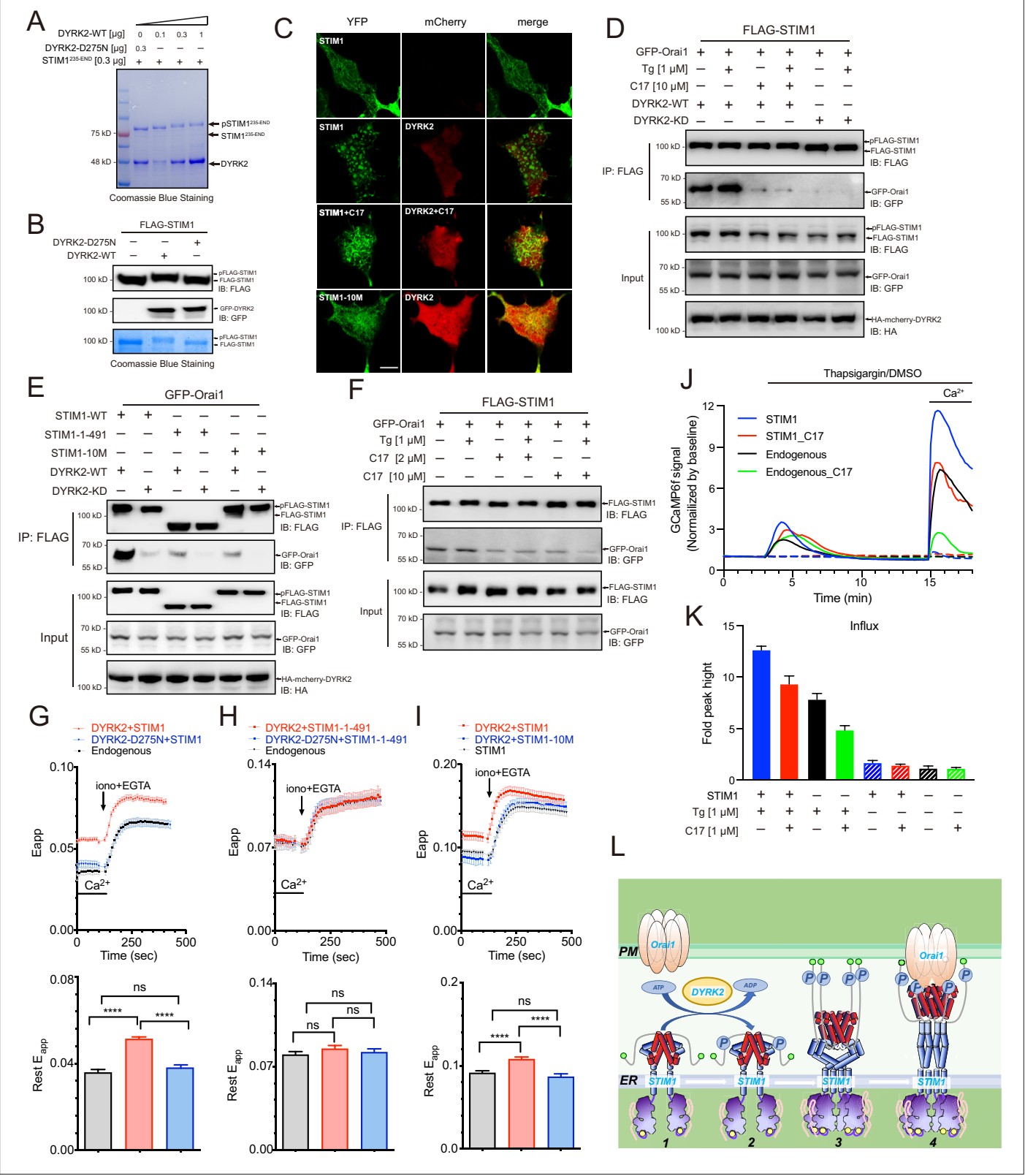

**Figure 5.** Phosphorylation of STIM1 by DYRK2 modulates SOCE. (**A**) DYRK2 directly phosphorylated STIM1. GST-STIM1$^{235-END}$ was incubated with wild-type or kinase-deficient DYRK2 in the presence of Mn-ATP for 30 min. Phosphorylation of GST-STIM1$^{235-END}$ was indicated by the mobility change of STIM1 in SDS-PAGE gel. (**B**) DYRK2 phosphorylated STIM1 *in vivo*. HEK293A cells were co-transfected with FLAG-STIM1 and DYRK2 for 36 h, then states immunoblotted with the indicated antibodies. (**C**) Typical confocal microscopy images showing the effects of mCherry-DYRK2 and/or C17 (1 μM)

*Figure 5 continued on next page*

*Figure 5 continued*

on the puncta formation of STIM1 in the HEK293 Orai1/Orai2/Orai3-TKO cells. The scale bar is 10 µm. The experiments were repeated, six cells were examined each time. (**D**) DYRK2 promoted the interaction between STIM1 and Orai1. HEK293A cells were co-transfected with FLAG-STIM1, GFP-Orai1, and DYRK2 for 36 hr. STIM1 was immunoprecipitated with FLAG agarose, and the associated proteins were analysed using the indicated antibodies. (**E**) Phosphosites mutations in STIM1 disrupt the interaction with Orai1. (**F**) C17 inhibits the interaction between FLAG-STIM1 and GFP-Orai1 without exogenously expressing DYRK2. (**G–I**) Effects of DYRK2 on the FRET signals between STIM1-YFP and CFP-Orai1. Upper panel, typical traces; lower panel, statistics. (**G**) HEK293 cells stably expressing STIM1-YFP and CFP-Orai1. (n = 3, ****, p < 0.0001. unpaired Student's t-test). (**H**) HEK293 STIM1-STIM2 DKO cells stably expressing Orai1-CFP cells transiently expressing STIM1-1-491-YFP (n = 3, unpaired Student's t-test). (**I**) HEK STIM1-STIM2 DKO cells transiently expressing STIM1-YFP (red) or STIM1-10M (blue). (n = 3, ****, p < 0.0001, unpaired Student's t-test). (**J**) C17 inhibited SOCE in HEK293A cells. HEK293A cells were transfected with GCAMP6f or GCAMP6f plus STIM1 for 24 hr and then treated with 1 µM C17 for 1 hr. Before thapsigargin treatment, the cell culture medium was switched to a Ca²⁺-free medium containing thapsigargin (1 µM, solid lines) or DMSO (dashed lines) was added to the cells, and 2 mM Ca²⁺ was added 12 min later. The red and green lines correspond to C17-treated cells. Blue and black lines represent untreated cells. GCAMP6f fluorescence was monitored by a Zeiss LSM 700 laser scanning confocal microscope. (**K**) Quantification of (**J**). The following number of cells were monitored: STIM1, 45 cells on 3 coverslips (blue solid line); STIM1 +C17 (1 µM), 48 cells on 3 coverslips (red solid line); endogenous, 47 cells on 3 coverslips (black solid line); endogenous +C17 (1 µM), 42 cells on 3 coverslips (green solid line). STIM1(-Tg), 43 cells on 3 coverslips (blue dashed line). STIM1 +C17 (1 µM) (-Tg), 43 cells on 3 coverslips (red dashed line); endogenous (-Tg), 43 cells on 3 coverslips (black dashed line); and endogenous +C17 (1 µM) (-Tg), 43 cells on 3 coverslips (green dashed line). Error bars represent the means ± SEM. (**L**) A hypothetic model depicts DYRK2-mediated STIM1 activation.

The online version of this article includes the following source data and figure supplement(s) for figure 5:

**Source data 1.** Raw data of Coomassie Blue Staining for *Figure 5A*.

**Source data 2.** Raw data of western blot and Coomassie Blue Staining for *Figure 5B*.

**Source data 3.** Raw data of western blot for *Figure 5D,E*.

**Source data 4.** Raw data of Western blot for *Figure 5F*.

**Source data 5.** Raw data of FRET responses between STIM1-YFP and CFP-Orai1 for Figure 5G, FRET responses between STIM1-1-491-YFP and CFP-Orai1 for Figure 5H, FRET responses between STIM1-YFP, STIM1-10M-YFP and CFP-Orai1 for *Figure 5I*.

**Source data 6.** Raw data of Store-operated Ca2+ entry (SOCE) analyses for *Figure 5J*.

**Figure supplement 1.** Quantitative analysis of phosphorylation sites on STIM1.

**Figure supplement 1—source data 1.** Raw data of quantitative analysis of phosphorylation sites on STIM1 by DYRK2 upon C17 treatment for *Figure 5—figure supplement 1*.

of DYRK2 bound to C17 revealed critical interactions that explain its high selectivity, including a hydrogen bond between the (S)–3-methylpyrrolidine ring and Glu352 in DYRK2.

C17 provided us with a unique tool to interrogate the physiological functions of DYRK2. We treated U266 cells with C17 and performed quantitative phosphoproteomic analyses. We found that the cellular phosphorylation pattern is significantly altered by C17, suggesting that DYRK2 likely has multiple cellular targets and is involved in a network of biological processes. We then identified several leading phosphosites that are downregulated and demonstrated that 4E-BP1 and STIM1 are bona fide substrates of DYRK2. We showed that DYRK2 efficiently phosphorylated 4E-BP1 at multiple sites, including Thr37, and combined treatment of C17 with AKT and MEK inhibitors resulted in marked suppression of 4E-BP1 phosphorylation. Therefore, DYRK2 likely functions synergistically with other kinases to regulate protein synthesis.

For the first time, we also discovered that DYRK2 could efficiently phosphorylate STIM1, and phosphorylation of STIM1 by DYRK2 substantially increased the interaction between STIM1 and ORAI1. Treating cells with C17 suppressed SOCE, validating the critical role of DYRK2 in regulating calcium entry into cells. These data allow us to present a hypothetical model showing how DYRK2 triggers the activation of STIM1 (*Figure 5L*). Under resting conditions, the cytosolic portion of STIM1 likely adopts an inactive conformation. DYRK2 can phosphorylate STIM1 and induce its oligomerization, which then interacts with the Orai1 channel and leads to its opening. One inadequacy of our study is the lack of further insight into the regulation mechanism of this process. In particular, what is the upstream signal that triggers DYRK2 activation? Nevertheless, our data offer a valuable model that allows further investigation of the relationship between DYRK2 and SOCE.

Recently, Mehnert et al. developed a multilayered proteomic workflow and determined how different pathological-related DYRK2 mutations altered protein conformation, substrates modification, and biological function (*Qin et al., 2016*). DYRK2 is implicated in regulating multiple cellular

processes, and the selective DYRK2 inhibitor we developed here will serve as a valuable tool in dissecting its complex downstream pathways.

## Materials and methods

### Antibodies and reagents

Antibodies used in this study were: anti-4E-PB1 (Cell Signaling Technology, RRID: AB_2097841), anti-phosphorylated 4E-BP1 (Thr37/46) (Cell Signaling Technology, RRID: AB_560835), anti-phosphorylated 4E-BP1 (Ser65) (Cell Signaling Technology, RRID: AB_330947), anti-phosphorylated 4E-BP1 (Thr70) (Cell Signaling Technology, RRID: AB_2798206), anti-HA (Cell Signaling Technology, RRID: AB_1549585), anti-Flag (Sigma, RRID: AB_259529), anti-Flag (Abcam, #ab205606), anti-GFP (Proteintech, RRID: AB_11182611), Anti-GFP (Abcam, #ab183734), anti-RPT3 (Thermo Fisher Scientific, RRID: AB_2781512), anti-GAPDH (TransGen Biotech, #HC301-01). Secondary antibodies were horseradish peroxidase (HRP)-conjugated anti-rabbit IgG (H + L) or HRP-conjugated anti-mouse IgG (H + L) purchased from Transgene Biotechnology (#HC101-01, #HC201-01). Rabbit anti-pThr25 polyclonal antibody was generated using the following phospho-peptide as immunogen: LSVSRPQ(pT) GLSFLGP as reported previously (*Guo et al., 2016*). Reagents used in this study were: AKTi-1/2 (Selleck, #S80837), PD0325901 (Aladdin, #P125494), Thapsigargin (Aladdin, #T135258). Inhibitors were dissolved in dimethyl sulfoxide. All chemical reagents were used as supplied by Sigma-Aldrich, J&K Scientific, Alfa Aesar Chemicals, Energy Chemicals and Bide Pharmatech. DCM, DMF, DMSO were distilled from calcium hydride; tetrahydrofuran was distilled from sodium/benzophenone ketyl prior to use.

### Cloning

The GCaMP6f, pEGFP-Orai1, and mCherry-STIM1 plasmids were kindly gifted from the Xiaowei Chen Lab (Peking University, China). The GFP-tagged human DYRK1A, 1B, 2, 3, 4, pLL3.7-DYRK2-shRNA, psPAX2, and pMD2.G plasmids were kindly gifted from the Xing Guo Lab (Zhejiang University, China). DYRK2$^{208-552}$ was subcloned into the pQlinkHx vector (Clontech) with an engineered N-terminal His tag. STIM1$^{235-END}$ and full-length 4EBP1 were subcloned into the pQlinkGx vector (Clontech) with an engineered N-terminal GST tag. Full-length STIM1 was subcloned into a pCCF vector (Clontech) with an engineered N-terminal FLAG tag. The HA-mcherry-DYRK2 and HA-mcherry-DYRK2-D275N plasmids were generated by modification of pEGFP-DYRK2 and pEGFP-DYRK2-D275N plasmids. HA-mcherry was PCR amplified from pmCherry-N1 plasmid and replaced EGFP by homologous recombination. All plasmids were verified by DNA sequencing.

### Cell culture, transfection, and infection

Mammalian cells were all grown in a humidified incubator with 5% $CO_2$ at 37 °C. HEK293T (RRID:CVCL_0063), HEK293A (Thermo Fisher, R70507), and HEK293 (RRID:CVCL_0045) cells were grown in Dulbecco's Modified Eagle Media (DMEM, Gibco) supplemented with 10% FBS, 4 mM L-glutamine, 100 U/mL penicillin, and 100 mg/mL streptomycin (Gibco). U266 (RRID:CVCL_0566) cells were grown in RPMI 1640 (Gibco) supplemented with 10% FBS, 4 mM L-glutamine, 100 U/mL penicillin, and 100 mg/mL streptomycin (Gibco). HCT116 cells (China Infrastructure of Cell Line Resources, 1101HUM-PUMC000158) were grown in Iscove's Modified Dulbecco's Medium (IMDM, Gibco) supplemented with 10% FBS, 4 mM L-glutamine, 100 U/mL penicillin, and 100 mg/mL streptomycin (Gibco). All cell lines were confirmed by STR (short tandem repeat) profiling and tested negative for mycoplasma contamination. All cell lines are not in the list of commonly misidentified cell lines maintained by the International Cell Line Authentication Committee (version 11). Transient transfection of HEK293T, HEK293A cells were carried out using Lipofectamine 2000 (Thermo Fisher Scientific) or X-tremeGENE 9 DNA Transfection reagent (Roche) as recommended by the manufacturer, and transfected cells were used in experiments 24–48 hr later. In Lipofectamine transfection, the cells were cultured to ~70–80% confluency in 10 cm dishes, followed by transfection with 10–12 µg plasmid. The cells were changed with fresh DMEM after 12 hr and incubated for 36 hr before further experiments. In X-tremeGENE 9 DNA transfection, the cells were cultured to ~50 confluency in 35 mm glass bottom dishes coated with poly-D-lysine, followed by transfection with 1–3 µg plasmid. The cells were changed with fresh DMEM after 6 hr and incubated for 24–36 hr before further experiments. Lentiviruses were

produced using the psPAX2 and pMD2.G packaging vectors. Viral media were passed through a pre-wetted 0.45 μm filter and mixed with 10 μg mL$^{-1}$ Polybrene (Sigma) before being added to recipient cells. Infected cells were selected with puromycin (1–2 μg mL$^{-1}$, Life Technologies) to generate stable populations.

## DYRK2 protein purification and co-crystallization

DYRK2$^{208-552}$ with an N-terminal 6 × His affinity tag and TEV protease cleavage site which expressed in *E. coli* BL21 (DE3). Bacterial cultures were grown at 37 °C in LB medium to an OD600 of 0.6–0.8 before induced with 0.5 mM IPTG overnight at 18 °C. Cells were collected by centrifugation and frozen at –80 °C. For protein purification, the cells were suspended in the lysis buffer (50 mM HEPES, pH 7.5, 500 mM NaCl, 20 mM imidazole, 5% glycerol, 5 mM β-mercaptoethanol, and 1 mM phenyl-methanesulfonylfluoride) and disrupted by sonication. The insoluble debris was removed by centrifugation. The supernatant was applied to a Ni-NTA column (GE Healthcare). The column was washed extensively with the wash buffer (50 mM HEPES, pH 7.5, 500 mM NaCl, 50 mM imidazole, 5% glycerol, and 5 mM β-mercaptoethanol) and bound DYRK2 protein was eluted using the elution buffer (50 mM HEPES, pH 7.5, 500 mM NaCl, 500 mM imidazole, 5% glycerol, and 5 mM β -mercaptoethanol). After cleavage with TEV protease, the protein sample was passed through a second Ni-NTA column to separate untagged DYRK2 from the uncut protein and the protease. Final purification was performed using a Superdex 200 gel filtration column (GE Healthcare), and the protein was eluted using the final buffer (25 mM HEPES, pH 7.5, 400 mM NaCl, 1 mM DTT, and 5% glycerol). Purify the DYRK2-D275N using the same method as shown above. Purified DYRK2 and DYRK2-D275N were concentrated to 10 mg mL$^{-1}$ and flash-frozen with liquid nitrogen.

DYRK2$^{208-552}$ was incubated with 200 μM compounds on ice before crystallization. The protein-compounds mixture was then mixed in a 1:1 ratio with the crystallization solution (0.36 M-0.5 M sodium citrate tribasic dihydrate, 0.01 M sodium borate, pH 7.5–9.5) in a final drop size of 2 μl. The DYRK2-compounds crystals were grown at 18 °C by the sitting-drop vapor diffusion method. Cuboid-shaped crystals appeared after 4–7 days. Crystals were cryoprotected in the crystallization solution supplemented with 35% glycerol before frozen in liquid nitrogen.

The X-ray diffraction data were collected at Shanghai Synchrotron Radiation Facility (SSRF) beamline BL17U. The diffraction data were indexed, integrated, and scaled using HKL-2000 (HKL Research). The structure was determined by molecular replacement using the published DYRK2 structure (PDB ID: 3K2L) (*Soundararajan et al., 2013*) as the search model using the Phaser program (*McCoy et al., 2007*). Chembiodraw (version 13.0) was used to generated the.cif files for compounds, and then compounds were fitted using the LigandFit program in Phenix (*Adams et al., 2010*). The structural model was further adjusted in Coot (*Emsley et al., 2010*) and refined using Phenix. The quality of the structural model was checked using the MolProbity program in Phenix. The crystallographic data and refinement statistics are summarized in *Figure 1—source data 1*.

## IC$_{50}$ determination

IC$_{50}$ determination was carried out using the ADP-Glo kinase assay system (Promega, Madison, WI). Active DYRK1A, DYRK1B, DYRK2, DYRK3, Haspin, and MARK3 were purified as reported previously. C17 IC$_{50}$ measurements were carried out against the kinases with final concentrations between 0.01 nM to 100 μM *in vitro* (C17 was added to the kinase reaction prior to ATP master mix). The values were expressed as a percentage of the DMSO control. DYRK isoforms (1 ng/μL diluted in 50 mM Tris-HCl pH7.5, 2 mM DTT) were assayed against Woodtide (KKISGRLSPIMTEQ) in a final volume of 5 μL containing 50 mM Tris pH 7.5, 150 μM substrate peptide, 5 mM MgCl$_2$ and 10–50 μM ATP (10 μM for DYRK2 and DYRK3, 25 μM for DYRK1A and 50 μM for DYRK1B) and incubated for 60 min at room temperature. Haspin (0.2 ng/μL diluted in 50 mM Tris-HCl pH7.5, 2 mM DTT) was assayed against a substrate peptide H3(1–21) (ARTKQTARKSTGGKAPRKQLA) in a final volume of 5 μL containing 50 mM Tris pH 7.5, 200 μM substrate peptide, 5 mM MgCl$_2$ and 200 μM ATP and incubated for 120 min at room temperature. MARK3 (1 ng/μL diluted in 50 mM Tris-HCl pH7.5, 2 mM DTT) was assayed against Cdc25C peptide (KKKVSRSGLYRSPSMPENLNRPR) in a final volume of 5 μL 50 mM Tris pH 7.5, 200 μM substrate peptide, 10 mM MgCl$_2$ and 5 μM ATP and incubated for 120 min at room temperature. After incubation, the ADP-Glo kinase assay system was used to determine kinase activity following the manufacturer's protocol. IC$_{50}$ curves

were developed as % of DMSO control and IC$_{50}$ values were calculated using GraphPad Prism 8.4.0 software. Results are means ± SD for triplicate reactions with similar results obtained in at least one other experiment.

## KINOMEscan kinase profiling

The KINOMEscan kinase profiling assay was carried out at The Largest Kinase Assay Panel in the world for Protein Kinase Profiling (https://www.discoverx.com). C17 kinase selectivity was determined against a panel of 468 protein kinases. Results are presented as a percentage of kinase activity in DMSO control reactions. Protein kinases were assayed *in vitro* with 500 nM final concentration of C17 and the results are presented as an average of triplicate reactions ± SD or in the form of comparative histograms.

## Quantitative phosphoproteomic analysis

Triplicate U266 cells treated with/without C17 were lysed by the lysis buffer containing 1% (v/v) Triton X-100, 7 M Urea, 50 mM Tris-HCl, pH 8.5, 1 mM pervanadate, protease inhibitor mixture (Roche), and phosphatase inhibitor mixtures (Roche). The cell lysates were firstly digested with trypsin (Promega, USA) by the in-solution digestion method (*Chen et al., 2018*). After desalting, the Ti$^{4+}$-IMAC tip was used to purify the phosphopeptides. The phosphopeptides were desalted by the C18 StageTip prior to the LC MS/MS analysis (*Chen et al., 2018*). An Easy-nLC 1200 system coupled with the Q-Exactive HF-X mass spectrometer (Thermo Fisher Scientific, USA) was used to analyze the phosphopeptide samples with 1 hr LC gradient. The raw files were searched against Human fasta database (71,772 protein entries, downloaded from Uniprot on March 27, 2018) by MaxQuant (version 1.5.5.1). The oxidation (M), deamidation (NQ), and Phospho (STY) were selected as the variable modifications for the phosphopeptide identification, while the carbamidomethyl was set as the fixed modification. The false discovery rate (FDR) was set to 0.01 on PTM site, peptide, and protein level. Label-free quantification (LFQ) and match between runs were set for the triplicate analysis data. The MaxQuant searching file 'Phospho (STY)Sites.txt' was loaded into the Perseus software (version 1.5.5.3) to make volcano plots using student's t-test and cutoff of 'FDR < 0.05 and S0 = 2'. The pathway analysis was performed using the Kyoto Encyclopedia of Genes and Genomes (KEGG) database with cutoff of adjusted p-value < 0.05.

## Quantitative RT-PCR

Total RNA from cells was extracted using the RNeasy Mini Kit (Qiagen) and reverse-transcribed with the PrimeScript Real Time reagent Kit (with genomic DNA Eraser, TAKARA). The product of reverse transcription was diluted five times then subjected to quantitative rtPCR reaction in Applied Biosystems ViATM7 Real-Time PCR System (Applied Biosystems). The 20 μl quantitative rtPCR reaction contained 2 μl of the reverse-transcription reaction mixture, 2 × Hieff quantitative rtPCR SYBR Green Master Mix (Yeasen), 0.2 μM quantitative rtPCR forward primer, 0.2 μM quantitative rtPCR reverse primer (*Figure 4—figure supplement 2*) and ddH$_2$O. The quantitative rtPCR reaction condition was as follows: 95 °C, 5 min; (95 °C. 10 s; 60 °C, 30 s) × 40 cycles; 95 °C, 15 s; 60 °C, 1 min; 95 °C. 15 s (collect fluorescence at a ramping rate of 0.05 °C s-1); 4 °C, hold. Data analysis was performed by QuantStudioTM Real-Time PCR Software v.1.3.

## STIM1 and 4EBP1 protein purification

The cytosolic domain of STIM1 (bases 235-END) with an N-terminal GST-tag and TEV protease cleavage site which expressed in *E. coli* BL21 (DE3). Bacterial cultures were grown at 37 °C in LB medium to an OD600 of 0.6–0.8 before induced with 0.5 mM IPTG overnight at 18 °C. Cells were collected by centrifugation and frozen at –80 °C. For protein purification, the cells were suspended in the lysis buffer 50 mM Tris-HCl (pH 7.5), 500 mM NaCl, 5 mM β-mercaptoethanol, and 1 mM phenylmethanesulfonylfluoride and disrupted by sonication. The insoluble debris was removed by centrifugation. The supernatant was applied to a glutathione-Sepharose column (GE Healthcare) and eluted in lysis buffer containing 20 mM glutathione. Purify the GST-4EBP1 using the same method as shown above. Purified STIM1 and 4EBP1 were flash-frozen with liquid nitrogen.

### *In vitro* kinase assays

DYRK2 kinase assays were performed in 50 mM HEPES, pH 7.5, 100 mM NaCl, 10 mM $MnCl_2$, 10 mM ATP using STIM1 or 4EBP1 as substrate. The kinase reactions were initiated by the addition of DYRK2 with indicated concentration. Assays (25 µl volume) were carried out at 30 °C for 30 min, and terminated by addition of SDS-PAGE buffer containing 20 mM EDTA and then boiled. The reaction mixtures were then separated by SDS-PAGE and visualized by Coomassie Blue staining or analyzed by immunoblot using primary antibodies as indicated throughout.

### Co-immunoprecipitation and western blotting

HEK293A cells were cultured and transfected as described above. After transfection, the cells were washed three times with $Ca^{2+}$-free buffer containing 10 mM HEPES, 10 mM D-glucose, 150 mM NaCl, 4 mM KCl, 3 mM $MgCl_2$ and 0.1 mM EGTA (pH 7.4). Treatment of DMEM containing 1 µM of C17 at 37 °C were used for DYRK2 inhibition. $Ca^{2+}$-store depletion was triggered by incubating cells with 2 µM thapsigargin for 20 min. The cells were then lysed with lysis buffer consisting of 50 mM Tris-HCl (pH 7.5), 1 mM EGTA, 1 mM EDTA, 1% (v/v) Nonidet P40 (substitute), 1 mM sodium orthovanadate, 50 mM sodium fluoride, 5 mM sodium pyrophosphate, 0.27 M sucrose, 2 mM dithiothreitol (DTT), 1 mM benzamidine, 0.1 mM PMSF (added before lysis), 1% (v/v) protease inhibitor cocktail (Roche) and 1% (v/v) Phosphatase Inhibitor Cocktail (Roche). Protein concentrations were determined with the BCA protein assay kit Pierce (Thermo-Pierce). For immunoprecipitations, lysates containing equal protein amounts were incubated with FLAG–beads 2 hr at 4 °C. FLAG–beads were washed three times with lysis buffer containing 0.15 M NaCl. Proteins were eluted from the FLAG–beads by addition of 300 µg FLAG peptides (Smart Lifesciences). Eluted proteins were reduced by addition of loading buffer with 4 mM DTT followed by heating at 95 °C for 10 min. For western blotting, samples were electrophoresed in 10% or 12% gels and transferred to PVDF membranes. All antibody dilutions and washes were carried out in Tris-buffered saline (TBS; 137 mM NaCl, 19 mM Tris HCl and 2.7 mM KCl, at pH 7.4) containing 0.1% Tween-20 (TBS-T). Membranes were blocked in 5% non-fat milk solution in TBS-T for 1 hr at room temperature, incubated with indicated primary antibodies overnight at 4 °C, and incubated with secondary antibodies (horseradish peroxidase-linked anti-mouse or anti-rabbit) for 1 hr at room temperature. Blots were developed with AMERSHAM ImageQuant 800 (GE Healthcare) and exposed to film.

### Quantitative analysis of phosphorylation sites on STIM1

Triplicate HEK293A cells co-transfected with GFP-ORAI1, FLAG-DRYK2 and FLAG-STIM1 for 36 hr was treated with 1 µM and 10 µM C17 respectively for 1 hr. After collected, cells were washed with the $Ca^{2+}$-free buffer to remove excess $Ca^{2+}$ and then lysed by the lysis buffer. For immunoprecipitations, lysates containing equal protein amounts were incubated with FLAG–beads for 1 hr at 4 °C, which were washed three times with the lysis buffer afterwards. The proteins were eluted from the FLAG–beads by addition of 500 µg FLAG peptides (Smart Lifesciences). Then the eluted proteins were digested with trypsin by the FASP digestion method (*Wiśniewski et al., 2009*). The peptides were analyzed on a Q Exactive Plus mass spectrometer (Thermo Fisher Scientific) with 1 hr LC gradient. The raw files were searched against Human fasta database (downloaded from Uniprot) by MaxQuant (version 1.6.3.4). The oxidation (M), deamidation (NQ), and Phospho (STY) were selected as the variable modifications for the phosphopeptide identification, while the carbamidomethyl was set as the fixed modification. The false discovery rate (FDR) was set to 0.01 on PTM site, peptide, and protein level. Label-free quantification (LFQ) and match between runs were set for the triplicate analysis data.

### Confocal microscopy

Confocal imaging were carried out with a ZEISS LSM880 imaging system equipped with 65 × oil objective (NA = 1.45, Zeiss), 488- and 543 nm laser, controlled by Zen 2.3 SP1 software. YFP (505 ± 35) and mCherry (640 ± 50) emission were collected with CaAsP PMT (Optical section, 1.1 µm). Image analysis was performed using Image J Fiji (NIH) (*Zheng et al., 2018*). Each repeat contains data from at least 6 cells.

## Fluorescence imaging

Fluorescence signals were recorded using a ZEISS obersever-7 microscope equipped with an X-Cite 120-Q (Lumen dynamics), brightline filter sets (Semrock Inc), a 40 × oil objective (NA = 1.30), and a Prime 95B Scientific CMOS (sCMOS) camera (Teledyne Imaging). This system was controlled by Slide book6 software (3i). For fluorescence resonance energy transfer (FRET) measurements, three-channel-corrected FRET include cyan fluorescent protein (CFP), yellow fluorescent protein(YFP) and FRET raw were collected with corresponding filters, $F_{CFP}$ ($438 \pm 12E_x$/ $483 \pm 16_{Em}$), $F_{YFP}$ ($510 \pm 10E_x$/$542 \pm 13.5_{Em}$) and $FRET_{raw}$ ($438 \pm 12E_x$/$542 \pm 13.5_{Em}$), every 10 s. Calibration of bleed through from FRET donor or acceptor to FRET channel (0.20, and 0.36, correspondingly), as well as the system-dependent factor, G (2.473) were done as described earlier (*Ma et al., 2015*). These parameters were then used to generate calculate FRET efficiency (Eapp) values from raw fluorescent signals, similar to those previously described (*Ma et al., 2017*). At least 16 cells were collected for each repeat. Corresponding results were calculated with Matlab 2014a software and plotted with GraphPad Prism 8.4.0 software. Representative traces of at least three independent experiments are shown as mean ± SEM.

## Confocal imaging and intracellular Ca²⁺ measurement

Intracellular $Ca^{2+}$ measurement was performed on a Zeiss LSM 700 laser scanning confocal microscope equipped with a 63 × oil immersion objective lens (N.A. = 1.4) controlled by ZEN software. GCaMP6f fluorescence was excited using a 488 nm line of solid-state laser and fluorescence emission was collected with a 490- to 555 nm band-pass filter; mCherry fluorescence was excited using a 555 nm line of solid-state laser and fluorescence emission was collected with a 580 nm long-pass filter. Two high-sensitivity PMTs were used for detection. Cells were imaged at 10 s intervals for up to 20 mins. All live cell imaging experiments were performed at room temperature. Data were processed and analyzed using Zen and ImageJ software.

For intracellular $Ca^{2+}$ measurement, HEK293A cells were plated on glass-bottom 35 mm dishes and transfected as described above. Cells were washed with $Ca^{2+}$ free buffer 3 times 24 hr after transfection. For DYRK2 inhibition, cells were treated with DMEM containing 1 μM of C17 at 37 for 1 hr before $Ca^{2+}$ free buffer rinse. Depletion of $Ca^{2+}$-stores was triggered by incubating cells with 1 μM thapsigargin in $Ca^{2+}$-free buffer, and Store-operated $Ca^{2+}$ entry (SOCE) was induced by addition of 2 mM $CaCl_2$ to thapsigargin containing buffer. One μM C17 was added for DYRK2 inhibition assay. The intracellular free calcium concentration was measured by monitoring the fold change of GCaMP6f fluorescence, the data were shown as the mean ± SEM.

## Statistics and data presentation

Most experiments were repeated three times with multiple technical replicates to be eligible for the indicated statistical analyses, and representative image has been shown. All results are presented as mean ± SD unless otherwise mentioned. Data were analysed using Graphpad Prism 8.4.0 statistical package.

## Acknowledgements

We thank the National Center for Protein Sciences at Peking University for assistance with crystal screening, and Shanghai Synchrotron Radiation Facility and KEK Photon Factory for assistance with X-ray data collection. This work is supported by National Key Research Development Plan (2017YFA0505200 to XL and JX, 2019YFA0802104 to YW, and 2020YFE0202200 to RT), the National Natural Science Foundation of China (91853202, 21625201, 21961142010, 21661140001, and 21521003 to XL, 31822014 to JX, 31700088 to WC, 91954205 to YW, and 91953118 to RT), and the Beijing Outstanding Young Scientist Program (BJJWZYJH01201910001001 to XL).

## Additional information

### Funding

| Funder | Grant reference number | Author |
|---|---|---|
| National Key Research and Development Plan | 2017YFA0505200 | Tiantian Wei<br>Jue Wang<br>Ruqi Liang<br>Wendong Chen<br>Yilan Chen<br>Mingzhe Ma<br>An He<br>Yifei Du<br>Wenjing Zhou<br>Zhiying Zhang<br>Xin Zeng<br>Chu Wang<br>Jin Lu<br>Xing Guo<br>Xiao-Wei Chen<br>Youjun Wang<br>Junyu Xiao<br>Xiaoguang Lei |
| National Natural Science Foundation of China | 91853202 | Tiantian Wei<br>Jue Wang<br>Ruqi Liang<br>Wendong Chen<br>Yilan Chen<br>Mingzhe Ma<br>An He<br>Yifei Du<br>Wenjing Zhou<br>Zhiying Zhang<br>Xin Zeng<br>Chu Wang<br>Jin Lu<br>Xing Guo<br>Xiao-Wei Chen<br>Youjun Wang<br>Junyu Xiao<br>Xiaoguang Lei |
| National Natural Science Foundation of China | 21625201 | Tiantian Wei<br>Jue Wang<br>Ruqi Liang<br>Wendong Chen<br>Yilan Chen<br>Mingzhe Ma<br>An He<br>Yifei Du<br>Wenjing Zhou<br>Zhiying Zhang<br>Xin Zeng<br>Chu Wang<br>Jin Lu<br>Xing Guo<br>Xiao-Wei Chen<br>Youjun Wang<br>Junyu Xiao<br>Xiaoguang Lei |

| Funder | Grant reference number | Author |
|---|---|---|
| National Natural Science Foundation of China | 21961142010 | Tiantian Wei<br>Jue Wang<br>Ruqi Liang<br>Wendong Chen<br>Yilan Chen<br>Mingzhe Ma<br>An He<br>Yifei Du<br>Wenjing Zhou<br>Zhiying Zhang<br>Xin Zeng<br>Chu Wang<br>Jin Lu<br>Xing Guo<br>Xiao-Wei Chen<br>Youjun Wang<br>Junyu Xiao<br>Xiaoguang Lei |
| National Natural Science Foundation of China | 21661140001 | Tiantian Wei<br>Jue Wang<br>Ruqi Liang<br>Wendong Chen<br>Yilan Chen<br>Mingzhe Ma<br>An He<br>Yifei Du<br>Wenjing Zhou<br>Zhiying Zhang<br>Xin Zeng<br>Chu Wang<br>Jin Lu<br>Xing Guo<br>Xiao-Wei Chen<br>Youjun Wang<br>Junyu Xiao<br>Xiaoguang Lei |
| National Natural Science Foundation of China | 21521003 | Tiantian Wei<br>Jue Wang<br>Ruqi Liang<br>Wendong Chen<br>Yilan Chen<br>Mingzhe Ma<br>An He<br>Yifei Du<br>Wenjing Zhou<br>Zhiying Zhang<br>Xin Zeng<br>Chu Wang<br>Jin Lu<br>Xing Guo<br>Xiao-Wei Chen<br>Youjun Wang<br>Junyu Xiao<br>Xiaoguang Lei |
| National Key Research Development Plan | 2019YFA0802104 | Youjun Wang |
| National Key Research Development Plan | 2020YFE0202200 | Ruijun Tian |
| National Natural Science Foundation of China | 31822014 | Junyu Xiao |
| National Natural Science Foundation of China | 31700088 | Wendong Chen |
| National Natural Science Foundation of China | 91954205 | Youjun Wang |

| Funder | Grant reference number | Author |
|---|---|---|
| National Natural Science Foundation of China | 91953118 | Ruijun Tian |
| Beijing Outstanding Young Scientist Program | BJJWZYJH01201910001001 | Xiaoguang Lei |

The funders had no role in study design, data collection and interpretation, or the decision to submit the work for publication.

## Author contributions

Tiantian Wei, Ruqi Liang, Data curation, Formal analysis, Investigation, Validation, Writing – original draft, Writing – review and editing; Jue Wang, Data curation, Formal analysis, Investigation, Writing – original draft, Writing – review and editing; Wendong Chen, Data curation, Formal analysis, Investigation, Resources, Validation, Writing – original draft, Writing – review and editing; Yilan Chen, Data curation, Formal analysis, Investigation, Validation; Mingzhe Ma, Investigation, Validation; An He, Yifei Du, Wenjing Zhou, Zhiying Zhang, Xin Zeng, Investigation; Chu Wang, Jin Lu, Xiao-Wei Chen, Youjun Wang, Supervision; Xing Guo, Investigation, Supervision; Ruijun Tian, Project administration, Supervision, Validation, Writing – review and editing; Junyu Xiao, Conceptualization, Formal analysis, Funding acquisition, Project administration, Supervision, Writing – original draft, Writing – review and editing; Xiaoguang Lei, Conceptualization, Data curation, Formal analysis, Funding acquisition, Project administration, Resources, Supervision, Validation, Writing – original draft, Writing – review and editing

## Author ORCIDs

Tiantian Wei http://orcid.org/0000-0001-8964-8839
Xiao-Wei Chen http://orcid.org/0000-0003-4564-5120
Youjun Wang http://orcid.org/0000-0003-0961-1716
Junyu Xiao http://orcid.org/0000-0003-1822-1701
Xiaoguang Lei http://orcid.org/0000-0002-0380-8035

## Decision letter and Author response

Decision letter https://doi.org/10.7554/eLife.77696.sa1
Author response https://doi.org/10.7554/eLife.77696.sa2

# Additional files

## Supplementary files
• Transparent reporting form
• Supplementary file 1. NMR figures.

## Data availability

The structural coordinates of DYRK2 in complex with compounds 5, 6, 7, 8, 10, 13, 14, 17, 18, 19, and 20 have been deposited in the Protein Data Bank with accession codes 7DH3, 7DG4, 7DH9, 7DHV, 7DHC, 7DHK, 7DHO, 7DJO, 7DL6, 7DHH, and 7DHN, respectively. All the raw mass spectrometry data as well as the identified and significantly regulated phosphosites tables have been deposited in the public proteomics repository MassIVE and are accessible at https://massive.ucsd.edu/ProteoSAFe/dataset.jsp?task=3d564e40a31b4443b615574675591cc4.

The following dataset was generated:

| Author(s) | Year | Dataset title | Dataset URL | Database and Identifier |
|---|---|---|---|---|
| Lei X, Tian R | 2022 | Selective inhibition reveals the regulatory function of DYRK2 in protein synthesis and calcium entry | https://massive.ucsd.edu/ProteoSAFe/dataset.jsp?task=3d564e40a31b4443b615574675591cc4 | MassIVE, MSV000087106 |

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

## Appendix 1

### General information for chemical synthesis

NMR spectra were recorded on a Varian 400 MHz spectrometer, Bruker 400 MHz NMR spectrometer (ARX400), Bruker 400 MHz NMR spectrometer (AVANCE III), Bruker-500M Hz NMR spectrometer (500 M) and Bruker-600M Hz NMR spectrometer (600 M) at ambient temperature with CDCl$_3$ as the solvent unless otherwise stated. Chemical shifts are reported in parts per million relative to CDCl$_3$ (1 H, δ 7.26; 13 C, δ 77.16) and MeOD (1 H, δ 3.31; 13 C, δ 49.00). Data for 1 H NMR are reported as follows: chemical shift, integration, multiplicity (s = singlet, d = doublet, t = triplet, q = quartet, quint = quintet, sixt = sixtet, m = multiplet) and coupling constants. High-resolution mass spectra were obtained at Peking University Mass Spectrometry Laboratory using a Bruker APEX Flash chromatography. The samples were analyzed by UPLC/MS on a Waters Auto Purification LC/MS system (Waters C18 5 μm 150 × 4.6 mm SunFire separation column) or prepared by HPLC/MS on a Waters Auto Purification LC/MS system (ACQUITY UPLC BEH C18 17 μm 2.1 × 50 mm column). Analytical thin layer chromatography was performed using 0.25 mm silica gel 60 F plates, using 250 nm UV light as the visualizing agent and a solution of phosphomolybdic acid and heat as developing agents. Flash chromatography was performed using 200–400 mesh silica gel. Yields refer to chromatographically pure materials, unless otherwise stated. All reactions were carried out in oven-dried glassware under an argon atmosphere unless otherwise noted.

Synthetic Procedures:

**Appendix 1—scheme 1.** Synthesis route of C1-C4.

**5-methoxy-2-((4-methoxyphenyl)amino) benzoic acid (1 a).** 2-bromo-5-methoxybenzoic acid (9.26 g, 40 mmol), 4-methoxyaniline (6.90 g, 56 mmol), copper (0.73 g, 11 mmol), cuprous oxide (0.82 g, 5.7 mmol) and potassium carbonate (7.74 g, 56 mmol) were added to 100 mL DMF, the mixture was stirred at 80 °C overnight. The resulting slurry was cooled to room temperature, and 2 M HCl was added into the mixture until the system became acidic and a large amount of solid was precipitated. After filtration, the precipitate was washed with water and dried to give compound **1** a (4.02 g, 37%) as a dark green solid. **1** a was used in next step without further purification.

**9-chloro-2,7-dimethoxyacridine (1b).** Compound **1** a (2.58 g, 9.45 mmol) was added in a sealed tube, and 30 mL of phosphorus oxychloride was added under argon atmosphere. The reaction was heated at 130 °C for 8 h. The resulting slurry was poured onto ice with vigorous stirring, and a large amount of a yellow solid was precipitated. After filtration, the precipitate was washed with water and dried to give compound **1b** (2.58 g, quant.) as an orange solid. **1b** was used in next step without further purification.

**_tert_-butyl (3-((2,7-dimethoxyacridin-9-yl)thio)propyl)carbamate (1).** To a solution of **1b** (50 mg, 0.183 mmol) in 5 mL of anhydrous DMF, sodium hydrosulfide hydrate powder (67% 22.9 mg, 0.274 mmol) was added under argon atmosphere, and the reaction was stirred at 50 °C for 2 h until full conversion of **1b**. _N_-Boc-3-aminopropyl bromide (60.9 mg, 0.274 mmol) and potassium carbonate (50.5 mg, 0.365 mmol) were added into the slurry and the reaction was allowed to react at room temperature overnight. The solvent was then evaporated and the residue was dissolved with dichloromethane and washed with water. The combined organic extracts were dried over anhydrous

$Na_2SO_4$ and concentrated *in vacuo*. The residue was purified by chromatography on a silica gel column to give compound **1** as a light yellow solid (47.4 mg, 61%). $^1$H NMR (400 MHz, $CDCl_3$) δ 8.10 (d, *J* = 9.3 Hz, 2 H), 7.94 (s, 2 H), 7.42 (d, *J* = 9.3 Hz, 2 H), 4.04 (s, 6 H), 3.20 (d, *J* = 5.9 Hz, 2 H), 2.95 (t, *J* = 7.2 Hz, 2 H), 1.70–1.61 (m, 2 H), 1.39 (s, 9 H). $^{13}$C NMR (151 MHz, $CDCl_3$) δ 158.5, 156.0, 144.3, 136.0, 132.1, 130.7, 124.1, 102.2, 79.5, 55.8, 39.6, 33.6, 30.8, 28.5. HRMS(ESI) [M + H]$^+$ calculated for $C_{23}H_{29}N_2O_4S$: 429.1843, found: 429.1831.

### Compounds 2–4
By employment of the above-described procedure, starting from **1b** and using suitable bromides, compounds 2–4 were prepared.

### *N*-(3-((2,7-dimethoxyacridin-9-yl)thio)propyl)acetamide (2). Yield 58%
$^1$H NMR (400 MHz, $CDCl_3$) δ 8.10 (d, *J* = 9.4 Hz, 2 H), 7.94 (d, *J* = 2.7 Hz, 2 H), 7.42 (dd, *J* = 9.4, 2.8 Hz, 2 H), 4.04 (s, 6 H), 3.30 (dd, *J* = 13.2, 6.7 Hz, 2 H), 2.96 (t, *J* = 7.3 Hz, 2 H), 1.88 (s, 3 H), 1.69–1.59 (m, 2 H). $^{13}$C NMR (151 MHz, $CDCl_3$) δ 170.2, 158.6, 144.3, 135.8, 132.2, 130.7, 124.1, 102.2, 55.8, 38.7, 33.6, 30.4, 23.4. HRMS(ESI) [M + H]$^+$ calculated for $C_{20}H_{23}N_2O_3S$: 371.1424, found: 371.1428.

### *N*-(4-((2,7-dimethoxyacridin-9-yl)thio)butyl)acetamide (3). Yield 57%
$^1$H NMR (400 MHz, $CDCl_3$) δ 8.10 (d, *J* = 9.3 Hz, 2 H), 7.95 (d, *J* = 2.7 Hz, 2 H), 7.42 (dd, *J* = 9.3, 2.7 Hz, 2 H), 4.04 (s, 6 H), 3.15 (dd, *J* = 13.1, 6.7 Hz, 2 H), 2.94 (t, *J* = 7.1 Hz, 2 H), 1.86 (s, 3 H), 1.59 (m, 2 H), 1.49 (m, 2 H). $^{13}$C NMR (101 MHz, $CDCl_3$) δ 170.1, 158.4, 144.2, 136.2, 132.0, 130.7, 124.0, 102.2, 55.8, 39.0, 35.9, 28.9, 27.5, 23.3. HRMS(ESI) [M + H]$^+$ calculated for $C_{21}H_{25}N_2O_3S$: 385.1580, found: 385.1572.

### 4-((2,7-dimethoxyacridin-9-yl)thio)butanenitrile (4). Yield 52%
$^1$H NMR (400 MHz, $CDCl_3$) δ 8.11 (d, *J* = 9.4 Hz, 2 H), 7.89 (d, *J* = 2.7 Hz, 2 H), 7.43 (dd, *J* = 9.4, 2.8 Hz, 2 H), 4.04 (s, 6 H), 3.07 (t, *J* = 7.0 Hz, 2 H), 2.47 (t, *J* = 7.0 Hz, 2 H), 1.82–1.72 (m, 2 H). $^{13}$C NMR (101 MHz, $CDCl_3$) δ 158.7, 144.3, 134.4, 132.3, 130.6, 124.2, 118.8, 101.7, 55.8, 34.4, 25.8, 16.4. HRMS(ESI) [M + H]$^+$ calculated for $C_{19}H_{19}N_2O_2S$: 339.1162, found: 339.1159.

**Appendix 1—scheme 2.** Synthesis route of C5-C6.

tert-butyl 4-((2,7-dimethoxyacridin-9-yl)thio)piperidine-1-carboxylate (5 a). To a solution of **1b** (50 mg, 0.183 mmol) in 5 mL of anhydrous DMF, sodium hydrosulfide hydrate powder (67% 22.9 mg, 0.274 mmol) was added under argon atmosphere, and the reaction was stirred at 50 °C for 2 h until full conversion of **1b**. 4-bromopiperidine-1-carboxylic acid tert-butyl ester (72.4 mg, 0.274 mmol) and potassium carbonate (50.5 mg, 0.365 mmol) were added into the slurry and the

reaction was allowed to react at room temperature overnight. The solvent was then evaporated and the residue was dissolved with dichloromethane and washed with water. The combined organic extracts were dried over anhydrous $Na_2SO_4$ and concentrated *in vacuo*. The residue was purified by chromatography on a silica gel column to give compound **5** a as a light yellow solid (45.4 mg, 55%). [1]H NMR: (400 MHz, $CDCl_3$) δ 8.08 (d, *J* = 9.3 Hz, 2 H), 7.93 (d, *J* = 2.7 Hz, 2 H), 7.40 (dd, *J* = 9.3, 2.8 Hz, 2 H), 4.01 (s, 6 H), 3.98–3.84 (m, 2 H), 3.25–3.17 (m, 1 H), 2.81 (ddd, *J* = 13.5, 10.5, 3.0 Hz, 2 H), 1.86–1.72 (m, 2 H), 1.72–1.58 (m, 2 H), 1.42 (s, 9 H). [13]C NMR (101 MHz, $CDCl_3$) δ 158.4, 154.7, 144.3, 134.8, 132.0, 131.1, 124.0, 102.4, 79.8, 55.7, 46.9, 43.2, 33.1, 28.5. HRMS(ESI) [M + H][+] calculated for $C_{25}H_{31}N_2O_4S$: 455.1999, found: 455.1995.

## Compounds 6 a
By employment of the above-described procedure, starting from **1b** and using suitable bromide, compound **6** a were prepared.

## tert-butyl 4-(((2,7-dimethoxyacridin-9-yl)thio)methyl)piperidine-1-carboxylate (6 a). Yield 79%
[1] H NMR (400 MHz, $CDCl_3$) δ 8.10 (d, *J* = 9.4 Hz, 2 H), 7.94 (d, *J* = 2.7 Hz, 2 H), 7.42 (dd, *J* = 9.4, 2.8 Hz, 2 H), 4.03 (d, *J* = 48.7 Hz, 6 H), 4.15–3.98 (overlapped, m, 2 H), 2.82 (d, *J* = 6.8 Hz, 2 H), 2.59 (t, *J* = 12.1 Hz, 2 H), 1.86 (br s, 2 H), 1.53–1.46 (m, 1 H), 1.43 (s, 9 H), 1.23–1.14 (m, *J* = 10.9 Hz, 2 H). [13]C NMR (101 MHz, $CDCl_3$) δ 158.4, 154.8, 144.2, 136.7, 132.0, 130.4, 123.9, 102.0, 79.5, 55.6, 43.6, 42.9, 36.8, 28.5. HRMS(ESI) [M + H][+] calculated for $C_{26}H_{36}N_2O_4S$: 469.2156, found: 469.2153.

**2,7-dimethoxy-9-(piperidin-4-ylthio)acridine (5)**. Compound **5** a (45.4 mg, 0.100 mmol) was dissolved in a 5% trifluoroacetic acid dichloromethane solution and the mixture was allowed to react at room temperature for 2 h. The solvent was then evaporated and the residue was dissolved with methanol, then purified by HPLC/MS on a Waters Auto Purification LC/MS system (ACQUITY UPLC BEH C18 17 µm 2.1 × 50 mm column) to afford **5** as a dark red solid (35.2 mg, 75%). [1]H NMR (400 MHz, MeOD) δ 8.05 (d, *J* = 9.4 Hz, 2 H), 8.00 (d, *J* = 2.7 Hz, 2 H), 7.49 (dd, *J* = 9.4, 2.8 Hz, 2 H), 4.05 (s, 6 H), 3.36–3.33 (m, 1 H), 2.99 (d, *J* = 6.8 Hz, 2 H), 2.85 (td, *J* = 12.8, 2.9 Hz, 2 H), 2.13 (d, *J* = 16.5 Hz, 2 H), 1.70–1.65 (m, 2 H). [13]C NMR (151 MHz, MeOD) δ 160.9, 144.0, 138.6, 132.9, 129.7, 126.7, 104.0, 56.0, 45.5, 44.4, 30.9. HRMS(ESI) [M + H][+] calculated for $C_{20}H_{23}N_2O_2S$: 355.1475, found: 355.1467.

**Compounds 6**
By employment of the above-described procedure, starting from **6** a, compounds **6** was prepared.

## 2,7-dimethoxy-9-((piperidin-4-ylmethyl)thio)acridine (6). Yield 56%
[1] H NMR (400 MHz, MeOD) δ 8.24 (d, *J* = 9.4 Hz, 2 H), 8.17 (d, *J* = 2.3 Hz, 2 H), 7.85 (dd, *J* = 9.4, 2.4 Hz, 2 H), 4.14 (s, 6 H), 3.75–3.68 (m, 1 H), 3.39 (d, *J* = 13.3 Hz, 2 H), 3.01 (t, *J* = 10.9 Hz, 2 H), 2.18–2.09 (m, 2 H), 2.05–2.01 (m, 2 H), 1.37–1.32 (m, 2 H). [13]C NMR (101 MHz, MeOD) δ 61.0, 151.0, 135.6, 131.8, 131.1, 124.0, 104.0, 56.8, 44.7, 44.2, 35.9, 29.1. HRMS(ESI) [M + H][+] calculated for $C_{21}H_{25}N_2O_2S$: 369.1631, found: 369.1638.

**Appendix 1—scheme 3.** General procedure of C7, C16-C19 synthesis.

**6,6'-azanediylbis-3-methoxybenzoic acid (1 a')**. 2-amino-5-methoxybenzoic acid (2.00 g, 8.66 mmol), 2-bromo-5-methoxybenzoic acid (0.11 g, 1.73 mmol), copper (0.11 g, 1.73 mmol), cuprous oxide (0.12 g, 0.87 mmol) and potassium carbonate (7.74 g, 56 mmol) were added to 15 mL DMF, the mixture was stirred at 80 °C overnight. The resulting slurry was cooled to room temperature, and 2 M HCl was added into the mixture until the system became acidic and a large amount of solid was precipitated. After filtration, the precipitate was washed with water and dried to give compound **1 a'** (2.00 g, 73%) as a green solid. **1 a'** was used in next step without further purification.

## 9-chloro-2,7-dimethoxyacridine-4-carboxylic acid (1b')

Compound **1 a'** (1.00 g, 3.16 mmol) was added in a sealed tube, and 10 mL of phosphorus oxychloride was added under argon atmosphere. The reaction was heated at 130 °C for 8 h. The resulting slurry was poured onto ice with vigorous stirring, and a large amount of a yellow solid was precipitated. After filtration, the precipitate was washed with water and dried to give compound **1b** (1.00 g, 95%) as an orange solid. **1b'** was used in next step without further purification. **methyl 9-chloro-2,7-dimethoxyacridine-4-carboxylate (1 c)**. To a suspension of **1b'** (0.89 g, 2.67 mmol) in 10 mL of dry dichloromethane, 0.40 mL of oxalyl chloride was added followed by one drop of DMF, and a large amount of bubbles was generated. The mixture was allowed to react at room temperature for 0.5 h until the system became a brownish black solution. Then the reaction was quenched by dry menthol at room temperature for 2 h, followed by the addition of triethylamine until the mixture became neutral. The system was diluted with dichloromethane, washed twice with brine, dried over anhydrous $Na_2SO_4$ and concentrated *in vacuo*. The residue was purified by chromatography on a silica gel column (dichloromethane / ethyl acetate = 95/5) to gave compound **1 c** as a yellow solid (0.74 g, 80%).[1] H NMR (400 MHz, CDCl$_3$) δ 8.12 (d, *J* = 9.3 Hz, 1 H), 7.70 (s, 1 H), 7.64 (s, 1 H), 7.46 (s, 1 H), 7.42 (d, *J* = 9.4 Hz, 1 H), 4.09 (s, 3 H), 4.03 (s, 6 H). $^{13}$C NMR (101 MHz, CDCl$_3$) δ 167.9, 159.0, 157.1, 144.8, 141.3, 135.9, 134.1, 132.6, 125.8, 125.7, 125.0, 124.9, 103.0, 99.6, 56.0, 55.8, 52.3. HRMS(ESI) [M + H]$^+$ calculated for $C_{17}H_{15}ClNO_4$: 332.0684, found: 332.0680.

**Appendix 1—scheme 4.** Synthesis route of C7.

**methyl-9-(((1-(tert-butoxycarbonyl)piperidin-4-yl)methyl)thio)–2,7-dimethoxyacridine-4-caboxylate (7 a).** To a solution of **1** c (0.38 g, 1.16 mmol) in 10 mL of anhydrous DMF, sodium hydrosulfide hydrate powder (70%, 0.10 g, 1.21 mmol) was added under argon atmosphere, and the reaction was stirred at 50 °C for 2 h until full conversion of **1** c. 1-Boc-4-bromomethylpiperidine (0.64 g, 2.31 mmol) and potassium carbonate (0.40 g, 2.89 mmol) were added into the slurry and the reaction was allowed to react at room temperature overnight. The solvent was then evaporated and the residue was dissolved with dichloromethane and washed with water. The combined organic extracts were dried over anhydrous $Na_2SO_4$ and concentrated in vacuo. The residue was purified by chromatography on a silica gel column (dichloromethane / ethyl acetate = 95/5) to give compound **7** a as a yellow solid (0.38 g, 63%). [1]H NMR (400 MHz, $CDCl_3$) δ 8.13 (d, $J$ = 9.4 Hz, 1 H), 8.08 (d, $J$ = 2.0 Hz, 1 H), 7.89 (s, 1 H), 7.68 (d, $J$ = 1.9 Hz, 1 H), 7.41 (dd, $J$ = 9.4, 1.9 Hz, 1 H), 4.09 (s, 3 H), 4.02 (s, 6 H), 2.78 (d, $J$ = 6.7 Hz, 2 H), 2.55 (t, $J$ = 12.2 Hz, 2 H), 1.82 (br s, 2 H), 1.45–1.35(m, 3 H), 1.43 (s, 9 H), 1.20–1.09 (m, 2 H).[13] C NMR (101 MHz, $CDCl_3$) δ 168.2, 158.9, 157.0, 154.8, 144.7, 141.2, 137.0, 134.6, 133.0, 130.7, 130.5, 124.4, 124.3, 105.11, 101.74, 79.6, 56.0, 55.7, 52.8, 43.8, 43.1 (br s), 36.8, 31.8, 28.5. HRMS(ESI) [M + H][+] calculated for $C_{28}H_{35}N_2O_6S$: 527.2210, found: 527.2212.

**Compounds 16** a-19a. By employment of the above-described procedure, starting from **1 c'** and using suitable bromides, compounds **16** a-19a were prepared. **methyl 9-((3-((tert-butoxycarbonyl)amino)propyl)thio)–2,7-dimethoxyacridine-4-carboxylate (16 a). Yield 76%.**[1] H NMR (400 MHz, $CDCl_3$) δ 8.12 (d, $J$ = 9.4 Hz, 1 H), 8.09 (d, $J$ = 2.8 Hz, 1 H), 7.89 (d, $J$ = 2.8 Hz, 1 H), 7.68 (dd, $J$ = 2.9, 0.8 Hz, 1 H), 7.41 (dd, $J$ = 9.3, 2.8 Hz, 1 H), 4.40 (br s, 1 H), 4.09 (s, 3 H), 4.04 (s, 3 H), 4.03 (s, 3 H), 3.17 (q, $J$ = 6.6 Hz, 2 H), 2.92 (t, $J$ = 7.3 Hz, 2 H), 1.61–1.59 (m, 2 H), 1.39 (s, 9 H). [13]C NMR (101 MHz, $CDCl_3$) δ 168.2, 158.8, 157.0, 155.9, 144.57, 141.08, 136.2, 134.5, 132.9, 130.8, 130.6, 124.3, 124.3, 105.1, 101.8, 55.9, 55.7, 52.8, 39.5, 33.7, 30.7, 28.4. HRMS(ESI) [M + H][+] calculated for $C_{25}H_{31}N_2O_6S$: 487.1889, found: 487.1897.

## methyl-(S)–9-(((1-(tert-butoxycarbonyl)pyrrolidin-3-yl)methyl)thio)–2,7-dimethoxyacridine-4-carboxylate (17 a). Yield 76%

[1] H NMR (400 MHz, $CDCl_3$) δ 8.13 (d, $J$ = 9.4 Hz, 1 H), 8.08 (d, $J$ = 2.4 Hz, 1 H), 7.88 (d, $J$ = 2.6 Hz, 1 H), 7.68 (d, $J$ = 2.7 Hz, 1 H), 7.42 (dd, $J$ = 9.4, 2.6 Hz, 1 H), 4.10 (s, 3 H), 4.05 (s, 3 H), 4.04 (s, 3 H), 3.61–3.14 (m, 2 H), 3.27–2.1 (m, 4 H), 2.10–2.01 (m, 1 H), 1.90–1.85 (m, 1 H), 1.68–1.59 (m, 1 H), 1.43 (s, 9 H). [13]C NMR (101 MHz, $CDCl_3$) δ 168.2, 159.0, 157.1, 154.5, 144.6, 141.1, 135.9, 134.7, 133.1, 130.7, 130.5, 124.4, 124.3, 104.9, 101.6, 79.4, 55.9 (d, $J$ = 96 Hz), 52.8, 51.0 (d, $J$ = 104 Hz), 45.2 (d, $J$ = 104 Hz), 39.6–39.3 (m), 38.4, 31.3, 30.7, 28.6. HRMS(ESI) [M + H][+] calculated for $C_{27}H_{33}N_2O_6S$: 513.2052, found: 513.2054.

### methyl-(R)–9-(((1-(tert-butoxycarbonyl)pyrrolidin-3-yl)methyl)thio)–2,7-dimethoxyacridine-4-carboxylate (18 a). Yield 84%

[1] H NMR (400 MHz, CDCl$_3$) δ 8.13 (d, $J$ = 9.4 Hz, 1 H), 8.07 (s, 1 H), 7.88 (s, 1 H), 7.68 (s, 1 H), 7.41 (d, $J$ = 8.8 Hz, 1 H), 4.09 (s, 3 H), 4.04 (s, 6 H), 3.62–3.31 (m, 2 H), 3.26–3.18 (m, 1 H), 3.10–2.79 (m, 3 H), 2.11–1.99 (m, 1 H), 1.97–1.86 (m, 1 H), 1.65–1.56 (m, 1 H), 1.43 (s, 9 H). [13]C NMR (126 MHz, CDCl$_3$) δ 168.2, 159.1, 157.2, 154.5, 144.7, 141.3, 135.9, 134.8, 133.2, 130.8, 130.6, 124.4, 124.3, 105.0, 101.7, 79.4, 55.9(d, $J$ = 105 Hz), 52.8, 51.1 (d, $J$ = 115 Hz), 45.2 (d, $J$ = 135 Hz), 39.7–39.4 (m), 38.5, 31.4, 30.7, 28.6. HRMS(ESI) [M + H]$^+$ calculated for C$_{27}$H$_{33}$N$_2$O$_6$S: 513.2052, found: 513.2054.

### methyl-9-(((1-(tert-butoxycarbonyl)azetidin-3-yl)methyl)thio)–2,7-dimethoxyacridine-4-carboxylate (19 a). Yield 75%

[1] H NMR (400 MHz, CDCl$_3$) δ 8.14 (d, $J$ = 9.4 Hz, 1 H), 8.03 (s, 1 H), 7.84 (s, 1 H), 7.68 (d, $J$ = 1.6 Hz, 1 H), 7.42 (d, $J$ = 9.4 Hz, 1 H), 4.10 (s, 3 H), 4.04 (s, 6 H), 3.85 (t, $J$ = 8.3 Hz, 2 H), 3.60 (s, 2 H), 3.10 (d, $J$ = 7.8 Hz, 2 H), 2.26–2.16 (m, 1 H), 1.40 (s, 9 H). [13]C NMR (126 MHz, CDCl$_3$) δ 168.2, 159.2, 157.3, 156.3, 144.7, 141.2, 134.9, 134.8, 133.2, 130.9, 130.7, 124.5, 124.3, 104.9, 101.6, 79.7, 56.0, 55.8, 54.0(br s), 52.8, 39.9, 29.0, 28.5. HRMS(ESI) [M + H]$^+$ calculated for C$_{26}$H$_{31}$N$_2$O$_6$S: 499.1897, found: 499.1895.

### *tert*-butyl-4-(((4-(hydroxymethyl)–2,7-dimethoxyacridin-9-yl)thio)methyl)piperidine-1-carboxylate (7b)

3.1 mL of 1.5 M DIBAL-H solution in toluene was added to a solution of **7** a (348 mg, 0.661 mmol) in 10 mL of dry dichloromethane at 0 °C, and the reaction was stirred at room temperature for 4 h. The reaction was quenched by adding saturated potassium hydrogen tartrate solution, diluted with dichloromethane, washed twice with brine, dried over anhydrous Na$_2$SO$_4$ and concentrated *in vacuo*. The residue was purified by chromatography on a silica gel column (dichloromethane / ethyl acetate = 90/10) to gave compound **7b** (180 mg, 55%) as yellow foam.[1]H NMR (400 MHz, CDCl$_3$) δ 8.03 (d, $J$ = 9.3 Hz, 1 H), 7.88 (s, 1 H), 7.81 (s, 1 H), 7.39 (d, $J$ = 9.3 Hz, 1 H), 7.27 (s, 1 H), 5.41 (s, 1 H), 5.21 (s, 2 H), 4.02 (s, 3 H), 4.00 (s, 3 H) 2.79 (d, $J$ = 6.7 Hz, 2 H), 2.57 (t, $J$ = 11.7 Hz, 2 H), 1.84 (br s, 2 H), 1.47–1.25 (m, 3 H), 1.43 (s, 9 H), 1.20–1.09 (m, 2 H). [13]C NMR (101 MHz, CDCl$_3$) δ 158.5, 157.9, 154.8, 143.3, 142.7, 140.4, 137.6, 131.9, 131.1, 130.5, 124.1, 122.1, 102.0, 101.4, 79.6, 65.0, 55.7, 55.7, 43.6, 43.1, 36.8, 31.8, 28.5. HRMS(ESI) [M + H]$^+$ calculated for C$_{27}$H$_{35}$N$_2$O$_5$S: 499.2258, found: 499.2261.

### Compounds 16b-19b

By employment of the above-described procedure, starting from **16** a-19a, compounds **16b-19b** were prepared.

### *tert*-butyl (3-((4-(hydroxymethyl)–2,7-dimethoxyacridin-9-yl)thio)propyl) carbamate (16b). Yield 68%

[1] H NMR (400 MHz, CDCl$_3$) δ 8.04 (d, $J$ = 9.3 Hz, 1 H), 7.92 (d, $J$ = 2.8 Hz, 1 H), 7.85 (d, $J$ = 2.8 Hz, 1 H), 7.41 (dd, $J$ = 9.3, 2.8 Hz, 1 H), 7.27 (d, $J$ = 2.7 Hz, 1 H), 5.42 (br s, 1 H), 5.21 (s, 2 H), 4.43 (br s, 1 H), 4.04 (s, 5 H), 4.01 (s, 5 H), 3.19 (d, $J$ = 6.7 Hz, 2 H), 2.93 (t, $J$ = 7.3 Hz, 2 H), 1.62 (t, $J$ = 7.0 Hz, 3 H), 1.39 (s, 9 H). [13]C NMR (101 MHz, CDCl$_3$) δ 158.6, 158.0, 156.0, 143.4, 142.7, 132.0, 131.3, 130.8, 124.2, 122.2, 102.1, 101.6, 65.1, 55.8, 55.8, 39.6, 33.7, 30.8, 28.5. HRMS(ESI) [M + H]$^+$ calculated for C$_{24}$H$_{31}$N$_2$O$_5$S: 459.1948, found: 459.1950.

### *tert*-butyl-(S)–3-(((4-(hydroxymethyl)–2,7-dimethoxyacridin-9-yl)thio) methyl)pyrrolidine-1-carboxylate (17b). 68%

[1] H NMR (400 MHz, CDCl$_3$) δ 8.06 (d, $J$ = 9.3 Hz, 1 H), 7.92 (d, $J$ = 2.6 Hz, 1 H), 7.85 (d, $J$ = 2.3 Hz, 1 H), 7.42 (dd, $J$ = 9.3, 2.5 Hz, 1 H), 7.29 (s, 1 H), 5.36 (br s, 1 H), 5.21 (s, 2 H), 4.05 (s, 3 H), 4.02 (s, 3 H), 3.62–3.32 (m, 2 H), 3.27–2.84 (m, 4 H), 2.15–2.05 (d, $J$ = 6.4 Hz, 1 H), 2.01–1.92 (m, 1 H), 1.70–1.60 (m, 1 H), 1.43 (d, $J$ = 4.8 Hz, 9 H). [13]C NMR (126 MHz, CDCl$_3$) δ 158.6, 158.0, 154.4, 143.4, 142.7, 140.5, 136.5, 131.9, 131.1, 130.5, 124.1, 122.1, 101.9, 101.3, 79.3, 64.9, 55.7, 51.1 (d, $J$ = 115 Hz), 45.1 (d, $J$ = 135 Hz), 39.4, 38.4, 31.2, 30.6, 28.5. HRMS(ESI) [M + H]$^+$ calculated for C$_{26}$H$_{33}$N$_2$O$_5$S: 485.2105, found: 485.2110.

### *tert*-butyl-(R)–3-(((4-(hydroxymethyl)–2,7-dimethoxyacridin-9-yl)thio) methyl)pyrrolidine-1-carboxylate (18b). 64%

[1] H NMR (400 MHz, CDCl$_3$) δ 8.05 (d, $J$ = 9.4 Hz, 1 H), 7.90 (d, $J$ = 2.7 Hz, 1 H), 7.83 (d, $J$ = 2.6 Hz, 1 H), 7.41 (dd, $J$ = 9.3, 2.7 Hz, 1 H), 7.28 (s, 1 H), 5.38 (br s, 1 H), 5.21 (s, 2 H), 4.04 (s, 3 H), 4.02 (s, 3 H), 3.64–3.20 (m, 4 H), 3.08–2.87 (m, 2 H), 2.14–2.03 (m, 1 H), 1.99–1.90 (m, 1 H), 1.68–1.57 (m, 1 H), 1.42 (d, $J$ = 5.2 Hz, 9 H). [13]C NMR (126 MHz, CDCl$_3$) δ 158.6, 158.0, 154.4, 143.3, 142.7, 140.5, 136.4, 131.9, 131.1, 130.5, 124.1, 122.0, 101.9, 101.3, 79.3, 64.9, 55.7, 50.9 (d, $J$ = 115 Hz), 45.1 (d, $J$ = 140 Hz), 39.4, 38.4, 31.2, 30.6, 28.5. HRMS(ESI) [M + H]$^+$ calculated for C$_{26}$H$_{33}$N$_2$O$_5$S: 485.2105, found: 485.2102.

### *tert*-butyl-3-(((4-(hydroxymethyl)–2,7-dimethoxyacridin-9-yl)thio)methyl) azetidine-1-carboxylate (19b). 50%

[1] H NMR (400 MHz, CDCl$_3$) δ 8.08 (d, $J$ = 9.3 Hz, 1 H), 7.87 (d, $J$ = 2.7 Hz, 1 H), 7.80 (d, $J$ = 2.7 Hz, 1 H), 7.43 (dd, $J$ = 9.3, 2.7 Hz, 1 H), 7.30 (d, $J$ = 2.6 Hz, 1 H), 5.22 (s, 2 H), 4.04 (s, 3 H), 4.02 (s, 3 H), 3.85 (t, $J$ = 8.4 Hz, 2 H), 3.59 (dd, $J$ = 8.8, 5.2 Hz, 2 H), 3.12 (d, $J$ = 7.9 Hz, 2 H), 2.24 (td, $J$ = 7.9, 4.1 Hz, 1 H), 1.39 (s, 9 H). [13]C NMR (126 MHz, CDCl$_3$) δ 158.7, 158.1, 156.2, 143.0, 142.5, 140.5, 135.6, 131.8, 131.2, 130.7, 124.3, 122.3, 101.8, 101.2, 79.6, 64.8, 55.7, 55.7, 39.7, 29.0, 28.3. HRMS(ESI) [M + H]$^+$ calculated for C$_{25}$H$_{31}$N$_2$O$_5$S: 471.1948, found: 471.1939.

### (2,7-dimethoxy-9-((piperidin-4-ylmethyl)thio)acridin-4-yl)methanol (7)

Compound **7b** (24.2 mg, 0.048 mmol) was dissolved in 2 mL of 5% trifluoroacetic acid dichloromethane solution and the mixture was allowed to react at room temperature for 2 h. The solvent was then evaporated and the residue was dissolved with methanol, then purified by HPLC/MS on a Waters Auto Purification LC/MS system (ACQUITY UPLC BEH C18 17 μm 2.1 × 50 mm column) to afford **7** (11.2 mg, 45%) as a dark red solid.[1]H NMR (400 MHz, MeOD) δ 8.09 (d, $J$ = 9.4 Hz, 1 H), 7.76 (d, $J$ = 2.2 Hz, 1 H), 7.71 (d, $J$ = 2.2 Hz, 1 H), 7.54 (d, $J$ = 1.1 Hz, 1 H), 7.50 (dd, $J$ = 9.4, 2.5 Hz, 1 H), 5.23 (s, 2 H), 4.04 (s, 3 H), 4.03 (s, 3 H), 3.33–3.30 (m, 2 H), 2.95 (d, $J$ = 6.7 Hz, 2 H), 2.83 (td, $J$ = 12.8, 1.6 Hz, 2 H), 2.07–1.99 (m, 2 H), 1.66–1.55 (m, 1 H), 1.51–1.40 (m, 2 H). [13]C NMR (126 MHz, MeOD) δ 160.0, 159.8, 144.0, 143.1, 143.0, 137.7, 132.9, 131.7, 131.4, 124.9, 121.8, 102.8, 101.7, 62.2, 56.2, 56.1, 44.8, 42.9, 35.7, 29.4. HRMS(ESI) [M + H]$^+$ calculated for C$_{22}$H$_{27}$N$_2$O$_3$S: 399.1737, found: 399.1732.

## Compounds 16–19

By employment of the above-described procedure, starting from **16b-19b**, compounds **16–19** were prepared.

### (9-((3-aminopropyl)thio)–2,7-dimethoxyacridin-4-yl)methanol (16). Yield 88%

[1] H NMR (400 MHz, MeOD) δ 8.13 (d, $J$ = 9.4 Hz, 1 H), 7.94 (d, $J$ = 26.6 Hz, 2 H), 7.58–7.54 (m, 1 H), 7.50 (dd, $J$ = 9.4, 2.7 Hz, 1 H), 5.31 (s, 2 H), 4.05 (s, 3 H), 4.05 (s, 3 H), 3.09 (t, $J$ = 7.4 Hz, 2 H), 2.95–2.88 (t, $J$ = 7.4 Hz, 2 H), 1.79–1.70 (m, 2 H). [13]C NMR (151 MHz, MeOD) δ 160.3, 160.0, 141.4, 140.9, 140.6, 139.8, 132.1, 131.7, 129.9, 127.0, 124.3, 103.2, 102.3, 62.1, 56.4, 56.3, 39.5, 34.1, 29.4.HRMS(ESI) [M + H]$^+$ calculated for C$_{19}$H$_{23}$N$_2$O$_3$S: 359.1424, found: 359.1428.

### (S)-(2,7-dimethoxy-9-((pyrrolidin-3-ylmethyl)thio)acridin-4-yl)methanol (17). Yield 93%

[1] H NMR (400 MHz, MeOD) δ 8.11 (d, $J$ = 9.4 Hz, 1 H), 7.81 (s, 1 H), 7.75 (s, 1 H), 7.57 (s, 1 H), 7.51 (d, $J$ = 9.4 Hz, 1 H), 5.27 (s, 2 H), 4.07 (s, 3 H), 4.07 (s, 3 H), 3.36–3.27 (m, 2 H), 3.19–3.05 (m, 3 H), 2.96 (dd, $J$ = 11.4, 8.3 Hz, 1 H), 2.25 (hept, $J$ = 7.6 Hz, 1 H), 2.17–2.05 (m, 1 H), 1.80–1.70 (m, 1 H). [13]C NMR (126 MHz, MeOD) δ 160.3, 160.1, 141.8, 141.4, 140.7, 139.9, 131.8, 131.4, 130.8, 126.4, 123.6, 103.0, 102.0, 62.1, 56.3, 56.2, 50.7, 46.2, 40.0, 39.5, 30.9. HRMS(ESI) [M + H]$^+$ calculated for C$_{21}$H$_{25}$N$_2$O$_3$S: 385.1580, found: 385.1580.

### (R)-(2,7-dimethoxy-9-((pyrrolidin-3-ylmethyl)thio)acridin-4-yl)methanol (18). Yield 64%

[1] H NMR (400 MHz, MeOD) δ 8.14 (d, $J$ = 9.3 Hz, 1 H), 7.96 (s, 1 H), 7.89 (s, 1 H), 7.58 (s, 1 H), 7.53–7.48 (m, 1 H), 5.32 (s, 2 H), 4.06 (s, 6 H), 3.35–3.26 (m, 2 H), 3.20–3.08 (m, 3 H), 2.97–2.92 (m, 1 H), 2.26 (hept, $J$ = 7.6 Hz, 1 H), 2.17–2.07 (m, 1 H), 1.80–1.71 (m, 1 H). [13]C NMR (126 MHz, MeOD)

δ 160.2, 160.0, 141.4, 141.1, 140.3, 140.2, 131.7, 131.3, 130.5, 126.5, 123.7, 103.0, 102.0, 62.1, 56.3, 56.2, 50.6, 46.1, 39.9, 39.6, 30.9. HRMS(ESI) [M + H]$^+$ calculated for $C_{21}H_{25}N_2O_3S$: 385.1580, found: 385.1579.

## (9-((azetidin-3-ylmethyl)thio)–2,7-dimethoxyacridin-4-yl)methanol (19). Yield 56%

$^1$H NMR (400 MHz, MeOD) δ 8.07 (d, $J$ = 9.4 Hz, 1 H), 7.80 (d, $J$ = 2.2 Hz, 1 H), 7.73 (d, $J$ = 2.1 Hz, 1 H), 7.53 (s, 1 H), 7.46 (dd, $J$ = 9.3, 2.2 Hz, 1 H), 5.28 (s, 2 H), 4.04 (s, 3 H), 4.03 (s, 3 H), 3.82 (t, $J$ = 9.7 Hz, 2 H), 3.69–3.62 (m, 2 H), 3.26 (d, $J$ = 7.9 Hz, 2 H), 2.71–2.58 (m, 1 H). $^{13}$C NMR (101 MHz, MeOD) δ 160.4, 160.2, 142.4, 142.0, 141.3, 138.0, 132.0, 131.7, 131.5, 126.2, 123.2, 102.7, 101.6, 62.0, 56.3, 56.2, 51.9, 38.9, 33.9. HRMS(ESI) [M + H]$^+$ calculated for $C_{20}H_{23}N_2O_3S$: 371.1424, found: 371.1431.

**Appendix 1—scheme 5.** Synthesis route of C8.

## 2,7-dimethoxy-9-((piperidin-4-ylmethyl)thio)acridine-4-carboxylic acid (8)

0.4 mL of aqueous1.0 M lithium hydroxide solution of was added into a solution of compound **7**a (42 mg, 0.080 mmol) 2 mL of tetrahydrofuran, and the mixture was reacted at 40 ° C for 20 h. After cooling to room temperature, 5 mL of a 10% solution of trifluoroacetic acid in dichloromethane was added to the system, and the mixture was reacted at room temperature for 6 h. The solvent was then evaporated and the residue was dissolved with methanol, then purified by HPLC/MS on a Waters Auto Purification LC/MS system (ACQUITY UPLC BEH C18 17 µm 2.1 × 50 mm column) to afford **8** (21.6 mg, 51%) as a dark red foam. $^1$H NMR (600 MHz, MeOD) δ 8.39 (d, $J$ = 2.8 Hz, 1 H), 8.23 (d, $J$ = 2.7 Hz, 1 H), 8.12 (d, $J$ = 9.3 Hz, 1 H), 7.97 (d, $J$ = 2.5 Hz, 1 H), 7.62 (dd, $J$ = 9.3, 2.6 Hz, 1 H), 4.09 (s, 3 H), 4.07 (s, 3 H), 3.36–3.33 (m, 2 H), 3.02 (d, $J$ = 6.8 Hz, 2 H), 2.89–2.82 (m, 2 H), 2.11 (d, $J$ = 14.0 Hz, 2 H), 1.70–1.64 (m, 1 H), 1.50–1.41 (m, 2 H). $^{13}$C NMR (126 MHz, MeOD) δ 168.7, 160.7, 158.9, 141.9, 141.4, 140.9, 131.6, 131.5, 130.7, 130.5, 128.0, 126.7, 108.6, 103.1, 56.6, 56.5, 44.8, 43.4, 35.7, 29.4. HRMS(ESI) [M + H]$^+$ calculated for $C_{22}H_{25}N_2O_4S$: 413.1530, found: 413.1526.

**Appendix 1—scheme 6.** Synthesis route of C9.

### *tert*-butyl 4-(((4-formyl-2,7-dimethoxyacridin-9-yl)thio)methyl) piperidine-1-carboxylate (9 a)

20.4 µl of oxalyl chloride was dissolved in 1 mL of dry tetrahydrofuran and the mixture was cooled to –78 ° C. 25.6 µl of dimethyl sulfoxide was slowly added to the system, the reaction was stirred at –78 ° C for 30 min, followed by addition of a solution of compound **7b** (60 mg, 0.12 mmol) in 1 mL of tetrahydrofuran solution, and the reaction was continued at –78 ° C for 1 h. 0.10 mL of triethylamine was then added, and the reaction was allowed to return to room temperature for 2 h. The solvent was then evaporated and the residue was purified by chromatography on a silica gel column (dichloromethane / ethyl acetate = 95/5) to gave compound **9** a (56 mg, 94%) as a light yellow solid. $^1$H NMR (400 MHz, CDCl$_3$) δ 11.54 (s, 1 H), 8.19 (d, *J* = 2.9 Hz, 1 H), 8.09 (d, *J* = 9.3 Hz, 1 H), 7.98 (d, *J* = 2.9 Hz, 1 H), 7.86 (d, *J* = 2.6 Hz, 1 H), 7.44 (dd, *J* = 9.4, 2.7 Hz, 1 H), 4.10–3.98 (m, 2 H), 4.04 (s, 3 H), 4.03 (s, 3 H), 2.80 (d, *J* = 6.8 Hz, 2 H), 2.58 (t, *J* = 12.2 Hz, 2 H), 1.83 (m, 2 H), 1.52–1.44 (m, 1 H), 1.43 (s, 9 H), 1.23–1.08 (m, 2 H). $^{13}$C NMR (101 MHz, CDCl$_3$) δ 192.8, 158.9, 157.6, 154.8, 144.5, 142.9, 137.3, 133.5, 132.5, 130.7, 130.7, 125.0, 123.0, 109.0, 101.9, 79.7, 56.0, 55.8, 43.8 (br s), 43.2, 36.9, 31.8, 28.5. HRMS(ESI) [M + H]$^+$ calculated for C$_{27}$H$_{33}$N$_2$O$_5$S: 497.2105, found: 497.2101.

### *tert*-butyl-4-(((4-(aminomethyl)–2,7-dimethoxyacridin-9-yl)thio)methyl) piperidine-1-carboxylate (9b)

Under argon atmosphere, compound **9** a (55 mg, 0.11 mmol), ammonium acetate (85 mg, 1.11 mmol), sodium cyanoborohydride (6.9 mg, 0.11 mmol), 4 mg of 4 A molecular sieves was dissolved in 3 mL of dry methanol at –5 °C, then the reaction was stilled at the same temperature overnight. After the removal of solid by filtration, the solvent was diluted with methylene chloride, washed with brine, and then evaporated under vacuum. The residue was purified by chromatography on a silica gel column (dichloromethane /methanol = 95/5) to gave compound **9b** (25 mg, 46%) as a light yellow solid. $^1$H NMR (400 MHz, CDCl$_3$) δ 7.54 (d, *J* = 2.6 Hz, 1 H), 7.52 (d, *J* = 2.5 Hz, 1 H), 7.30 (d, *J* = 9.3 Hz, 1 H), 7.08 (d, *J* = 2.3 Hz, 1 H), 6.90 (dd, *J* = 9.3, 2.6 Hz, 1 H), 4.86 (s, 2 H), 4.06–3.95 (m, 2 H),4.04 (s, 3 H), 3.96 (s, 3 H), 2.62 (d, *J* = 6.7 Hz, 2 H), 2.52 (t, *J* = 12.0 Hz, 2 H), 1.85–1.70 (m, 2 H), 1.41 (s, 9 H), 1.35–1.30 (m, 1 H), 1.18–1.04 (m, 2 H). $^{13}$C NMR (151 MHz, CDCl$_3$) δ 158.7, 157.2, 154.8, 142.5, 140.9, 137.2, 131.2 (overlapped), 130.0, 129.6, 126.0, 124.0, 103.2, 101.5, 79.7, 55.7 (overlapped), 50.4, 43.6 (br s), 43.1, 36.8, 31.8, 28.5. HRMS(ESI) [M + H]$^+$ calculated for C$_{27}$H$_{36}$N$_3$O$_4$S: 498.2421, found: 498.2420.

### (2,7-dimethoxy-9-((piperidin-4-ylmethyl)thio)acridin-4-yl)methanamine (9)

Compound **9b** (13.5 mg, 0.027 mmol) was dissolved in 2 mL of 5% trifluoroacetic acid dichloromethane solution and the mixture was allowed to react at room temperature for 2 h. The solvent was then evaporated and the residue was dissolved with methanol, then purified by HPLC/MS on a Waters Auto Purification LC/MS system (ACQUITY UPLC BEH C18 17 µm 2.1 × 50 mm column) to afford **9** (6.3 mg, 45%) as a dark red foam. $^1$H NMR (400 MHz, MeOD) δ 7.55 (dd, *J* = 18.2, 2.6 Hz, 2 H), 7.47 (d, *J* = 9.3 Hz, 1 H), 7.17 (d, *J* = 2.5 Hz, 1 H), 6.95 (dd, *J* = 9.3, 2.7 Hz, 1 H), 4.14–4.07 (m, 3 H), 4.02 (s, 3 H), 2.86–2.72 (m, 4 H), 2.07–1.98 (m, 3 H), 1.57–1.31 (m, 4 H). $^{13}$C NMR (151 MHz, MeOD) δ 160.2, 158.7, 143.5, 141.6, 137.9, 132.0, 131.7, 130.9, 130.4, 127.6, 125.2, 104.1, 102.5, 56.3, 56.2, 50.7, 44.8, 42.9, 35.6, 29.4. HRMS(ESI) [M + H]$^+$ calculated for C$_{22}$H$_{28}$N$_3$O$_2$S: 398.1897, found: 398.1896.

**Appendix 1—scheme 7.** Synthesis route of C10.

## (9-chloro-2,7-dimethoxyacridin-4-yl)methanol (10 a)

0.28 mL of 1.0 M DIBAL-H solution in toluene was added to a solution of **1** c (40 mg, 0.12 mmol) in 5 mL of dry dichloromethane at −70 °C, then the reaction was stilled at the same temperature overnight. The reaction was quenched by adding saturated potassium hydrogen tartrate solution, diluted with dichloromethane, washed twice with brine, dried over anhydrous $Na_2SO_4$ and concentrated *in vacuo*. The residue was purified by chromatography on a silica gel column (dichloromethane / ethyl acetate = 90/10) to gave compound **10** a (180 mg, 55%) as a light yellow solid. [1]H NMR (400 MHz, CDCl_3) δ 8.00 (d, *J* = 9.3 Hz, 1 H), 7.44–7.34 (m, 3 H), 7.27 (s, 1 H), 5.28 (br s, 1 H), 5.19 (s, 2 H), 4.02 (s, 3 H), 4.00 (s, 3 H). [13]C NMR (101 MHz, CDCl_3) δ 158.6, 158.0, 143.4, 142.9, 140.1, 136.3, 131.5, 126.0, 125.6, 124.7, 122.5, 99.8, 99.2, 64.8, 55.8, 55.8. HRMS(ESI) [M + H]+ calculated for $C_{16}H_{15}ClNO_3$: 304.0735, found: 304.0732.

## 9-chloro-4-(difluoromethyl)–2,7-dimethoxyacridine (10b)

To a solution of compound **10** a (30 mg, 0.10 mmol) in 3 mL of dichloromethane, Dess-Martin oxidant (60 mg, 0.14 mmol) was added and the reaction was allowed to stir at room temperature for 2 h. The solution was diluted with dichloromethane, washed twice with brine, dried over anhydrous $Na_2SO_4$ and concentrated *in vacuo*. Under argon atmosphere, the crude product was dissolved in 2 mL of dry dichloromethane, 0.1 mL of diethylaminosulfur trifluoride was added, and the mixture was reacted at room temperature overnight. The reaction was quenched by sodium bicarbonate aqueous solution, then diluted with dichloromethane, washed twice with brine, dried over anhydrous $Na_2SO_4$ and concentrated *in vacuo*. The residue was purified by chromatography on a silica gel column (petroleum ether / ethyl acetate = 5/1) to afford **10b** (7.4 mg, 23%) as white solid. [1]H NMR (400 MHz, CDCl_3) δ 8.08 (d, *J* = 9.3 Hz, 1 H), 7.91 (t, *J* = 55.4 Hz, 1 H), 7.76–7.73 (m, 1 H), 7.59 (d, *J* = 2.6 Hz, 1 H), 7.48–7.41 (m, 2 H), 4.05 (s, 3 H), 4.03 (s, 3 H). [13]C NMR (151 MHz, CDCl_3) δ 158.9, 157.6, 144.3, 141.2 (t, *J* = 19 Hz), 135.9, 133.8 (t, *J* = 85 Hz), 132.2, 125.9, 125.6, 125.0, 121.8 (t, *J* = 27 Hz), 112.1 (t, *J* = 942 Hz), 102.4, 99.7, 56.0, 55.9. HRMS(ESI) [M + H]+ calculated for $C_{16}H_{13}ClF_2NO_2$: 324.0597, found: 324.0597.

## *tert*-butyl 4-(((4-(difluoromethyl)–2,7-dimethoxyacridin-9-yl)thio)methyl)piperidine-1-carboxylate (10 c)

To a solution of **10b** (7.4 mg, 0.023 mmol) in 2 mL of anhydrous DMF, sodium hydrosulfide hydrate powder (70%, 1.9 mg, 0.034 mmol) was added under argon atmosphere, and the reaction was stirred at 60 °C overnight. 1-Boc-4-bromomethylpiperidine (12.7 mg, 0.0461 mmoll) and potassium carbonate (15.8 mg, 0.114 mmol) were added into the slurry and the reaction was allowed to react at room temperature overnight. The solvent was then evaporated and the residue was dissolved with dichloromethane and washed with water. The combined organic extracts were dried over anhydrous $Na_2SO_4$ and concentrated *in vacuo*. The residue was purified by flash chromatography on a silica gel column (Petroleum ether / ethyl acetate = 3/1) to give compound **10** c as a yellow solid (5.3 mg, 45%) and used for next step.

## 4-(difluoromethyl)–2,7-dimethoxy-9-((piperidin-4-ylmethyl)thio)acridine (10)

Compound **10** c (5.3 mg, 0.010 mmol) was dissolved in 2 mL of 5% trifluoroacetic acid dichloromethane solution and the mixture was allowed to react at room temperature for 2 h. The solvent was then evaporated and the residue was dissolved with methanol, then purified by HPLC/MS on a Waters Auto Purification LC/MS system (ACQUITY UPLC BEH C18 17 μm 2.1 × 50 mm column) to afford **10** (3.8 mg, 70%) as a yellow solid. [1]H NMR (400 MHz, MeOD) δ 8.08 (d, *J* = 6.3 Hz, 1 H), 7.93 (d, *J* = 2.8 Hz, 1 H), 7.89 (t, *J* = 59.6 Hz, 1 H) 7.89 (d, *J* = 4.1 Hz, 1 H), 7.68 (dd, *J* = 2.8, 1.4 Hz, 1 H), 7.47 (dd, *J* = 9.4, 2.8 Hz, 1 H), 4.09–4.05(overlapped, m, 2 H), 4.06 (s, 3 H), 4.04 (s, 3 H), 2.96 (d, *J* = 6.7 Hz, 2 H), 2.90–2.78 (m, 2 H), 2.11 (d, *J* = 14.4 Hz, 2 H), 1.68–58 (m, 1 H), 1.52–1.43 (m, 2 H). [13]C NMR (151 MHz, MeOD) δ 160.4, 158.9, 145.1, 142.0, 137.7, 135.5 (t, *J* = 85 Hz), 133.4, 131.7, 131.4, 125.6, 122.0 (t, *J* = 27 Hz), 113.3 (t, *J* = 937 Hz), 105.3, 102.7, 56.4, 56.2, 44.8, 43.0, 35.7, 29.5. HRMS(ESI) [M + H]+ calculated for $C_{22}H_{25}F_2N_2O_2S$: 419.1599, found: 419.1587.

**Appendix 1—scheme 8.** General procedure of C11-C14 synthesis.

General procedures of synthesizing **1 c'/1d;** To a suspension of compound **1b** (1.65 g, 6.05 mmol) in 200 mL of dry dichloromethane was slowly added 30 mL of boron tribromide, and the mixture was reacted at 0 ° C for 2 h. Methanol was added to the reaction system to quench the reaction, and the solvent was removed under reduced pressure. The crude product was separated and purified with a silica gel column (dichloromethane / methanol = 92/8) to obtain **1 c'** (0.22 g, 14%) and the reported compound **1d** (0.80 g, 56%), with 0.70 g of the starting material recovered. **1 c':** $^1$H NMR (400 MHz, MeOD) δ 8.08 (dd, $J$ = 9.4, 4.8 Hz, 2 H), 7.64 (dd, $J$ = 7.6, 4.4 Hz, 4 H), 4.07 (s, 3 H). $^{13}$C NMR (151 MHz, MeOD) δ 160.8, 159.2, 129.2, 127.8, 127.2, 104.6, 101.3, 56.6. HRMS(ESI) [M + H]$^+$ calculated for $C_{14}H_{11}ClNO_2$: 260.0473, found: 260.0471.

General procedures of synthesizing **11 a**,**12a**: To a solution of **1 c'** (94.5 mg, 0.364 mmol, 100 mol%) in 10 mL of anhydrous DMF, potassium carbonate (75.4 mg, 0.546 mmol, 150 mol%) silver oxide (126.5 mg, 0.546 mmol, 150 mol%) and suitable bromides (0.546 mmol, 150 mol%) was added. The reaction was allowed to react at 40 °C until full conversion. The reaction solution was spin-dried under reduced pressure, and water / dichloromethane was separated. The solvent was then evaporated and the residue was dissolved with dichloromethane and washed with water. The combined organic extracts were dried over anhydrous $Na_2SO_4$ and concentrated *in vacuo*. The residue was purified by chromatography on a silica gel column (Petroleum ether / ethyl acetate = 9/1) to give the desired compound.

## 2-(benzyloxy)–9-chloro-7-methoxyacridine (11 a). Yield 100%

$^1$ H NMR (400 MHz, CDCl$_3$) δ 8.08 (dd, $J$ = 9.3, 7.6 Hz, 2 H), 7.60 (d, $J$ = 2.6 Hz, 1 H), 7.56–7.35 (m, 8 H), 5.27 (s, 2 H), 4.02 (s, 3 H). $^{13}$C NMR (101 MHz, CDCl$_3$) δ 158.4, 157.5, 144.4, 144.3, 136.2, 135.7, 131.6, 131.5, 128.3, 127.9, 128.0, 125.6, 125.6, 124.7, 124.6, 101.2, 99.9, 70.6, 55.8. HRMS(ESI) [M + H]$^+$ calculated for $C_{21}H_{17}ClNO_2$: 350.0942, found: 350.0941.

## 9-chloro-2-isopropoxy-7-methoxyacridine (12 a). Yield 78%

$^1$ H NMR (400 MHz, CDCl$_3$) δ 8.08 (dd, $J$ = 9.4, 2.2 Hz, 2 H), 7.51 (dd, $J$ = 13.1, 2.7 Hz, 2 H), 7.41 (ddd, $J$ = 9.3, 7.5, 2.7 Hz, 2 H), 4.85 (p, $J$ = 6.0 Hz, 1 H), 4.03 (s, 3 H), 1.48 (d, $J$ = 6.0 Hz, 6 H). $^{13}$C NMR (151 MHz, CDCl$_3$) δ 158.5, 156.7, 144.4, 144.3, 135.5, 131.7, 131.7, 125.8, 125.6, 125.5, 124.5, 101.8, 99.9, 70.5, 55.8, 22.0. HRMS(ESI) [M + H]$^+$ calculated for $C_{17}H_{17}ClNO_2$: 302.0942, found: 302.0942.

General procedures of synthesizing **11–14**: To a solution of suitable acridine chloride (100 mol%) in 2 mL of anhydrous DMF, sodium hydrosulfide hydrate powder (150 mol%) was added under argon atmosphere, and the reaction was stirred at 50 °C for 3 h. Suitable bromides (200 mol%) and potassium carbonate (500 mol%) were added into the slurry and the reaction was allowed to react at room temperature overnight. The solvent was then evaporated and the residue was dissolved with dichloromethane and washed with water. The combined organic extracts were dried over anhydrous $Na_2SO_4$ and concentrated *in vacuo*. The residue was dissolved in 2 mL of 5% trifluoroacetic acid dichloromethane solution and the mixture was allowed to react at room temperature for 2 h. The solvent was then evaporated and the residue was dissolved with methanol, then purified by HPLC/

MS on a Waters Auto Purification LC/MS system (ACQUITY UPLC BEH C18 17 μm 2.1 × 50 mm column) to afford desired product.

### 3-((2-(benzyloxy)–7-methoxyacridin-9-yl)thio)propan-1-amine (11). Yield 9.6%

[1] H NMR (400 MHz, MeOD) δ 8.13 (dd, J = 12.1, 9.4 Hz, 2 H), 8.01 (dd, J = 8.5, 2.6 Hz, 2 H), 7.81 (dd, J = 9.4, 2.7 Hz, 1 H), 7.72 (dd, J = 9.4, 2.7 Hz, 1 H), 7.60–7.34 (m, 5 H), 5.42 (s, 2 H), 4.08 (s, 3 H), 3.00 (t, J = 7.6 Hz, 2 H), 2.85 (t, J = 7.6 Hz, 2 H), 1.70 (quint, J = 7.6 Hz, 2 H). [13]C NMR (151 MHz, MeOD) δ 160.9, 159.6, 146.7, 138.5, 138.4, 137.6, 132.4, 132.0, 130.0, 129.9, 129.8, 129.4, 128.7, 126.4, 126.2, 105.7, 103.8, 71.8, 56.7, 39.4, 34.6, 29.4. HRMS(ESI) [M + H]$^+$ calculated for $C_{24}H_{25}N_2O_2S$: 405.1631, found: 405.1630.

### 2-(benzyloxy)–7-methoxy-9-(piperidin-4-ylthio)acridine (12). 24%

[1] H NMR (400 MHz, MeOD) δ 7.99 (dd, J = 14.3, 9.4 Hz, 2 H), 7.89 (d, J = 2.4 Hz, 2 H), 7.58–7.50 (m, 3 H), 7.47–7.40 (m, 3 H), 7.39–7.33 (m, 1 H), 5.36 (s, 2 H), 4.01 (s, 3 H), 3.26 (d, J = 13.1 Hz, 2 H), 3.19–3.10 (m, 1 H), 2.81 (t, J = 10.8 Hz, 2 H), 1.86 (d, J = 11.5 Hz, 2 H), 1.70 (td, J = 14.4, 3.7 Hz, 2 H). [13]C NMR (101 MHz, MeOD) δ 159.9, 158.7, 144.7, 144.6, 138.3, 135.9, 132.6, 132.2, 131.8, 131.7, 129.8, 129.1, 128.3, 125.8, 125.6, 105.0, 103.3, 71.2, 56.2, 45.5, 45.1, 32.0. HRMS(ESI) [M + H]$^+$ calculated for $C_{26}H_{27}N_2O_2S$: 431.1788, found: 431.1781.

### 2-(benzyloxy)–7-methoxy-9-((piperidin-4-ylmethyl)thio)acridine (13). 19%

[1] H NMR (400 MHz, MeOD) δ 8.01–7.89 (m, 2 H), 7.86–7.75 (m, 2 H), 7.58–7.31 (m, 7 H), 5.32 (s, 2 H), 3.99 (s, 3 H), 3.27 (d, J = 13.1 Hz, 2 H), 2.75 (t, J = 12.5 Hz, 2 H), 2.68 (d, J = 6.4 Hz, 2 H), 1.94 (d, J = 13.2 Hz, 2 H), 1.44–1.28 (m, 3 H). [13]C NMR (151 MHz, MeOD) δ 160.0, 158.7, 144.7, 144.7, 138.6, 138.3, 132.0, 131.9, 131.7, 131.3, 129.8, 129.2, 125.8, 125.6, 104.9, 103.0, 71.4, 56.2, 45.1, 43.0, 36.0, 30.0. HRMS(ESI) [M + H]$^+$ calculated for $C_{27}H_{29}N_2O_2S$: 445.1944, found: 445.1945.

### 2-isopropoxy-7-methoxy-9-((piperidin-4-ylmethyl)thio)acridine (14). 26%

[1] H NMR (400 MHz, MeOD) δ 8.15 (dd, J = 9.4, 3.6 Hz, 2 H), 8.06 (t, J = 3.0 Hz, 2 H), 7.71 (td, J = 9.3, 2.6 Hz, 2 H), 4.98–4.92 (m, 1 H), 4.09 (s, 3 H), 3.36 (d, J = 12.9 Hz, 2 H), 3.11 (d, J = 6.8 Hz, 2 H), 2.88 (t, J = 11.9 Hz, 2 H), 2.12 (d, J = 14.0 Hz, 2 H), 1.79–1.69 (m, 1 H), 1.54–1.41 (overlapped, m, 2 H), 1.51 (s, 3 H), 1.49 (s, 3 H). [13]C NMR (151 MHz, MeOD) δ 160.8, 158.9, 139.5, 139.3, 132.1, 129.9, 129.0, 127.4, 127.3, 105.6, 103.7, 72.2, 56.6, 44.8, 43.8, 36.0, 29.4, 22.0. HRMS(ESI) [M + H]$^+$ calculated for $C_{23}H_{29}N_2O_2S$: 397.1944, found: 397.1943.

**Appendix 1—scheme 9.** Synthesis route of C15.

### 2,7-bis(benzyloxy)–9-chloroacridine (15 a)

To a solution of 1d (94.5 mg, 0.364 mmol) in 10 mL of anhydrous DMF, potassium carbonate (150.8 mg, 1.092 mmol) silver oxide (253.0 mg, 1.092 mmol) and benzyl bromide (100 μl) was added. The reaction was allowed to react at 40 °C until full conversion. The reaction solution was spin-dried under reduced pressure, and water / dichloromethane was separated. The solvent was then evaporated and the residue was dissolved with dichloromethane and washed with water. The combined organic extracts were dried over anhydrous Na$_2$SO$_4$ and concentrated in vacuo. The residue was purified by chromatography on a silica gel column (Petroleum ether / ethyl acetate = 9/1) to give 15 a (163.9 mg, quant.) as a yellow solid. [1]H NMR (400 MHz, CDCl$_3$) δ 8.10 (d, J =

9.4 Hz, 2 H), 7.62 (d, J = 2.7 Hz, 2 H), 7.57–7.36 (m, 12 H), 5.28 (s, 4 H). $^{13}$C NMR (101 MHz, CDCl$_3$) δ 157.6, 144.5, 136.3, 135.9, 131.8, 128.9, 128.5, 128.0, 125.6, 124.8, 101.2, 70.6. HRMS(ESI) [M + H]$^+$ calculated for C$_{27}$H$_{21}$ClNO$_2$: 426.1255, found: 426.1255.

## 2,7-bis(benzyloxy)–9-((piperidin-4-ylmethyl)thio)acridine (15)

To a solution of **15** a (20 mg, 0.047 mmol) in 2 mL of anhydrous DMF, sodium hydrosulfide hydrate powder (70%, 5.6 mg, 0.070 mmol) was added under argon atmosphere, and the reaction was stirred at 50 °C for 3 h. *tert*-butyl 4-(bromomethyl)piperidine-1-carboxylate (26.1 mg, 0.094 mmol) and potassium carbonate (32.4 mg, 0.235 mol) were added into the slurry and the reaction was allowed to react at room temperature overnight. The solvent was then evaporated and the residue was dissolved with dichloromethane and washed with water. The combined organic extracts were dried over anhydrous Na$_2$SO$_4$ and concentrated *in vacuo*. The residue was dissolved in 2 mL of 5% trifluoroacetic acid dichloromethane solution and the mixture was allowed to react at room temperature for 2 h. The solvent was then evaporated and the residue was dissolved with methanol, then purified by HPLC/MS on a Waters Auto Purification LC/MS system (ACQUITY UPLC BEH C18 17 μm 2.1 × 50 mm column) to afford **15** (10.8 mg, 36%). $^1$H NMR (400 MHz, MeOD) δ 8.09 (d, J = 9.4 Hz, 2 H), 7.85 (d, J = 2.5 Hz, 2 H), 7.73 (dd, J = 9.4, 2.6 Hz, 2 H), 7.43 (ddd, J = 28.6, 28.0, 7.3 Hz, 10 H), 5.37 (s, 4 H), 3.22 (d, J = 12.8 Hz, 2 H), 2.73–2.62 (m, 4 H), 1.75 (d, J = 13.6 Hz, 2 H), 1.41–1.31 (m, 1 H), 1.30–1.16 (m, 2 H). $^{13}$C NMR (101 MHz, MeOD) δ 159.4, 137.6, 137.6, 131.5, 130.3, 129.9, 129.4, 128.5, 125.8, 105.5, 71.8, 44.6, 43.6, 35.8, 29.1. HRMS(ESI) [M + H]$^+$ calculated for C$_{33}$H$_{33}$N$_2$O$_2$S: 521.2257, found: 521.2258.

**Appendix 1—scheme 10.** Synthesis route of C20.

## *tert*-butyl-(S)–3-(((4-(bromomethyl)–2,7-dimethoxyacridin-9-yl)thio) methyl)pyrrolidine-1-carboxylate (20 a). 17b

(13 mg, 0.027 mmol), Triphenylphosphine (10.6 mg, 0.040 mmol), N-bromosuccinimide (7.2 mg, 0.040 mmol) was dissolved in 1 mL of anhydrous tetrahydrofuran and the reaction was stirred at room temperature for 30 min. The reaction was concentrated *in vacuo* and the residue was purified by chromatography on a silica gel column (pure dichloromethane) to give compound **20** a (9.6 mg, 65%) as a yellow solid. $^1$H NMR (400 MHz, CDCl$_3$) δ 8.16 (d, J = 9.3 Hz, 1 H), 7.94 (s, 1 H), 7.91 (s, 1 H), 7.59 (s, 1 H), 7.43 (d, J = 9.2 Hz, 1 H), 5.31 (d, J = 8.9 Hz, 2 H), 4.05 (s, 3 H), 4.03 (s, 3 H), 3.64–3.33 (m, 2 H), 3.30–2.84 (m, 4 H), 2.12 (br s, 1 H), 2.01–1.92 (m, 1 H), 1.67–1.60 (m, 1 H), 1.43 (s, 9 H). $^{13}$C NMR (101 MHz, CDCl$_3$) δ 158.8, 157.8, 154.6, 143.8, 141.7, 138.8, 135.8, 132.9, 130.8, 130.7, 124.7, 124.1, 102.9, 101.8, 79.4, 55.8, 51.0 (d, J = 108 Hz), 50.9, 45.2 (d, J = 112 Hz),

39.5, 31.4, 30.7, 29.7, 28.6. HRMS(ESI) [M + H]$^+$ calculated for $C_{26}H_{32}BrN_2O_4S$: 547.1261, found: 547.1255.

(S)–2-((2,7-dimethoxy-9-((pyrrolidin-3-ylmethyl)thio)acridin-4-yl)methyl)guanidine (20).

Under argon atmosphere, N, N'-di-Boc-guanidine (3.1 mg, 12.1 µmol) and potassium carbonate (1.7 mg, 12.1 µmol) was added into the solution of **20** a (5.5 mg, 10.0 µmol) in 1 mL of anhydrous DMF. The reaction was performed at 50 °C for 2.5 h. The reaction was concentrated *in vacuo* and the residue was dissolved in 4 mL of dichloromethane containing 5% trifluoroacetic acid and reacted at room temperature for 2 h. The solvent was then evaporated and the residue was dissolved with methanol, then purified by HPLC/MS on a Waters Auto Purification LC/MS system (ACQUITY UPLC BEH C18 17 µm 2.1 × 50 mm column) to give compound **20** (2.0 mg, 27%). $^1$H NMR (400 MHz, MeOD) δ 8.11 (d, J = 9.4 Hz, 1 H), 7.96 (d, J = 2.7 Hz, 1 H), 7.94 (d, J = 2.7 Hz, 1 H), 7.52–7.48 (m, 2 H), 5.04 (s, 2 H), 4.07 (s, 3 H), 4.06 (s, 3 H), 3.39–3.32 (m, 2 H), 3.19–2.98 (m, 4 H), 2.28 (hept, J = 7.6 Hz, 1 H), 2.20–2.09 (m, 1 H), 1.83–1.73 (m, 1 H). $^{13}$C NMR (151 MHz, MeOD) δ 160.5, 159.6, 159.0, 144.3, 142.9, 138.5, 137.4, 132.8, 132.1, 131.8, 125.7, 125.1, 103.2, 102.8, 56.3, 50.8, 46.2, 43.1, 40.0, 39.4, 31.0. HRMS(ESI) [M + H]$^+$ calculated for $C_{22}H_{28}N_5O_2S$: 426.1958, found: 426.1946.

**Appendix 1—scheme 11.** Synthesis route of C21.

## ethyl 5-((4-methoxyphenyl)amino)–2-(methylthio)thiazole-4-carboxylate (21 a)

Under argon atmosphere, a solution of a known compound 23 (1.04 g, 5.02 mmol) in 4 mL of anhydrous tetrahydrofuran was added dropwise to a solution of potassium tert-butoxide (0.79 g, 7.02 mmol) in THF at –78 °C, followed by a solution of compound 24 (0.83 g, 5.02 mmol) in 4 mL of anhydrous tetrahydrofuran. The reaction was performed at –78 °C for 0.5 h and then slowly warmed up to room temperature and reacted overnight. The reaction was quenched by adding saturated ammonium chloride, diluted with dichloromethane, washed twice with brine, dried over anhydrous $Na_2SO_4$ and concentrated *in vacuo*. The residue was purified by chromatography on a silica gel column (pure dichloromethane) to give compound **21** a (0.29 g, 18%) as a light yellow solid. $^1$H NMR (400 MHz, CDCl$_3$) δ 9.39 (s, 1 H), 7.17–7.11 (m, 2 H), 6.91–6.83 (m, 2 H), 4.38 (q, J = 7.1 Hz, 2 H), 3.77 (s, 3 H), 2.57 (s, 3 H), 1.39(t, J = 7.1 Hz, 3 H). $^{13}$C NMR (101 MHz, CDCl$_3$) δ 164.7, 158.5, 156.8, 145.4, 134.2, 121.9, 121.5, 114.9, 60.8, 55.5, 17.7, 14.6. HRMS(ESI) [M + H]$^+$ calculated for $C_{14}H_{17}N_2O_3S_2$: 325.0675, found: 325.0673.

## 9-chloro-7-methoxy-2-(methylthio)thiazolo[5,4-b]quinolone (21b)

Compound **21** a (198 mg, 0.61 mmol) was added in a sealed tube, and 5 mL of phosphorus oxychloride was added under argon atmosphere. The reaction was heated at 130 °C overnight. The resulting slurry was poured onto ice with vigorous stirring to make full quenching. The mixture was diluted with dichloromethane, washed twice with brine, dried over anhydrous $Na_2SO_4$ and concentrated *in vacuo*. The residue was purified by chromatography on a silica gel column (pure dichloromethane) to give compound **21b** (74 mg, 41%) as a light yellow solid. $^1$H NMR (400 MHz, CDCl$_3$) δ 7.97 (d, J = 9.2 Hz, 1 H), 7.55 (d, J = 2.8 Hz, 1 H), 7.40 (dd, J = 9.2, 2.8 Hz, 1 H), 4.01 (s, 3 H), 2.88 (s, 3 H). $^{13}$C

NMR (101 MHz, CDCl$_3$) δ 172.1, 158.5, 157.6, 142.6, 142.4, 130.2, 129.5, 126.2, 123.1, 101.8, 55.8, 15.5. HRMS(ESI) [M + H]$^+$ calculated for C$_{12}$H$_{10}$ClN$_2$OS$_2$: 296.9918, found: 296.9917.

## tert-butyl-4-(((7-methoxy-2-(methylthio)thiazolo[5,4-b]quinolin-9-yl)thio)methyl)piperidine-1-carboxylate (21 c)

To a solution of **21b** (0.020 g, 0.067 mmol) in 5 mL of anhydrous DMF, sodium hydrosulfide hydrate powder (70%, 10.8 mg, 0.135 mmol) was added under argon atmosphere and the reaction was stirred at 50 °C for 2 h until full conversion. 1-Boc-4-bromomethylpiperidine (37.5 mg, 0.135 mmol) and potassium carbonate (27.9 mg, 0.202 mmol) were added into the slurry and the reaction was allowed to react at room temperature overnight. The solvent was then evaporated and the residue was dissolved with dichloromethane and washed with water. The combined organic extracts were dried over anhydrous Na$_2$SO$_4$ and concentrated *in vacuo*. The residue was purified by chromatography on a silica gel column (dichloromethane / ethyl acetate = 95/5) to give compound **21** c (14.9 mg, 45%) as a light yellow solid. $^1$H NMR (400 MHz, CDCl$_3$) δ 7.94 (d, *J* = 9.2 Hz, 1 H), 7.85 (d, *J* = 2.8 Hz, 1 H), 7.38 (dd, *J* = 9.2, 2.8 Hz, 1 H), 4.12–4.01 (m, 2 H), 4.00 (s, 3 H), 3.47 (d, *J* = 6.9 Hz, 2 H), 2.84 (s, 3 H), 2.61 (t, *J* = 12.7 Hz, 2 H), 1.86 (d, *J* = 13.1 Hz, 2 H), 1.65–1.57 (m, 1 H), 1.44 (s, 9 H), 1.23–1.14 (m, 2 H). $^{13}$C NMR (101 MHz, CDCl$_3$) δ 169.9, 158.1, 158.0, 154.9, 145.1, 141.7, 132.9, 130.3, 128.7, 122.4, 103.5, 79.6, 55.8, 43.9 (br s), 41.1, 36.9, 31.7, 28.6, 15.5. HRMS(ESI) [M + H]$^+$ calculated for C$_{23}$H$_{30}$N$_3$O$_3$S$_3$: 492.1444, found: 492.1435.

## 7-methoxy-2-(methylthio)–9-((piperidin-4-ylmethyl)thio)thiazolo[5,4-b]quinolone (21)

**21** c (14.9 mg, 0.030 mmol) was dissolved in 2 mL of 5% trifluoroacetic acid dichloromethane solution and the mixture was allowed to react at room temperature for 2 h. The solvent was then evaporated and the residue was dissolved with methanol, then purified by HPLC/MS on a Waters Auto Purification LC/MS system (ACQUITY UPLC BEH C18 17 μm 2.1 × 50 mm column) to afford **21** (14.7 mg, 96%). $^1$H NMR (400 MHz, MeOD) δ 7.85 (d, *J* = 9.2 Hz, 1 H), 7.81 (d, *J* = 2.7 Hz, 1 H), 7.40 (dd, *J* = 9.2, 2.8 Hz, 1 H), 3.98 (s, 3 H), 3.62 (d, *J* = 6.9 Hz, 2 H), 3.37–3.32 (m, 2 H), 2.88 (s, 3 H), 2.91–2.83 (overlapped, m, 2 H), 2.12 (d, *J* = 13.8 Hz, 2 H), 1.82–1.70 (m, 1 H), 1.55–1.42 (m, 2 H). $^{13}$C NMR (151 MHz, MeOD) δ 172.0, 159.6, 158.8, 146.1, 142.4, 133.9, 130.7, 129.6, 123.5, 104.3, 97.5, 56.2, 44.9, 40.7, 35.7, 29.3, 15.7. HRMS(ESI) [M + H]$^+$ calculated for C$_{18}$H$_{22}$N$_3$OS$_3$: 392.0920, found: 392.0920.

**Appendix 1—scheme 12.** Synthesis route of C22.

## Ethyl-5-((4-methoxy-2-(methoxycarbonyl)phenyl)amino)–2-(methylthio)thiazole-4-carboxylate (22 a)

Under argon atmosphere, a solution of a known compound 23 (150 mg, 0.73 mmol) in 1 mL of anhydrous tetrahydrofuran was added dropwise to a solution of potassium tert-butoxide (112 mg, 0.92 mmol) in 8 mL of anhydrous tetrahydrofuran at –60 °C, followed by a solution of known compound

24' (150 mg, 0.66 mmol) in 1 mL of anhydrous tetrahydrofuran. The reaction was performed at –60 °C for 1.5 h and then slowly warmed up to room temperature and reacted overnight. The reaction was quenched by adding saturated ammonium chloride, diluted with dichloromethane, washed twice with brine, dried over anhydrous $Na_2SO_4$ and concentrated *in vacuo*. The residue was purified by chromatography on a silica gel column (pure dichloromethane) to give compound **22** a (69.2 mg, 28%) as a yellow solid. [1]H NMR (400 MHz, $CDCl_3$) δ 11.61 (s, 1 H), 7.58–7.50 (m, 2 H), 7.11 (dd, *J* = 9.0, 3.2 Hz, 1 H), 4.49 (q, *J* = 7.1 Hz, 2 H), 3.97 (s, 3 H), 3.83 (s, 3 H), 2.64 (s, 3 H), 1.42 (t, *J* = 7.1 Hz, 3 H). [13]C NMR (101 MHz, $CDCl_3$) δ 167.1, 163.9, 154.3, 153.6, 147.0, 137.0, 125.6, 121.0, 117.9, 117.4, 115.8, 61.1, 55.9, 52.7, 17.7, 14.7. HRMS(ESI) [M + H]$^+$ calculated for $C_{16}H_{19}N_2O_5S_2$: 383.0730, found: 383.0729.

### Methyl-9-chloro-7-methoxy-2-(methylthio)thiazolo[5,4-b]quinoline-5-carboxylate (22b)

Compound **22** a (50 mg, 0.14 mmol) was added in a sealed tube, and 1 mL of phosphorus oxychloride was added under argon atmosphere. The reaction was heated at 130 °C overnight. The resulting slurry was poured onto ice with vigorous stirring to make full quenching. The mixture was diluted with dichloromethane, washed twice with brine, dried over anhydrous $Na_2SO_4$ and concentrated *in vacuo*. The residue was purified by chromatography on a silica gel column (pure dichloromethane) to give compound **22b** (15 mg, 32%) as a light yellow solid. [1]H NMR (400 MHz, $CDCl_3$) δ 7.69 (s, 2 H), 4.05 (s, 3 H), 4.02 (s, 3 H), 2.88 (s, 3 H). [13]C NMR (101 MHz, $CDCl_3$) δ 173.4, 167.5, 157.2, 142.6, 139.3, 132.7, 129.4, 126.6, 123.5, 105.1, 56.1, 53.0, 15.5. HRMS(ESI) [M + H]$^+$ calculated for $C_{14}H_{12}ClN_2O_3S_2$: 354.9972, found: 354.9973.

### Methyl-9-(((1-(tert-butoxycarbonyl)piperidin-4-yl)methyl)thio)–7-methoxy-2-(methylthio)thiazolo[5,4-b]quinoline-5-carboxylate (22 c)

To a solution of **22b** (15 mg, 0.042 mmol) in 2 mL of anhydrous DMF, sodium hydrosulfide hydrate powder (70%, 6.7 mg, 0.084 mmol) was added under argon atmosphere, and the reaction was stirred at 50 °C for 2 h until full conversion of **1** c. 1-Boc-4-bromomethylpiperidine (29.3 mg, 0.105 mmol) and potassium carbonate (17.4 mg, 0.126 mmol) were added into the slurry and the reaction was allowed to react at room temperature overnight. The solvent was then evaporated and the residue was dissolved with dichloromethane and washed with water. The combined organic extracts were dried over anhydrous $Na_2SO_4$ and concentrated *in vacuo*. The residue was purified by chromatography on a silica gel column (dichloromethane / ethyl acetate = 95/5) to give compound **22 c** (16.2 mg, 70%) as a light yellow solid. [1]H NMR (400 MHz, $CDCl_3$) δ 8.02 (d, *J* = 2.9 Hz, 1 H), 7.65 (d, *J* = 2.8 Hz, 1 H), 4.12–3.96 (overlapped, m, 2 H), 4.04 (s, 3 H), 4.00 (s, 3 H), 3.44 (d, *J* = 6.9 Hz, 2 H), 2.84 (s, 3 H), 2.58 (t, *J* = 12.6 Hz, 2 H), 1.82 (d, *J* = 12.9 Hz, 2 H), 1.61–1.52 (m, 1 H), 1.44 (s, 9 H), 1.23–1.13 (m, 2 H). [13]C NMR (151 MHz, $CDCl_3$) δ 171.2, 167.8, 158.8, 156.6, 154.8, 145.2, 138.5, 132.7, 132.6, 129.1, 122.5, 106.7, 79.4, 55.9, 52.8, 44.3–43.2 (m), 41.1, 36.7, 31.5, 28.4, 15.4. HRMS(ESI) [M + H]$^+$ calculated for $C_{25}H_{32}N_3O_5S_3$: 550.1499, found: 550.1492.

### (7-methoxy-2-(methylthio)–9-((piperidin-4-ylmethyl)thio)thiazolo[5,4-b]quinolin-5-yl)methaol (22)

87 µl of 1.0 M DIBAL-H solution in toluene was added to a solution of **22** c (16.0 mg, 0.029 mmol) in 1 mL of dry dichloromethane at –60 °C, and the reaction was slowly warmed to room temperature and reacted overnight. The reaction was quenched by adding saturated potassium hydrogen tartrate solution, diluted with dichloromethane, washed twice with brine, dried over anhydrous $Na_2SO_4$ and concentrated *in vacuo*. The residue was dissolved in 4 mL of 5% trifluoroacetic acid dichloromethane solution and the mixture was allowed to react at room temperature for 2 h. The solvent was then evaporated and the residue was dissolved with methanol, then purified by HPLC/MS on a Waters Auto Purification LC/MS system (ACQUITY UPLC BEH C18 17 µm 2.1 × 50 mm column) to afford **22** (5.3 mg, 34%). [1]H NMR (400 MHz, MeOD) δ 7.78 (d, *J* = 2.8 Hz, 1 H), 7.52 (d, *J* = 2.7 Hz, 1 H), 5.20 (s, 2 H), 3.98 (s, 3 H), 3.58 (d, *J* = 6.9 Hz, 2 H), 3.37–3.32 (m, 2 H), 2.88 (s, 3 H), 2.88–2.81 (m, 2 H), 2.15–2.08 (m, 2 H), 1.78–1.69 (m, 1 H), 1.55–1.41 (m, 2 H). [13]C NMR (151 MHz, MeOD) δ 172.2, 159.5, 157.9, 146.2, 142.2, 140.4, 133.5, 129.8, 120.6, 102.9, 61.7, 56.0, 44.9, 40.8, 35.7, 29.3, 15.6. HRMS(ESI) [M + H]$^+$ calculated for $C_{19}H_{24}N_3O_2S_3$: 422.1025, found: 422.1013.

# Appendix 2

## Appendix 2—key resources table

| Reagent type (species) or resource | Designation | Source or reference | Identifiers | Additional information |
|---|---|---|---|---|
| cell line (*Homo-sapiens*) | HEK293T | American Type Culture Collection | Cat#: CRL-3216, RRID: CVCL_0063 | |
| cell line (*Homo-sapiens*) | HEK293A | Thermo Fisher | Cat#: R70507 | |
| cell line (*Homo-sapiens*) | HEK293 | American Type Culture Collection | Cat#: CRL-1573, RRID: CVCL_0045 | |
| cell line (*Homo-sapiens*) | U266 | American Type Culture Collection | Cat#: TIB-196, RRID: CVCL_0566 | |
| cell line (*Homo-sapiens*) | HCT116 | China Infrastructure of Cell Line Resources | Cat#: 1101HUM-PUMC000158 | |
| antibody | anti-4EPB1(Rabbit polyclonal) | Cell Signaling Technology | Cat#: 9644, RRID: AB_2097841 | WB (1:1000) |
| antibody | anti-phosphorylated 4E-BP1 (Thr37/46) (Rabbit monoclonal) | Cell Signaling Technology | Cat#: 2855, RRID: AB_560835 | WB (1:1000) |
| antibody | anti-phosphorylated 4E-BP1 (Ser65) (Rabbit monoclonal) | Cell Signaling Technology | Cat#: 9451, RRID: AB_330947 | WB (1:1000) |
| antibody | anti-phosphorylated 4E-BP1 (Thr70) (Rabbit monoclonal) | Cell Signaling Technology | Cat#: 13396, RRID: AB_2798206 | WB (1:1000) |
| antibody | anti-HA (Rabbit monoclonal) | Cell Signaling Technology | Cat#: 3724, RRID: AB_1549585 | WB (1:1000) |
| antibody | anti-FLAG (Mouse monoclonal) | Sigma-Aldrich | Cat#: F3165, RRID: AB_259529 | WB (1:5000) |
| antibody | anti-FLAG (Rabbit monoclonal) | Abcam | Cat#: ab205606 | WB (1:5000) |
| antibody | anti-GFP (Rabbit monoclonal) | Abcam | Cat#: ab183734 | WB (1:5000) |
| antibody | anti-GFP (Mouse monoclonal) | Proteintech | Cat#: 66002–1-Ig, RRID: AB_11182611 | WB (1:5000) |
| antibody | anti-RPT3 (Rabbit polyclonal) | Thermo Fisher Scientific | Cat#: A303-849A-M, RRID: AB_2781512 | WB (1:1000) |
| antibody | anti-pThr25 (Rabbit polyclonal) | *Guo et al., 2016* | N/A | WB (1:500) |
| antibody | anti-GAPDH (Mouse monoclonal) | Transgene Biotechnology | Cat#: HC301-01 | WB (1:5000) |
| antibody | anti-mouse-IgG-HRP (Goat monoclonal) | Transgene Biotechnology | Cat#: HS201-01 | WB (1:5000) |
| antibody | anti-rabbit-IgG-HRP (Goat monoclonal) | Transgene Biotechnology | Cat#: HS101-01 | WB (1:5000) |
| recombinant DNA reagent | GCaMP6f (plasmid) | Xiaowei Chen Lab (Peking University, China) | N/A | |
| recombinant DNA reagent | pEGFP-Orai1 (plasmid) | Xiaowei Chen Lab (Peking University, China) | N/A | |
| recombinant DNA reagent | mCherry-STIM1 (plasmid) | Xiaowei Chen Lab (Peking University, China) | N/A | |
| recombinant DNA reagent | pLL3.7-DYRK2-shRNA (plasmid) | Xing Guo Lab (Zhejiang University, China) | *Guo et al., 2016* | |

*Appendix 2 Continued on next page*

*Appendix 2 Continued*

| Reagent type (species) or resource | Designation | Source or reference | Identifiers | Additional information |
|---|---|---|---|---|
| recombinant DNA reagent | Flag-STIM1 (plasmid) | This paper | N/A | This plasmid was generated by modification of mCherry-STIM1 plasmid. |
| recombinant DNA reagent | pQlinkHx- DYRK2$^{208-552}$ (plasmid) | This paper | N/A | This plasmid was generated by modification of pEGFP-DYRK2 plasmid. |
| recombinant DNA reagent | pQlinkGx- STIM1$^{235-END}$ (plasmid) | This paper | N/A | This plasmid was generated by modification of mCherry-STIM1 plasmid. |
| recombinant DNA reagent | HA-mcherry-DYRK2 (plasmid) | This paper | N/A | This plasmid was generated by modification of pEGFP-DYRK2 plasmid. |
| recombinant DNA reagent | HA-mcherry-DYRK2-D275N (plasmid) | This paper | N/A | This plasmid was generated by modification of pEGFP-DYRK2-D275N plasmid. |
| peptide, recombinant protein | Flag peptide: DYKDDDDK | Smart Lifesciences | Cat#: SLR01002 | |
| peptide, recombinant protein | GST-MARK3 protein | Carna Biosciences | Cat#: 02–122 | |
| peptide, recombinant protein | GST-Haspin protein | Carna Biosciences | Cat#: 05–111 | |
| strain, strain background (*Escherichia coli*) | BL21(DE3) | Sigma-Aldrich | Cat#: CMC0016 | Electrocompetent cells |
| chemical compound, drug | AKTi-1/2 | Selleck | Cat#: S80837 | |
| chemical compound, drug | PD0325901 | Aladdin | Cat#: P125494 | |
| chemical compound, drug | Thapsigargin | Aladdin | Cat#: T135258 | |
| chemical compound, drug | X-tremeGENE 9 DNA Transfection Reagent | Roche | Cat#: 19129300 | |
| chemical compound, drug | Lipofectamine 2000 | Thermo Fisher Scientific | Cat#: 11668019 | |
| chemical compound, drug | protease inhibitor mixture | Roche | Cat#: 11697498001 | |
| chemical compound, drug | phosphatase inhibitor mixtures | Roche | Cat#: 04906837001 | |
| chemical compound, drug | Ionomycin | Sigma-Aldrich | CAS: 56092-81-0 | |
| chemical compound, drug | 2-Bromo-5-methoxybenzoic acid | J&K Scientific | CAS: 22921-68-2 | |
| chemical compound, drug | 2-Amino-5-Methoxybenzoic acid | Energy Chemicals | CAS: 6705-03-9 | |
| chemical compound, drug | p-Anisidine | J&K Scientific | CAS: 104-94-9 | |
| chemical compound, drug | 1-Boc-4-Bromomethylpiperidine | Bide Pharmatech | CAS: 158407-04-6 | |
| chemical compound, drug | (S)–1-Boc-3-(Bromomethyl) pyrrolidine | Bide Pharmatech | CAS: 1067230-64-1 | |

*Appendix 2 Continued*

| Reagent type (species) or resource | Designation | Source or reference | Identifiers | Additional information |
|---|---|---|---|---|
| chemical compound, drug | (R)–1-Boc-3-(Bromomethyl)pyrrolidine | Bide Pharmatech Ltd | CAS: 1067230-65-2 | |
| chemical compound, drug | 4-Methoxyphenyl isothiocyanate | Energy Chemicals | CAS: 2284-20-0 | |
| chemical compound, drug | tert-butyl 4-bromopiperidine-1-carboxylate | J&K Scientific | CAS: 180695-79-8 | |
| chemical compound, drug | DIBAL-H | Alfa Aesar Chemicals | CAS: 1191-15-7 | |
| chemical compound, drug | Dess-Martin | Alfa Aesar Chemicals | CAS: 87413-09-0 | |
| chemical compound, drug | Boron Tribromide | Sigma-Aldrich | CAS: 10294-33-4 | |
| chemical compound, drug | Urea | Sigma-Aldrich | CAS: 57-13-6 | |
| chemical compound, drug | 2-Amino-2-(hydroxymethyl)–1,3-propanediol | Sigma-Aldrich | CAS: 77-86-1 | |
| chemical compound, drug | Sodium orthovanadate | NEW ENGLAND BioLabs | Cat#: P0758S | |
| commercial assay or kit | Ni Sepharose 6 Fast Flow | GE healthcare | Cat#: 17531803 | |
| commercial assay or kit | Glutathione Sepharose 4B beads | GE healthcare | Cat#: 17-0756-05 | |
| commercial assay or kit | ANTI-FLAG M2 Affinity Gel | Sigma-Aldrich | Cat#: A2220 | |
| commercial assay or kit | Superdex 200 Increase 10/300 GL | GE healthcare | Cat#: 28990944 | |
| commercial assay or kit | BCA Protein Assay Kit Pierce | Thermo-Pierce | Cat#: 23,227 | |
| commercial assay or kit | ADP-Glo kinase assay | Promega | Cat#: V9102 | |
| software, algorithm | Chembiodraw | http://www.perkinelmer.co.uk/category/chemdraw | RRID:SCR_016768 v13 | |
| software, algorithm | GraphPad Prism | GraphPad Software | RRID:SCR_002798 v8.4.0 | |
| software, algorithm | ImageJ (Fiji) | *Schindelin et al., 2012* | RRID:SCR_003070 | |
| software, algorithm | Matlab | https://ww2.mathworks.cn/products/matlab.html | N/A | v2014a |
| software, algorithm | HKL-2000 | HKL Research | RRID:SCR_015547 | |
| software, algorithm | Phenix | https://www.phenix-online.org/ | RRID:SCR_014224 v1.19.2 | |
| software, algorithm | Coot | http://www2.mrc-lmb.cam.ac.uk/personal/pemsley/coot/ | RRID:SCR_014222 v0.9 | |
| software, algorithm | Maxquant | http://www.biochem.mpg.de/5111795/maxquant | RRID: SCR_014485 | v1.5.5.1 |
| software, algorithm | Perseus | http://coxdocs.org/doku.php?id=perseus:start | RRID: SCR_015753 | v1.5.5.3 |
| software, algorithm | Thermo Xcalibur | https://www.thermofisher.cn/order/catalog/product/OPTON-30965 | RRID: SCR_014593 | v4.1.50 |

