## [Editor Report]

This manuscript will be of interest to researchers studying signal transduction and protein kinase function. The authors developed a selective ATP-competitive inhibitor (C17) for the kinase DYRK2 and use this reagent in combination with global phosphoproteomics to identify potential cellular substrates. Two substrate proteins, 4E-BP1 and STIM1 were biochemically validated as DYRK2 substrates. C17 will likely be a useful pharmacological reagent for exploring DYRK2 function and the phosphoproteomic dataset generated in this study serves as a good starting point for understanding the roles that DYRK2 plays in signaling.

---

## [Decision Letter]

**Decision letter after peer review:**

[Editors’ note: the authors submitted for reconsideration following the decision after peer review. What follows is the decision letter after the first round of review.]

Thank you for submitting your work entitled "Selective inhibition reveals the regulatory function of DYRK2 in protein synthesis and calcium entry" for consideration by *eLife*. Your article has been reviewed by 3 peer reviewers, including Hening Lin as the Reviewing Editor and Reviewer #1, and the evaluation has been overseen by a Senior Editor.

Comments to the Authors:

We are sorry to say that, after consultation with the reviewers, we have decided that your work is not suitable for publication at *eLife*.

As you will see from the review comments, although the reviewers found the work interesting and potentially important, they feel that a substantial amount of work will be needed to support the conclusions about the new substrates identified. The decision is not a reflection of a lack of interest from the reviewers, but more a reflection that the time and effort required to carryout these experiments might be exceeding what *eLife*'s policy permits. Thus, we would consider a future new submission after you address the review concerns, if you choose to pursue that direction.

*Reviewer #1:*

DYRK2 is a kinase with multiple reported functions and targeting it has been shown to have potential in treating multiple myeloma and triple-negative breast cancers. The authors in this manuscript aimed to develop better small molecule inhibitors based on their previously reported inhibitor LDN192960, which has an acridine core. They also aimed to gain further information regarding the function of DYRK2 with the inhibitors they developed. By modifying the substituents on the acridine core structure, they made about two dozens of compounds and tested their effects on DYRK2 and related kinases. The best inhibitor, C17, is more potent and selective than previous LDN192960. They further used C17 in a phosphorproteomics study to identify potential substrate proteins of DYRK2. Among the potential substrates, they validated 4E-BP1 and STIM1, and showed that STIM1 phosphorylation promotes its interaction with ORAI1 and store-operated calcium entry.

The manuscript has many strengths and potential impacts: (1) The inhibitors are better than previously reported inhibitors (such as a few fold better in potency and selectivity, and do not target DNA) and will be useful tool compounds to further understand the function of DYRK2 and explore the therapeutic potential of targeting DYRK2; (2) The characterization of the compounds is very thorough and includes X-ray crystal structures of DYRK2 in complex with some of the inhibitors and a kinome scan; (3) The phoshorproteomics may provide a rich information source to understand the function of DYRK2. (4) Identifying STIM1 as new substrate connects DYRK2 to calcium signaling and provides new insights into the function of DYRK2.

The manuscript also has a few minor weaknesses. C17, while very potent and fairly selective, also targets a few other kinases. Thus, the potential substrates identified in the proteomics is only tentative. Similarly, for the 4E-BP1 and STIM1 validation, given that C17 is not completely selective for DYRK2, additional confirmation by DYRK2 knockdown or knockout would be helpful.

For validation of 4E-BP1 and STIM1 as DYRK2 substrates, the authors used C17, purified proteins, and overexpression system. Although this is nice, knockdown or knockout of DYRK2 to show that endogenous 4E-BP1 and STIM1 phosphorylation decreases would be important to support that STIM1 and 4E-BP1 are physiological substrates of DYRK2. This is especially true given that C17 also targets a few other kinases.

In the discussion, the authors mentioned that C17 does not target DNA, unlike LDN192960. This is an important improvement and is worth mentioning earlier in the manuscript.

Given that the authors have previously shown that LDN192960 can inhibit myeloma and triple-negative breast cancers, I am curious whether C17 has better properties than LDN192960 in this perspective. This is just a curiosity and would not affect the conclusion of this manuscript, but if the authors have the information, it might be good to include it here as I believe other readers will also appreciate the information.

*Reviewer #2:*

The authors performed structure-based optimization of a previously identified inhibitor of DYRK2. Through iterative rounds of synthesis, biochemical testing, and structure determination they identified inhibitor C17, which has impressive selectivity for DYRK2 and is sufficiently potent for intracellular studies. C17 is used in combination with phosphoproteomics to identify 100s of potential substrates of DYRK2. The authors validate two potential substrates-4E-BB1 and STIM1-and use C17 to explore how inhibiting DYRK2 affects intracellular processes that STIM1 is involved with. The authors structure-based design efforts are impressive and the inhibitor generated in this study will likely be useful to researchers interested in exploring DYRK2's intracellular function. Furthermore, the phosphoproteomic dataset generated is a nice resource for future exploration. However, the studies into how DYRK2's phosphorylation of 4E-BP1 and STIM1 affects their roles in translation and store-operated calcium entry, respectively, are underdeveloped, limiting the overall biological insight that these efforts provide.

The inhibitor potency and selectivity data are rigorous and all of the proper assays have been performed. The overall medicinal chemistry effort is impressive and noteworthy.

The authors obtained impressive coverage in their phosphoproteomic screen in U266 cells (~15,000 phospho-Ser/Thr peptides). However, it is unclear why the authors used one high single concentration (10 μM) of C17 for their substrate identification efforts. No justification was provided.

Convincing biochemical data that purified DYRK2 is able to phosphorylate 4E-BP1 is provided (Figure 4b, 4c). However, the modest ability of C17 to block 4E-BP1 phosphorylation in cells (Figure 4a, 4d) raises concerns that DYRK2 does not play a significant role in regulating this protein. The fact that C17 can further enhance the combined effects of an Akti and an MEKi on 4E-BP1 phosphorylation is intriguing but the prolonged treatment used (6 hours) makes it unclear how direct this effect is.

The *in vitro* data with purified DYRK2 and STIM1 is confusing and not entirely convincing (Figure 5a). Very high concentrations of DYRK2 (1-5 μM if my math is correct) were used and the gel shift observed with STIM1 is confusing. It appears that quantitative phosphorylation of STIM1 is observed at the lowest concentration of DYRK2 used and that increasing concentrations of this kinase lead to a greater shift, which I interpret as multiple additional phosphorylation events. Furthermore, the authors do not show which sites in their purified STIM1 construct are being phosphorylated by DYRK2.

The results in Figure 5b demonstrating that DYRK2 overexpression leads to an increased interaction between STIM1 and Orail1, which C17 reverses, is intriguing. However, no evidence is provided that the phospho-sites identified (Ser519 and Ser521) in the phosphoproteomic substrate screen are involved in this process. This limits the overall mechanistic insight provided by Figures 5b, 5c. This is also true for the SOCE studies in Figure 5d, 5e. The overall effects observed are interesting but the data shown do not provide sufficient mechanistic insight into how DYRK2 is exactly involved in SOCE.

I feel that this study is a very nice starting point but is insufficiently developed in its exploration of DYRK2 function.

*Reviewer #3:*

Wei and co-workers aim to develop a small molecule inhibitor of DYRK2 and utilize this compound as a chemical probe to identify DYRK2 targets. Structure-guided optimization based on an acridine inhibitor identified in previous studies (LDN192960) culminates in 22 compounds of which compound 17 is the most potent inhibitor of DYRK2 (9 nM). Compound 17 is most selective for DYRK2 although it still exhibits moderate activity (26-87 nM) against other kinases (haspin, DYRK3, MARK3). With compound 17 in hand, the authors perform label-free quantitative phosphoproteomics and select two proteins of the 337 significantly down-regulated phosphosites for validation and follow up. Both proteins, eIF4E and STIM1, are evaluated as substrates of DYRK2 *in vitro* and in cells.

The strengths of this paper are the novel acridine compounds that have strong structural rational and characterization for their activity against DYRK2 and several additional kinases. Compound 17 is marginally more potent than the parent compound LDN192860 against DYRK2 (9 nM vs 13 nM reported in previous work, respectively), but has higher selectivity for the target kinase. *in vitro* this compound is very strong compound for selectively inhibiting DYRK2 and the authors take a fairly comprehensive look at the structure activity relationship and back this up with crystallography data.

However, the cellular demonstration of the new compound 17 in cells is incomplete and relatively weak. The primary weakness is that Compound 17 is first evaluated in transfected HEK293 cells, followed by U266 cells (a myeloma cell line) for phosphoproteomics experiments, and a return to transfected HEK293 cells and a new cell line HCT116 for follow up studies on two targets eIF3E and STIM1, which makes it more difficult to conclude whether or not Compound 17 is effectively hitting DYRK2 in cells. In particular, connections between expected changes in phosphorylation due to compound 17 hitting DYRK2 are not clear when transitioning from HEK293 cells to U266 cells where the phosphoproteomics is conducted, the phosphosites that are evaluated in follow up are not entirely congruent with what was observed in the mass spectrometry experiment or cross-validated back to the different cell lines that are employed. These concerns can be allayed with additional evaluation of compound 17 across cell types and demonstration that reduced phosphorylation events are similar between Compound 17 treatment and knockdown of DRYK2 across the cell lines examined.

The authors use 10 µM of Compound 17 in the phosphoproteomics study in U266 cells, when they show the compound is effective in HEK293T cells at 1 µM concentrations; given the fact that the compound has cross reactivity at higher concentrations of compound, it is difficult to be sure that the phosphosite changes in this dataset are occurring due to DYRK2 inhibition alone. It is unclear whether the phosphosites on RPT3 as a substrate of DYRK2 and the additional phosphosites that are highlighted on eIF4E and STIM1 are observed in this dataset.

There is only Western blot data for DYRK2 in HEK293T cells, but none in U266 cells to validate the mass spectrometry data. The equal number of phosphosites that are reduced and increased in the label-free quantification after 1 h also is indicative of additional targets and pathways that are affected.

The follow up studies with eIF4E are done first *in vitro* and then in HCT116 cells. It is clear that DYRK2 is phosphorylating eIF4E *in vitro*, but unclear if DYRK2 is doing so in cells and Compound 17 is acting as the inhibitor. Western blot data for the enzyme DYRK2 and a knockout study showing similar trends would help clarify.

The mass spectrometry data was not accessible or provided as supplementary tables in the review process. I find it surprising that so many phosphosites could be identified with the phosphoproteomics workflow that is described (1 h run of Ti/IMAC enriched samples without fractionation). Please double check that a stringent FDR at 0.01 for filtering their data for high confidence phosphopeptide assignments, in addition to the site localization metrics that are described. Whether they did that or not is not apparent from the mass spec data methods. The volcano cutoff of is a separate metric (which should really read p-value in the methods section, not FDR).

Figure 4a needs addition of eIF4E expression levels and Flag expression levels, no indication of treatment time in text.

[Editors’ note: further revisions were suggested prior to acceptance, as described below.]

Thank you for resubmitting your work entitled "Selective inhibition reveals the regulatory function of DYRK2 in protein synthesis and calcium entry" for further consideration by *eLife*. Your revised article has been reviewed by 2 peer reviewers, including Hening Lin as the Reviewing Editor and Reviewer #1, and the evaluation has been overseen by Philip Cole as the Senior Editor.

The manuscript has been improved but there are some remaining issues that need to be addressed, as outlined below:

Essential revisions:

1. Please reanalyze the phosphosite data and fix the discrepancy pointed out by Reviewer 2.

2. Revise the statement "this is the largest phosphoproteomic datasest prepared to date for DYRK2 substrate profiling," considering that C17 has activity against other kinases at the concentrations that were used for the phosphoproteomics.

*Reviewer #1:*

DYRK2 is a kinase with multiple reported functions and targeting it has been shown to have potential in treating multiple myeloma and triple-negative breast cancers. The authors in this manuscript aimed to develop better small molecule inhibitors based on their previously reported inhibitor LDN192960, which has an acridine core. They also aimed to gain further information regarding the function of DYRK2 with the inhibitors they developed. By modifying the substituents on the acridine core structure, they made about two dozen compounds and tested their effects on DYRK2 and related kinases. The best inhibitor, C17, is more potent and selective than previous LDN192960. They further used C17 in a phosphorproteomics study to identify potential substrate proteins of DYRK2. Among the potential substrates, they validated 4E-BP1 and STIM1, and showed that STIM1 phosphorylation promotes its interaction with ORAI1 and store-operated calcium entry.

The manuscript has many strengths and potential impacts: (1) The inhibitors are better than previously reported inhibitors (such as a few fold better in potency and selectivity, and do not target DNA) and will be useful tool compounds to further understand the function of DYRK2 and explore the therapeutic potential of targeting DYRK2; (2) The characterization of the compounds is very thorough and includes X-ray crystal structures of DYRK2 in complex with some of the inhibitors and a kinome scan; (3) The phoshorproteomics may provide a rich information source to understand the function of DYRK2. (4) Identifying STIM1 as new substrate connects DYRK2 to calcium signaling and provides new insights into the function of DYRK2. The data presented indicated the STIM1 phosphorylation by DYRK2 promotes STIM1's interaction with Oria1.

The work has only very minor weaknesses. C17, while very potent, is not completely selective. Thus, the potential substrates identified in the proteomics should be considered tentative unless otherwise validated like what was done for STIM1 and 4E-BP1. The therapeutic potential of C17 is also not clear.

The authors have addressed my previous concerns. The mechanistic understanding on the effect of STIM1 phosphorylation is a nice improvement. I support the publication of the manuscript.

*Reviewer #2:*

Wei and co-workers developed a small molecule inhibitor of DYRK2 and utilize this compound as a chemical probe to identify DYRK2 targets. Structure-guided optimization based on an acridine inhibitor identified in previous studies (LDN192960) culminates in 22 compounds of which compound 17 is the most potent inhibitor of DYRK2 (9 nM). Compound 17 is most selective for DYRK2 although it still exhibits moderate activity (26-87 nM) against other kinases (haspin, DYRK3, MARK3). With compound 17 in hand, the authors perform label-free quantitative phosphoproteomics and select two proteins of the 337 significantly down-regulated phosphosites for validation and follow up. Both proteins, eIF4E and STIM1, are evaluated as substrates of DYRK2 *in vitro* and in cells.

The strengths of this paper are the novel acridine compounds that have strong structural rational and characterization for their activity against DYRK2 and several additional kinases. Compound 17 is marginally more potent than the parent compound LDN192860 against DYRK2 (9 nM vs 13 nM reported in previous work, respectively), but has higher selectivity for the target kinase. *in vitro* this compound is very strong compound for selectively inhibiting DYRK2 and the authors take a fairly comprehensive look at the structure activity relationship and back this up with crystallography data. Compound 17 is additionally useful as a probe for discovery of DYRK2 substrates, which the authors demonstrate with phosphoproteomics followed by systematic studies of the DYRK2-dependent phosphorylation of eIF4E and STIM1.

The weaknesses that can be pursued in future studies include a more systematic look at the phosphoproteomic changes elicited by Compound 17 at a lower concentration. In the current dataset, Compound 17 is dosed to cells at 10 µM concentrations, which is likely to affect multiple kinases, based on the measured IC50 values and kinome profiling performed *in vitro*. Therefore in the current data, Compound 17 produces large changes to the phosphoproteome that include, but are not unique to, eIF4E and STIM1.

The authors have significantly enhanced their evaluation of the connection between DYRK2 and eIF4E phosphorylation, using the probe molecule C17. The new data has stronger controls to show the phosphorylation of eIF4E is dependent on DYRK2 activity in cells and *in vitro* (Figure 4) and connection to calcium signaling (Figure 5). This manuscript still needs clarification about the mass spectrometry data analysis, but following the author's careful examination of this data, I am happy to recommend the manuscript for publication in *eLife*.

- The authors report in the manuscript >15,000 phosphosites identified and annotate this in a venn diagram in Figure 3B. However, they report 7583 phosphosites in the volcano plot and in the supplementary table (Figure 3C). This major discrepancy needs to be reanalyzed and corrected.

- Regarding the statement "this is the largest phosphoproteomic datasest prepared to date for DYRK2 substrate profiling," I suggest this be revised as C17 has activity against other kinases at the concentrations that were used for the phosphoproteomics.

---

## [Author Response]

[Editors’ note: the authors resubmitted a revised version of the paper for consideration. What follows is the authors’ response to the first round of review.]

Reviewer #1:[…]For validation of 4E-BP1 and STIM1 as DYRK2 substrates, the authors used C17, purified proteins, and overexpression system. Although this is nice, knockdown or knockout of DYRK2 to show that endogenous 4E-BP1 and STIM1 phosphorylation decreases would be important to support that STIM1 and 4E-BP1 are physiological substrates of DYRK2. This is especially true given that C17 also targets a few other kinases.

We performed a knockdown experiment as suggested and found that knockdown of endogenous DYRK2 using a short hairpin RNA (shRNA) significantly reduced the phosphorylation of 4E-BP1 (Figure 4B).

In the discussion, the authors mentioned that C17 does not target DNA, unlike LDN192960. This is an important improvement and is worth mentioning earlier in the manuscript.

We incorporated the data showing that C17 does not target DNA in the Results section (Figure 2— figure supplement 3).

Given that the authors have previously shown that LDN192960 can inhibit myeloma and triple-negative breast cancers, I am curious whether C17 has better properties than LDN192960 in this perspective. This is just a curiosity and would not affect the conclusion of this manuscript, but if the authors have the information, it might be good to include it here as I believe other readers will also appreciate the information.

As we showed in the paper, C17 is a much more specific DYRK2 inhibitor compared to LDN192960. However, when we examined the pharmacokinetic property of C17, we found that it is not a suitable drug for therapy. In fact, we have made further modifications to C17 to improve the pharmacokinetics and obtained promising therapeutic outcomes in a myeloma model. We will publish these fascinating results in another paper.

Reviewer #2:The authors performed structure-based optimization of a previously identified inhibitor of DYRK2. Through iterative rounds of synthesis, biochemical testing, and structure determination they identified inhibitor C17, which has impressive selectivity for DYRK2 and is sufficiently potent for intracellular studies. C17 is used in combination with phosphoproteomics to identify 100s of potential substrates of DYRK2. The authors validate two potential substrates-4E-BB1 and STIM1-and use C17 to explore how inhibiting DYRK2 affects intracellular processes that STIM1 is involved with. The authors structure-based design efforts are impressive and the inhibitor generated in this study will likely be useful to researchers interested in exploring DYRK2's intracellular function. Furthermore, the phosphoproteomic dataset generated is a nice resource for future exploration. However, the studies into how DYRK2's phosphorylation of 4E-BP1 and STIM1 affects their roles in translation and store-operated calcium entry, respectively, are underdeveloped, limiting the overall biological insight that these efforts provide.The inhibitor potency and selectivity data are rigorous and all of the proper assays have been performed. The overall medicinal chemistry effort is impressive and noteworthy.The authors obtained impressive coverage in their phosphoproteomic screen in U266 cells (~15,000 phospho-Ser/Thr peptides). However, it is unclear why the authors used one high single concentration (10 μM) of C17 for their substrate identification efforts. No justification was provided.

In a separate line of research, we explored the potential of C17 to serve as a drug for myeloma therapy. We found that although C17 is a very potent and specific DYRK2 inhibitor *in vitro*, it is not very stable in vivo, likely due to the labile thioester bond in its structure. In fact, we have made further modifications to C17 to improve the pharmacokinetics, and recently obtained promising therapeutic outcomes in a myeloma model. These fascinating results will be published in another paper. The phosphoproteomics experiment presented in this manuscript was performed before the new compound was available, and a higher concentration of C17 (10 μM) was arbitrarily chosen to ensure a sufficient amount of intact C17 in the cells to elicit its inhibitory function. We want to emphasize that the phosphoproteomics experiment only provided a starting point for us to explore the function of DYRK2 further. We have additionally performed multiple experiments to validate 4EBP1 and STIM1 are genuine targets of DYRK2, as described in our paper.

Convincing biochemical data that purified DYRK2 is able to phosphorylate 4E-BP1 is provided (Figure 4b, 4c). However, the modest ability of C17 to block 4E-BP1 phosphorylation in cells (Figure 4a, 4d) raises concerns that DYRK2 does not play a significant role in regulating this protein. The fact that C17 can further enhance the combined effects of an Akti and an MEKi on 4E-BP1 phosphorylation is intriguing but the prolonged treatment used (6 hours) makes it unclear how direct this effect is.

We have consolidated the experiment using 1-hour treatment in three different cell lines (Figure 4F-H). We have also performed a knockdown experiment using a short hairpin RNA (shRNA) of DRYK2 and found that the knockdown also significantly reduced the phosphorylation of 4E-BP1 (Figure 4B). Indeed, it is well-established that multiple kinases target 4E-BP1. Nevertheless, our data strongly suggest that DYRK2 also plays a role in 4E-BP1 phosphorylation.

The *in vitro* data with purified DYRK2 and STIM1 is confusing and not entirely convincing (Figure 5a). Very high concentrations of DYRK2 (1-5 μM if my math is correct) were used and the gel shift observed with STIM1 is confusing. It appears that quantitative phosphorylation of STIM1 is observed at the lowest concentration of DYRK2 used and that increasing concentrations of this kinase lead to a greater shift, which I interpret as multiple additional phosphorylation events. Furthermore, the authors do not show which sites in their purified STIM1 construct are being phosphorylated by DYRK2.

Indeed, DYRK2 can phosphorylate multiple sites in STIM1. The revised manuscript sought to pinpoint DYRK2-specific phosphorylation sites further using label-free quantitative mass spectrometry analyses. We showed that the phosphorylation levels of at least eight phosphosites on four peptides of STIM1 were significantly reduced upon treatment with C17 compared with the untreated sample (Figure 5—figure supplement 1A-B), including Ser519 and Ser521 that were identified in the U266 phosphoproteome analysis (Figure 3C).

The results in Figure 5b demonstrating that DYRK2 overexpression leads to an increased interaction between STIM1 and Orail1, which C17 reverses, is intriguing. However, no evidence is provided that the phospho-sites identified (Ser519 and Ser521) in the phosphoproteomic substrate screen are involved in this process. This limits the overall mechanistic insight provided by Figures 5b, 5c. This is also true for the SOCE studies in Figure 5d, 5e. The overall effects observed are interesting but the data shown do not provide sufficient mechanistic insight into how DYRK2 is exactly involved in SOCE.

Besides the phosphorylation site mapping experiment described above, we also performed several additional experiments to provide more mechanistic insights into DYRK2’s role in phosphorylating STIM1 and regulating SOCE:

1. To assess the functional outcome of STIM1 phosphorylation by DYRK2, we co-expressed STIM1 and DYRK2 in an Orai-deficient (Orai-KO) cell line. DYRK2 induced the appearance of STIM1 puncta under resting conditions, indicating that DYRK2 promotes STIM1 oligomerization (Figure 5C). In contrast, the STIM1 puncta were not observed in the presence of C17. DYRK2 also failed to promote punctate formation of STIM1-10M, a STIM1 variant with ten potential DYRK2 phosphorylation sites mutated to Ala.

2. We also consolidated the STIM1-Orai1 co-immunoprecipitation experiment. We showed that expression of WT DYRK2 significantly increased the interaction between STIM1 and Orai1, whereas expression of DYRK2-KD exerted no such effect (Figure 5D). Treating cells with C17 effectively abolished the DYRK2-dependent STIM1-Orai1 interaction. Notably, both STIM1-1491, a C-terminal truncated STIM1, and STIM1-10M displayed significantly reduced interaction with Orai1 even in the presence of WT DYRK2 (Figure 5E), suggesting that DYRK2-mediated phosphorylation is essential to promote the binding between STIM1 and Orai1. C17 also decreased the interaction between STIM1 and Orai1 without exogenously expressed DYRK2 (Figure 5F).

3. We further examined fluorescence resonance energy transfer (FRET) between STIM1-YFP and CFP-Orai1 to validate the regulatory function of DYRK2 on STIM1-Orai1 interaction. The FRET signals between STIM1-YFP and CFP-Orai1 were significantly increased in HEK293 cells in the presence of WT DYRK2 (Figure 5G). The FRET signals between STIM1-1-491 and Orai1 were unaltered by DYRK2 (Figure 5H), indicating that the effect of DYRK2 is dependent on the C-terminal region of STIM1. Furthermore, the FRET signals between STIM1-10M and Orai1 were unaffected by DYRK2 (Figure 5I). These results are consistent with the co-immunoprecipitation results and demonstrate that DYRK2 can promote the STIM1-Orai1 interaction via STIM1 phosphorylation.

These new results, together with the SOCE analyses we presented previously (Figure 5J, 5K), provide much more mechanistic insights into the function of DYRK2 in SOCE, and strongly suggest that DRYK2 can directly enhance SOCE by phosphorylating multiple sites in STIM1 and promoting its interaction with ORAI1, which can all be effectively inhibited by C17.

I feel that this study is a very nice starting point but is insufficiently developed in its exploration of DYRK2 function.

As described above, we have performed multiple additional experiments to more rigorously test the direct function of DYRK2 in phosphorylation of 4E-BP1 and STIM1. We believe these results have provided more profound insights into the function of DYRK2.

Reviewer #3:[…]The authors use 10 µM of Compound 17 in the phosphoproteomics study in U266 cells, when they show the compound is effective in HEK293T cells at 1 µM concentrations; given the fact that the compound has cross reactivity at higher concentrations of compound, it is difficult to be sure that the phosphosite changes in this dataset are occurring due to DYRK2 inhibition alone. It is unclear whether the phosphosites on RPT3 as a substrate of DYRK2 and the additional phosphosites that are highlighted on eIF4E and STIM1 are observed in this dataset.

1. As we explained above to Reviewer #2, the U266 phosphoproteomics experiment presented in this manuscript was performed before a new compound with better stability was available, and a higher concentration of C17 (10 μM) was arbitrarily chosen to ensure that there are sufficient amount of intact C17 in the cells to elicit its inhibitory function. Again, we want to emphasize that the phosphoproteomics experiment only provided a starting point for us to explore the function of DYRK2 further. We have additionally performed multiple experiments to validate 4E-BP1 and STIM1 are genuine targets of DYRK2, as described in our paper.

2. Unfortunately, the RPT3 T25 site was not observed in the U266 phosphoproteomics experiment, likely due to a technical reason. The peptide containing the RPT3 T25 site is long and challenging to be enriched and observed in a regular mass spec experiment. Nevertheless, we clearly demonstrated that C17 effectively inhibited DYRK2-mediated Rpt3-T25 phosphorylation (Figure 2D).

3. All phosphosites on eIF4E and STIM1 observed in this dataset are shown in Author response table 1 and Author response table 2:

**Author response table 1. sa2table1:** 

Gene names	phosphosites	Significant	-Log 10 p-valueTreat/Control	log2 fold change (Treat/Control)
EIF4EBP1	T37	+	1.07391	-2.93
EIF4EBP1	S112		0.06947	-0.25
EIF4EBP1	S65		0.07381	-0.04
EIF4EBP1	T70		1.08312	-1.24
EIF4EBP2	T46		0.39658	2.16

**Author response table 2. sa2table2:** 

Gene names	phosphosites	Significant	-Log Student's T-test pvalue Treat_Control	log2 fold change (Treat/Control)
STIM1	S519	+	1.8471	-3.32
STIM1	S521	+	4.45808	-4.63
STIM1	S621	+	1.77212	2.58
STIM1	S628	+	3.32645	1.84
STIM1	S668		0.11102	-0.12
STIM1	S575		2.20548	-1.14
STIM1	S512		0.6069	0.52
STIM1	S618		0.78188	2.17
STIM1	T626		1.61538	1.27

There is only Western blot data for DYRK2 in HEK293T cells, but none in U266 cells to validate the mass spectrometry data. The equal number of phosphosites that are reduced and increased in the label-free quantification after 1 h also is indicative of additional targets and pathways that are affected.

1. In the revised manuscript, we showed that C17 reduced the phosphorylation of 4E-BP1 in three different cell lines, including HEK293A (Figure 4F), HCT116 (Figure 4G), and U266 (Figure 4H).

2. While we agree with the reviewer that additional targets and pathways are likely affected by C17, we believe that we have demonstrated the direct function of DRYK2 in phosphorylating 4E-BP1 and STIM1 in the revised manuscript.

The follow up studies with eIF4E are done first *in vitro* and then in HCT116 cells. It is clear that DYRK2 is phosphorylating eIF4E *in vitro*, but unclear if DYRK2 is doing so in cells and Compound 17 is acting as the inhibitor. Western blot data for the enzyme DYRK2 and a knockout study showing similar trends would help clarify.

As we described above, we have shown that C17 reduced the phosphorylation of 4E-BP1 in three different cell lines, including HEK293A, HCT116, and U266 cells (Figure 4F-H). We also showed that the knockdown of endogenous DYRK2 using a short hairpin RNA (shRNA) significantly reduced the phosphorylation of 4E-BP1 (Figure 4B).

The mass spectrometry data was not accessible or provided as supplementary tables in the review process. I find it surprising that so many phosphosites could be identified with the phosphoproteomics workflow that is described (1 h run of Ti/IMAC enriched samples without fractionation). Please double check that a stringent FDR at 0.01 for filtering their data for high confidence phosphopeptide assignments, in addition to the site localization metrics that are described. Whether they did that or not is not apparent from the mass spec data methods. The volcano cutoff of is a separate metric (which should really read p-value in the methods section, not FDR).

We apologize that the mass spectrometry data was not accessible in the review process. We have uploaded the data to an FTP server “ftp://massive.ucsd.edu/MSV000087106/updates/2021-12-19_MaoYiheng_21671dc9/other/”.

We have checked that FDR<0.01 was used for high confidence phosphopeptide assignments. When the MaxQuant searched the raw files, FDR based on posterior error probability (PEP) was determined by searching a reverse database and was set to 0.01 for both proteins and peptides. This information has been added to the methods in the manuscript.

We thank the reviewer’s thoughtful analysis and are sorry about our inadequate explanation for our cut-off selection in volcano plots. All the volcano plots and cut off were made according to the well-established workflow developed by Mann’s lab, which is routinely available in the Perseus software package (Mol. Cell, 2013, 49, 368; Nat. Methods, 2016, 13, 731). We adopt the default FDR cut-off of 5%, which ensures significant hits with FDR-adjusted p-value < 0.05.

Figure 4a needs addition of eIF4E expression levels and Flag expression levels, no indication of treatment time in text.

We have included the blot for 4E-BP1 expression level in the revised Figure 4A and indicated the treatment time (1 hr) in the figure legend. As this experiment examined the phosphorylation of endogenous 4E-BP1, no Flag blot was involved.

[Editors’ note: what follows is the authors’ response to the second round of review.]

Essential revisions:Reviewer #2:[…]- The authors report in the manuscript >15,000 phosphosites identified and annotate this in a venn diagram in Figure 3B. However, they report 7583 phosphosites in the volcano plot and in the supplementary table (Figure 3C). This major discrepancy needs to be reanalyzed and corrected.

We appreciate this expert reviewer for this very insightful comment. The reasons for the major discrepancy between the identified phosphosites and the sites in the volcano plot are as follows:

The raw table output from the database search software MaxQuant reported 15939 phosphosites. After filtering the reverse and contamination phosphosites, 15755 sites were identified and used to plot the pie chart in Figure 3B. Since these identified phosphosites are reliably identified by MS/MS spectra and properly filtered by wellaccepted software, we concluded the identification of >15,000 phosphosites in Figure 3B.

To further localize the correct S/T/Y amino acids in the identified peptide and quantify their relative abundance, these phosphosites were further processed by Perseus software. The Class I sites with localization probability score of more than 0.75 were kept for further quantitative analysis. The phosphosites with three valid values in at least one treatment or control group were observed. Finally, 7583 sites were visualized in the volcano plot, as shown in Figure 3C.

Collectively, we presented the phosphosite data at identification and quantification levels with rigorous site localization evaluation. Additionally, both the raw table and processed table were uploaded to the FTP server

"ftp://massive.ucsd.edu/MSV000087106/updates/2021-12-

19_MaoYiheng_21671dc9/other/". This is the same FTP server described in the manuscript’s data availability section.

- Regarding the statement "this is the largest phosphoproteomic datasest prepared to date for DYRK2 substrate profiling," I suggest this be revised as C17 has activity against other kinases at the concentrations that were used for the phosphoproteomics.

We thank the reviewer for this critical comment. This statement has been revised to "this is a very comprehensive phosphoproteomic dataset prepared for DYRK2 substrate profiling by treating the U266 cells with 10 μM of C17." in the revised manuscript.